# A nitric-oxide driven chemotactic nanomotor for enhanced immunotherapy of glioblastoma

Huan Chen[1,3], Ting Li[1,3], Zhiyong Liu[1,3], Shuwan Tang[1], Jintao Tong[2], Yingfang Tao[1], Zinan Zhao[1], Nan Li[1], Chun Mao [1] ✉, Jian Shen [1] & Mimi Wan [1] ✉

The major challenges of immunotherapy for glioblastoma are that drugs cannot target tumor sites accurately and properly activate complex immune responses. Herein, we design and prepare a kind of chemotactic nanomotor loaded with brain endothelial cell targeting agent angiopep-2 and anti-tumor drug (Lonidamine modified with mitochondrial targeting agent triphenyl-phosphine, TLND). Reactive oxygen species and inducible nitric oxide synthase (ROS/iNOS), which are specifically highly expressed in glioblastoma microenvironment, are used as chemoattractants to induce the chemotactic behavior of the nanomotors. We propose a precise targeting strategy of brain endothelial cells-tumor cells-mitochondria. Results verified that the released NO and TLND can regulate the immune circulation through multiple steps to enhance the effect of immunotherapy, including triggering the immunogenic cell death of tumor, inducing dendritic cells to mature, promoting cytotoxic T cells infiltration, and regulating tumor microenvironment. Moreover, this treatment strategy can form an effective immune memory effect to prevent tumor metastasis and recurrence.

Glioblastoma (GBM) is one of the most destructive and fatal tumors[1–3]. Immunotherapy can restart and maintain the tumor immune cycle, restore the normal anti-tumor immune response of the body to inhibit and eliminate tumors, thus providing the possibility of avoiding cancer recurrence and treating metastatic tumors[4]. In this process, it will not cause damage to normal cells, so it has attracted extensive attention[5,6]. However, it still faces enormous challenges.

One of the most important challenges faced by immunotherapy is the targeting problem of therapeutic agents. The existence of blood-brain barrier (BBB) seriously hinders the drug delivery efficiency in brain, and it is difficult for drugs to accumulate in brain tumor tissue after penetrating BBB[7]. Currently, the strategies using nanotechnology to improve the efficiency of drug targeting tumors are mainly divided into two categories: chemical recognition and microenvironment response[8,9]. For the former, researchers generally adopt a step-by-step

targeting method, that is, to modify the transporters that can specifically target endothelial cells (such as transferrin, lactoferrin, apolipoprotein E, penetrating peptide) and the components that specifically target tumor cells (such as folic acid, hyaluronic acid, ursolic acid) to break through the BBB and target tumor tissue[8]. Yet, there may be individual differences in chemical recognition strategies, that is, the expression levels of surface receptors of endothelial cells at BBB or tumor cell are different in individuals and brain tumor types[10]. For the latter, inflammatory neutrophils loaded nanodrugs are usually used to achieve the purpose of targeting to brain tumors owing to the fact that the level of inflammatory factor is up-regulated after surgical resection of brain tumors[11]. However, researchers reported that because of its high sensitivity to therapeutic agents, it is only applicable to the postoperative brain tumor environment and cannot respond to the microenvironment of orthotopic tumor[7].

[1]National and Local Joint Engineering Research Center of Biomedical Functional Materials, School of Chemistry and Materials Science, Nanjing Normal University, 210023 Nanjing, China. [2]College of Chemistry and Molecular Engineering, Peking University, 100871 Beijing, China. [3]These authors contributed equally: Huan Chen, Ting Li, Zhiyong Liu. ✉e-mail: maochun@njnu.edu.cn; wanmimi@njnu.edu.cn

Another important challenge faced by immunotherapy is how to properly activate the complex immune mechanism. As early as 2013, researchers proposed the concept of tumor immune cycle, including the following steps. (1, 2) release and presentation of tumor antigen; (3, 4) activation and transport of effector T cells; (5) infiltration of T cells into tumor tissue; (6, 7) recognition and clearance of tumor cells by T cell[12]. Current research often uses different drugs to intervene in the limited steps of the above cycle. For example, chemotherapeutic drugs are often used to induce tumor immunogenic cell death (ICD), so as to expose tumor immune antigen and promote the presentation of tumor antigen (steps 1 and 2)[13]; or immune checkpoint inhibitors are used to enhance the activity of T cells (steps 6 and 7)[14,15]. Yet, due to immune escape in other steps, the intervention effect of limited steps may not be sufficient to effectively enhance the effect of immunotherapy[16]. Moreover, if different drugs are used to interfere with the above steps at the same time, the therapeutic agent and administration mode need to be designed in a complex way, and there is a risk of excessive immune response, namely immune related adverse events (irAEs)[17].

In fact, during its formation, brain tumors will lead to the microenvironment of high concentration of reactive oxygen species (ROS) and inducible nitric oxide synthase (iNOS), which is significantly higher than that of normal tissues. ROS concentration in the tumor environment can be as high as $100\,\mu M$, while the concentration of ROS in normal tissues is only about $20\,nM$[18]. And iNOS is overexpressed in tumor tissue but not in normal tissues[19,20]. This obvious concentration gradient of ROS and iNOS makes it possible to design a targeting strategy that can respond to the microenvironment of brain tumor. Up to now, there is no report that nano-drug delivery systems can achieve the goal of targeting by responding to the concentration gradient of ROS/iNOS and forming a chemotactic behavior. Nanomotors with autonomous mobility are expected to achieve drug delivery in response to microenvironment of GBM through chemotaxis of chemical attractants[21].

Thus, in this work, a kind of nitric oxide (NO)-driven zwitterionic polymer-based nanomotor was designed and constructed, which was obtained by using L-arginine derivatives as monomers (Fig. 1). It is also modified with angiopep-2 (Ang), a polypeptide that can specifically target brain endothelial cells and GBM cells at the BBB site, and then load triphenylphosphine Lonidamine (obtained by modifying Lonidamine (LND) with mitochondrial targeting group triphenylphosphine (TPP), which can destroy tumor metabolism symbiosis, denoted as TLND). The high-density active group guanidine group in the nanomotor can react with ROS/iNOS in the GBM microenvironment of to produce NO. The highly expressed ROS/iNOS in tumor tissue can be used as chemical attractants to induce nanomotors to show chemotaxis to tumor microenvironment (TME) by using the specific affinity of enzyme-substrate[22].

Aiming at the difficulties of brain tumor targeting, we propose a combined step-by-step targeting strategy. First, the brain endothelial cells at BBB site are recognized by the modified polypeptide Ang on the surface of nanomotor, and upregulated ROS/iNOS level in tumor can attract nanomotor to penetrate BBB and target brain tumor tissue through chemotaxis. Meanwhile, with the increase of ROS/iNOS concentration with the increase of tumor tissue density, the deep penetration of nanomotors in tumor tissue can also be achieved. Then, when the nanomotor enters the tumor cell, the loaded TLND can target the mitochondria to play a better role in destroying tumor metabolism symbiosis and promoting tumor apoptosis. Thus, based on the combination of chemical recognition and microenvironment response strategies, a step-by-step targeting strategy for "brain endothelial cells-tumor cell-mitochondria" has been realized.

As for the complex mechanism problems faced by immunotherapy of brain tumors, the nanomotor constructed in this work aims to achieve multi-step intervention synergy in the tumor immune cycle through its beneficial product NO released in the microenvironment of brain tumors and the loaded drug TLND (Fig. 1). The high concentration NO produced in the chemotaxis process of nanomotor can be used as an ICD inducer to enhance the production of tumor immune antigen[23], thus it can promote the maturation of antigen presenting cells and the activation of T cells (steps 1–3)[24]. Furthermore, NO can normalize abnormal blood vessels in tumor tissue, thereby improving the transport efficiency of T cells at tumor sites (step 4)[25]. NO can also up-regulate matrix metalloproteinases by reacting with ROS to produce ONOO⁻, and promote the degradation of tumor extracellular matrix[26], so as to effectively improve the infiltration of T cells and drugs in tumor tissue (step 5); NO can also regulate the polarization of macrophage towards M1 phenotype, inhibit the expression of anti-programmed cell death ligand 1 (PDL1) in cancer cells, and suppress Treg cell level (step 6). Meanwhile, the specific selective inhibition of drug TLND on aerobic glycolysis and energy metabolism in tumor cells makes it possible to improve the tumor immune suppression microenvironment by destroying the tumor metabolic symbiosis process (step 7)[27]. In a word, the nanomotor release NO in response to TME through matrix component (L-arginine derivative), and become an important therapeutic agent for multistep intervention of brain tumor immune enhancement due to the chemotaxis behavior of nanomotors, which constitutes a cascading effect in the treatment process. It should be noted that in this system, there is no direct use of immune checkpoint inhibitors, and multi-step intervention is achieved by promoting NO to give full play to its versatility through nanomotor chemotaxis, which is different from most of the current literature using different drugs to achieve multiple functions (Supplementary Table 1).

## Results

### Characterizations and motion analysis of PAMSe nanomotors
First, the methacrylamidated arginine monomer (Arg-Me) (Supplementary Fig. 1) and the diselenide cross-linker were prepared, respectively (Supplementary Fig. 2). The nuclear magnetic resonance (¹H NMR) spectrum of Arg-Me showed characteristic peaks originating from two hydrogens on C = C bond (Supplementary Fig. 3)[28], and the peaks of molecular weight 243.2 and 241.3 appeared in the mass spectra (Supplementary Fig. 4)[29], which may result from the gain and loss of H atom in Arg-Me. Meantime, the appearance of the characteristic peak of hydrogen in methyl group from the ¹H NMR spectrum of the diselenide cross-linker (Supplementary Fig. 5)[30] confirmed the successful synthesis of diselenide cross-linker. Then, Arg-Me and diselenide crosslinker were used as reactants by adding the initiator azobisisobutyronitrile to obtain PAMSe nanomotors through free radical polymerization reaction (Supplementary Fig. 6), in which P representing polymers, AM representing Arg-Me, Se representing diselenide crosslinker. The morphology and elemental distribution of PAMSe nanomotors were characterized. As shown in Fig. 2a, b and Supplementary Fig. 7, the size of the PAMSe nanomotor was about 200 nm with negative charge ($-34 \pm 0.7$ mV), and TEM-assisted elemental mapping results displayed the existence of Se elements in the particles in addition to C, N, and O elements (Supplementary Fig. 8). The structure of PAMSe can be further characterized by Fourier transform infrared spectroscopy (FTIR) spectrum, in which the peaks at 1650-1550 cm⁻¹ may be attributed to the stretching vibration peak of C = O[31], the in-plane bending vibration of N-H bond and the C = N bond of guanidine group, while the disappearance of the characteristic peak of C = C double bond that originally belonged to the cross-linker may prove that the reaction of free radical polymerization occurred (Supplementary Fig. 9)[32].

Theoretically, compared with the NO-driven nanomotors which L-arginine was loaded by post-loading method, the NO-driven nanomotor constructed in this work, in which the active group (guanidine group in L-arginine) is introduced into the nanomotor substrate by in situ covalent bonding method, can have more effective, stable and

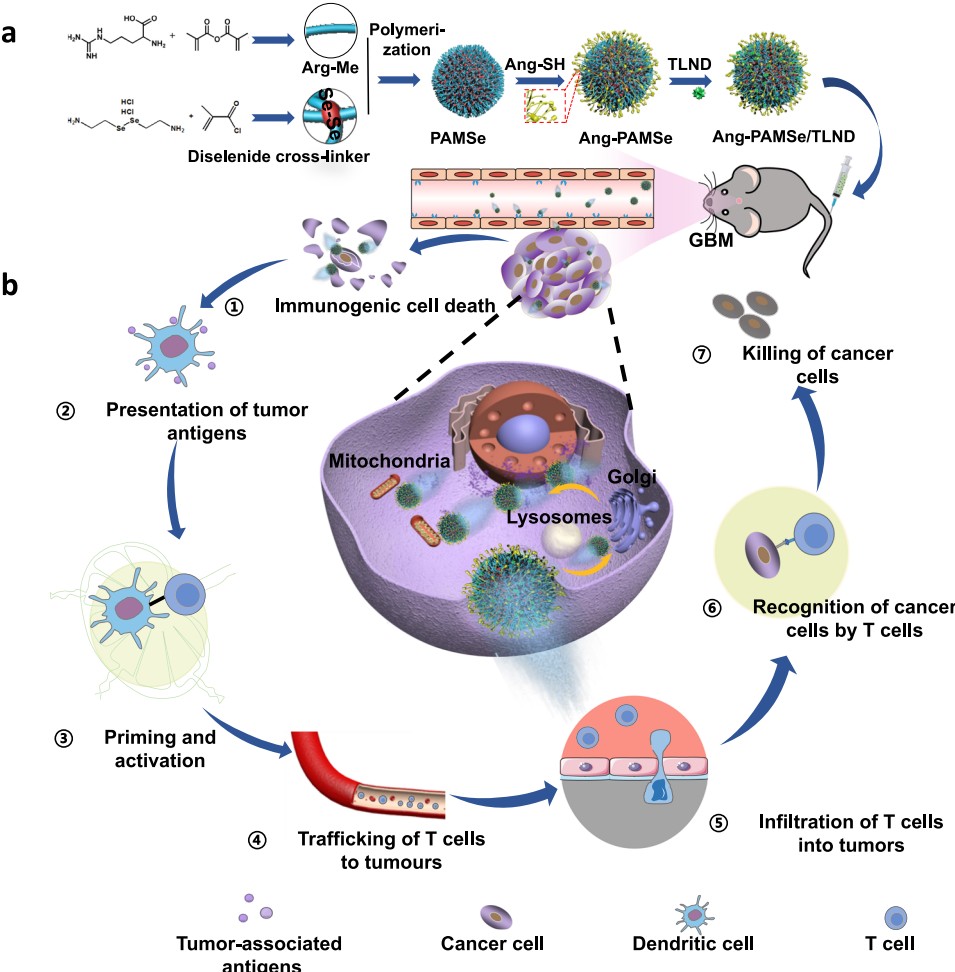

**Fig. 1 | General design of nanomotor and the proposed mechanism of enhancing immunotherapy for glioblastoma. a** Preparation process of nanomotor and **b** the schematic illustration of cascading effect for enhanced immunotherapy of glioblastoma.

durable NO release performance. In order to verify this advantage, we also constructed a kind of NO-driven nanomotor named as PMSe/A, in which methacrylic acid without guanidine group at the end group was used as the monomer and polymerized to prepare PMSe nanoparticles[33] as the carrier, and then L-arginine was post-loaded on the surface of PMSe (refer to Supplementary Fig. 10 for the preparation process). PMSe/A has similar particle size (-200 nm) and surface charge ($-42.9 \pm 1.2$ mV) as PAMSe (Supplementary Fig. 11). Then NO release performance, motility, and uptake efficiency by tumor cells were detected to compare the difference between these two kinds of nanomotors. We first examined the NO release performance of the two types of nanomotors in the tumor cell environment. As shown in Fig. 2c, the PAMSe nanomotor can continuously release NO in the tumor cell environment for at least 96 h, with the release amount of 4.7 µM (meanwhile PAMSe can be degraded in simulated ROS environment (Supplementary Figs. 12 and 13), while the PMSe/A nanomotor can release NO for about 24 h, with a cumulative concentration of only 1.6 µM. Higher NO release also endowed PAMSe nanomotor with better motion ability, and we tracked the trajectory of PAMSe and PMSe/A nanomotors in the tumor cell environment (with high concentration of ROS/iNOS) (Fig. 2d, e, Supplementary Fig. 14, and Supplementary Movies 1−3). Upon analysis of the motion trajectories of both PAMSe and PMS/A nanomotor, the average motion velocity of PAMSe nanomotors in the tumor cell environment was $5.2 \pm 1.0$ µm s$^{-1}$, which was about 1.5 times higher than that of PMS/A ($2.1 \pm 0.6$ µm s$^{-1}$). In addition, the results of cellular uptake (Supplementary Fig. 15) and real-time live

cell imaging (Supplementary Fig. 16 and Supplementary Movie 4) demonstrated that the PAMSe nanomotors were more efficiently taken up by cells after being incubated with cells for 1 h (about 4.1 times as much as the PMSe/A).

## Chemotactic behavior of nanomotors
To validate the chemotaxis of PAMSe nanomotors to highly expressed ROS/iNOS in GBM, we investigated the chemotaxis kinetics of nanomotors along the concentration gradient of chemokines (ROS/iNOS). As shown in Fig. 2f, in order to simulate the concentration gradient of chemokines, agarose gel containing GL261 cancer cell lysate was added to the Y-channel chamber (III) to continuously generate the concentration gradient of chemokines ROS/iNOS, while agarose gel containing bEnd.3 cells lysate was added to the chamber (II), which was used as control[34]. As shown in Supplementary Fig. 17, the concentrations of iNOS and ROS (in the case of superoxide anion, $O_2^{\cdot-}$) at different positions in the Y channel were detected. The results showed that the concentration of the chemoattractant ROS/iNOS was higher at the location containing GL261 lysate (location 1), where the concentration of iNOS was about 2.5 µM and the concentration of $O_2^{\cdot-}$ was about 58.2 nM; the concentration of iNOS at location 5 (the geometric position farthest from the chemical inducer) was about 0.2 µM and the concentration of $O_2^{\cdot-}$ was about 6.8 nM. Therefore, over time, ROS/iNOS can diffuse outward along the geometry of the Y-type channel, creating a concentration gradient. The PAMSe nanomotors or PMSe non-nanomotors were loaded into the Y-type channel of the chamber

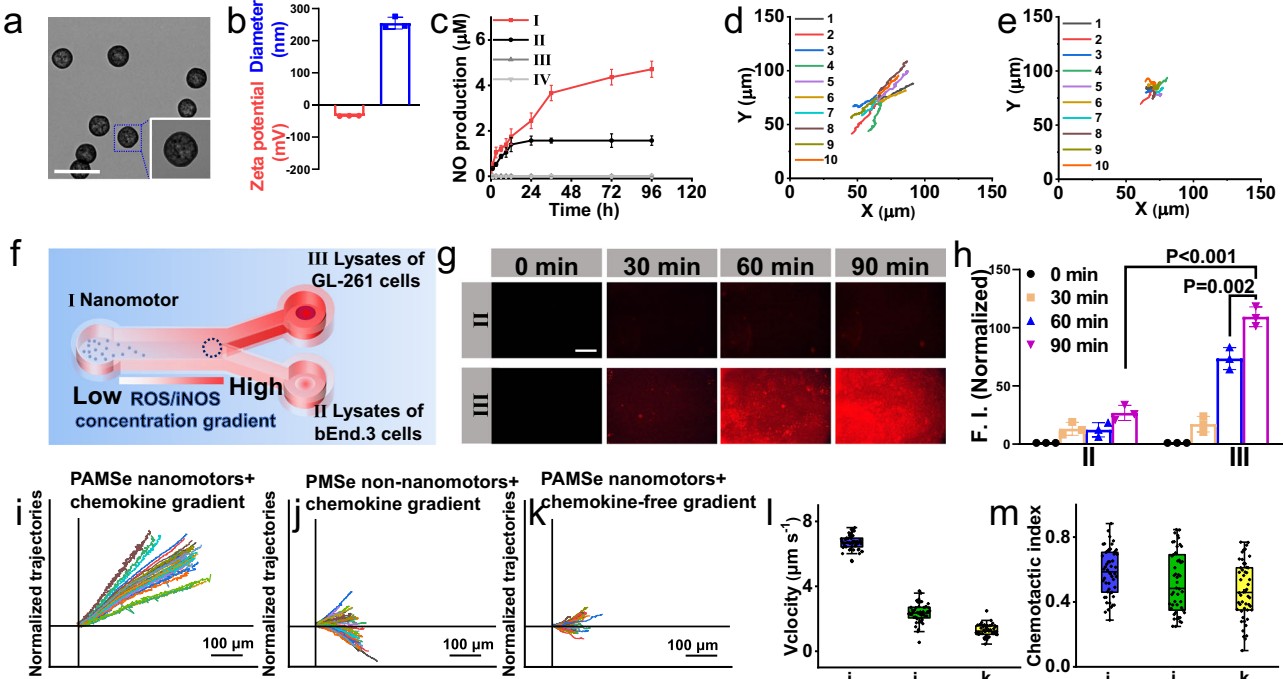

**Fig. 2 | Characterizations of the structure and movement behavior of PAMSe nanomotors. a** TEM image of PAMSe nanomotors (Scale bar: 500 nm) ($n = 3$ independent samples). **b** Zeta potential and particle size of PAMSe nanomotors ($n = 3$ independent samples). **c** NO release profiles from PAMSe nanomotors and post-loaded nanomotors (PMSe/A) in different environments, (I) PAMSe nanomotors and (II) PMSe/A nanomotors in cancer cellular environment, (III) PAMSe nanomotors and (IV) PMSe/A nanomotors in Dulbecco's modified eagle medium (DMEM) solution ($n = 3$ independent samples). Representative trajectories of the **d** PAMSe and **e** PMSe/A nanomotors under cancer cellular environment over 10 s (Supplementary Movies 1–3, samples in a representative experiment ($n = 10$ independent samples). **f** Schematic of chemotactic motion of PAMSe nanomotors along the concentration gradient of chemokine. **g** Fluorescent images (Scale bar: 400 μm) of the different chambers and **h** the corresponding fluorescence quantification values at 0, 30, 60, and 90 min, after adding nanomotors to the left side (I) of the Y-shaped device ($n = 3$ independent samples). Trajectories of **i** PAMSe nanomotors, **j** PMSe non-nanomotors on chemokine concentration gradients, and **k** PAMSe nanomotors on a chemokine-free concentration gradient (the video is intercepted for 100 s) (Supplementary Movie 5). **l** Chemotaxis velocity and **m** chemotactic index (ratio of distance to path length) of samples in **i–k**. Data in **i–m**, $n = 50$ independent samples. Data in box plots **l** and **m** show the mean value and extend from 25 to 75%, while the whiskers extend from the minimal to maximal values, which are based on 50 particles in each group. Data in **b, c, h, l, m** are mean ± s.d. Statistical significance was assessed by one-way ANOVA with post hoc LSD tests. Source data and exact *p* values are provided in the Source data file.

(I), respectively. CLSM was used to continuously monitor the fluorescence of the samples in chamber (II) and chamber (III) (Fig. 2g, h and Supplementary Fig. 18). The results showed that the PAMSe nanomotor moved toward the agarose region containing GL261 cancer cell lysate, and the fluorescence signal captured by PAMSe nanomotor was 4.1 times of that on the other side (at 90 min), while the addition of PMSe non-nanomotor only showed undifferentiated weak fluorescence in both regions (II) and (III) at the same time. In addition, PAMSe nanomotors were loaded in the Y model of agarose gel (without concentration gradient of chemokines), the brain endothelial cell lysate was added to the two channels on the right side, and similarly negligible fluorescence was observed in (II) and (III) after 90 min later. Next, in order to observe the chemotaxis kinetic behavior of PAMSe nanomotors more visually, we set the CLSM lens at the fork on the right side of the Y model (blue circle in Fig. 2f). Figure 2i, j displayed the chemotaxis motion of PAMSe nanomotors and PMSe non-nanomotors in the presence of chemotaxis factor concentration gradients (Supplementary Movie 5). It can be seen that the PAMSe nanomotors moved directionally toward the higher concentration of chemokines, while the PMSe non-nanomotors showed non-selective random diffusion behavior. Also, PAMSe nanomotors also showed only non-selective random diffusion behavior (Fig. 2k) in the environment without concentration gradient of chemokines (agarose gel containing bEnd.3 cells lysate was added to chamber II and III). We also tested the corresponding chemotactic velocity (Fig. 2l) and chemotactic index (the ratio of total displacement to path length) to estimate the migration

persistence (Fig. 2m), and the results showed that the PAMSe nanomotors had a maximum chemotactic velocity of 6.7 μm s⁻¹ and a maximum chemotactic index of 0.6 under the chemokine concentration gradient. All these results demonstrated that PAMSe nanomotors can sense the concentration gradient of chemokines and move to higher concentration areas.

In addition, we placed GL261 cell lysates of different densities ($10^3$, $10^4$, $10^5$, and $10^6$) into the Y-channel (III) and quantified the concentrations of intracellular iNOS and ROS at the corresponding cell densities (Supplementary Fig. 19). Meanwhile, we tracked the fluorescence changes of nanomotors within the Y-channel (III) over 90 min (Supplementary Fig. 20). As shown in Supplementary Fig. 21, the intracellular iNOS concentrations of GL261 cancer cells at densities of $10^3$, $10^4$, $10^5$ and $10^6$ cells mL⁻¹ were approximately 2.0, 4.1, 9.8, and 17.3 μM, and the intracellular $O_2^·$ were approximately 4.4, 11.6, 36.0 and 63.9 nM, respectively, similar to the concentrations reported in other literatures[35]. When the original density of GL261 cells decreased from $10^6$ to $10^3$ cells mL⁻¹, chemotactic behavior of PAMSe nanomotors can no longer be observed within 90 min, indicating that $10^3$ cells mL⁻¹ of GL261 cells may not provide enough chemotactic agent to induce sufficient force chemotactic movement of the nanomotors. In particular, we also detected the directional motion of the PAMSe nanomotors in the microfluidic channels[21] (see the "Methods" section for details). As seen in Fig. 3a–d and Supplementary Movie 6, when cancer cell lysate substrate mimicking the chemokine ROS/iNOS was passed from the lower side channel, a chemotactic movement of the PAMSe

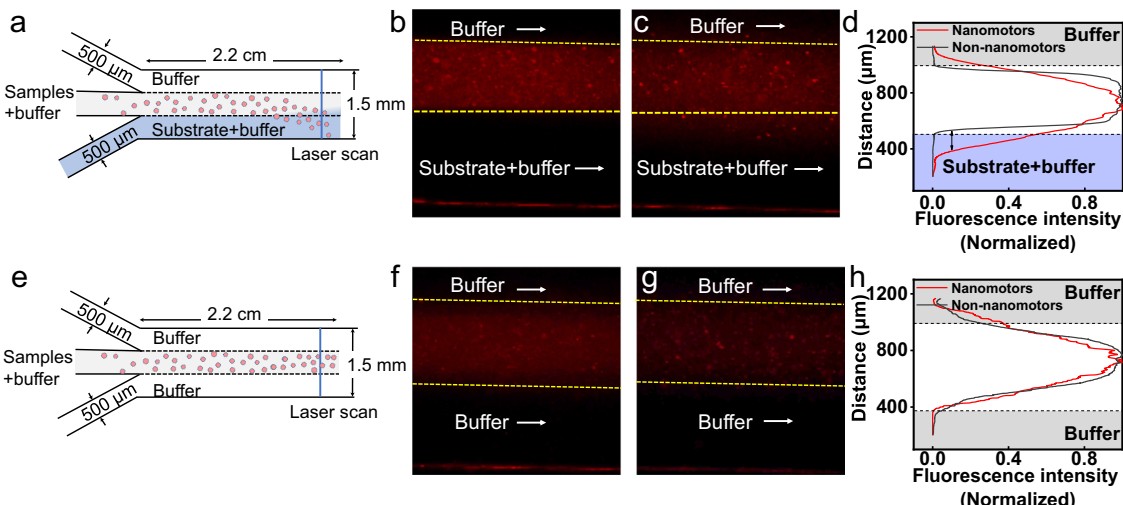

**Fig. 3 | The movement behavior of PAMSe nanomotors in microfluidic device.**
**a** The schematic illustration of three-inlet one-outlet microfluidic device. Typical confocal laser scanning microscopy (CLSM) images of the microfluidic channels for **b** PMSe non-nanomotors or **c** PAMSe nanomotors passing through the middle channel and the substrate (lysate of cancer cells)/buffer pass through the channels on each side (*n* = 3 independent experiment). Supplementary Movie 6, 60s). **d** Fluorescence intensity distribution graph with the normalized fluorescence intensity to a value between 0-1. **e** The schematic illustration and CLSM images of the microfluidic channels for **f** PMSe non-nanomotors or **g** PAMSe nanomotors passing through the middle channel and the buffer pass through both of the branch channels. **h** Fluorescence intensity distribution graph with the normalized (*n* = 3 independent experiment). Typical the volume flow rate of 0.6 mL h$^{-1}$ through each inlet was used to maintain the interaction time.

nanomotor towards the substrate region occurred at about 150 μm (in 1 min). As a control, we also set up a microfluidic device (Fig. 3e), where the buffer solution passed through two channels, and the solutions of PMSe non-nanomotors (Fig. 3f) and PAMSe nanomotors (Fig. 3g) were passed through the middle channel, respectively. Both the distribution of fluorescent fluid from the CLSM images and the normalized results of fluorescence intensity distribution (Fig. 3h) indicated that neither of the two particles showed significant chemotactic behavior (Supplementary Movie 6), which may be attributed to the absence of chemotaxis inducers.

## Cascading effect on the regulation of cellular metabolism by PAMSe/TLND nanomotors

LND can selectively inhibit the aerobic glycolysis and energy metabolism of tumor cells[27], so effective targeting of mitochondria can significantly improve its inhibition effect. To achieve the specific targeting of LND to mitochondria in tumor cells, we prepared TLND by modifying LND with TPP with mitochondrial targeting function (Supplementary Figs. 21 and 22). The structure of TLND was characterized by $^1$H NMR spectrum and its successful synthesis was confirmed (Supplementary Fig. 23)[36]. In this case, TLND can be used as an inhibitor of aerobic glycolysis and energy metabolism in tumor cells to inhibit the succinate-ubiquinone reductase activity of respiratory complex II, thus generating excess ROS[37]. The generated ROS can further act as reactants to promote the process of NO generation from PAMSe in the system and improve the efficiency of nanomotor movement, resulting in a cascading effect as shown in Fig. 4a. We loaded TLND into PAMSe nanomotors with a drug loading capacity (LC%) of about 20.4% and an encapsulation rate (EE %) of about 87.1%. The cumulative TLND drug release curves of PAMSe/TLND in phosphate buffer solution (PBS) and PBS containing 0.002% H$_2$O$_2$ (Supplementary Fig. 24) showed that the drug can be released for at least 120 h in the presence of H$_2$O$_2$, while in PBS solution, the drug release amount was greatly reduced, demonstrating the ROS-responsive release performance of the PAMSe/TLND nanomotor.

To verify the above cascading effect, we incubated cancer cells with TLND, PAMSe nanomotors and PAMSe/TLND nanomotors for 24 h, and continuously monitored the changes of ROS and NO amount in tumor cells with a fluorescent probe for ROS and a fluorescent probe for NO, respectively. As seen in Fig. 4b, c and Supplementary Fig. 25, the addition of TLND increased the ROS production in tumor cells compared with the control group (the fluorescence intensity was about 12 times higher than that of the control group), while the PAMSe nanomotor depleted the intracellular ROS to some extent (its fluorescence intensity was lower than that of the control group). The addition of PAMSe/TLND nanomotors caused the fluorescence intensity of ROS in cells to be between the TLND and PAMSe nanomotor groups. Similarly, as seen in Fig. 4d, e and Supplementary Fig. 26, the PAMSe nanomotors produced NO by consuming ROS in the cancer cell environment compared to the control group (fluorescence intensity was about 14 times higher than

that of the control), whereas the addition of TLND had no significant effect on the cellular NO production. Furthermore, it was detected that the addition of PAMSe/TLND nanomotors has the maximum NO production, which can be attributed to TLND providing sufficient ROS for NO-producing PAMSe, thus demonstrating the existence and role of the cascading reaction.

Then, we focused on evaluating the cascading effect of PAMSe/ TLND nanomotors on the metabolic regulation of tumor cells. As shown in Fig. 4f, the trends of intracellular and extracellular lactate content were examined separately after treatment with different samples. The intracellular lactate content of TLND-treated cells increased to some extent (2.6 mmol g$^{-1}$ protein in 24 h), while the extracellular lactate content did not change much (maintained at about 0.9 mmol g$^{-1}$ protein), which confirmed that TLND can indeed inhibit the lactate excretion from tumor cells. The trend of lactate production in the cells after PAMSe nanomotor treatment was similar to that of the control group, indicating that nanomotors with autonomous motility only have little effect on this metabolic process. Treatment of cells with PAMSe/TLND nanomotors with autonomous motility and with TLND release capacity revealed a significant increase in intracellular lactate content over time (4.2 mmol g$^{-1}$ protein in 24 h) and a slight increase in extracellular lactate content (1.1 mmol g$^{-1}$ protein in 24 h). This result confirmed that PAMSe/TLND nanomotors can induce lactate accumulation in cancer cells. To more clearly describe

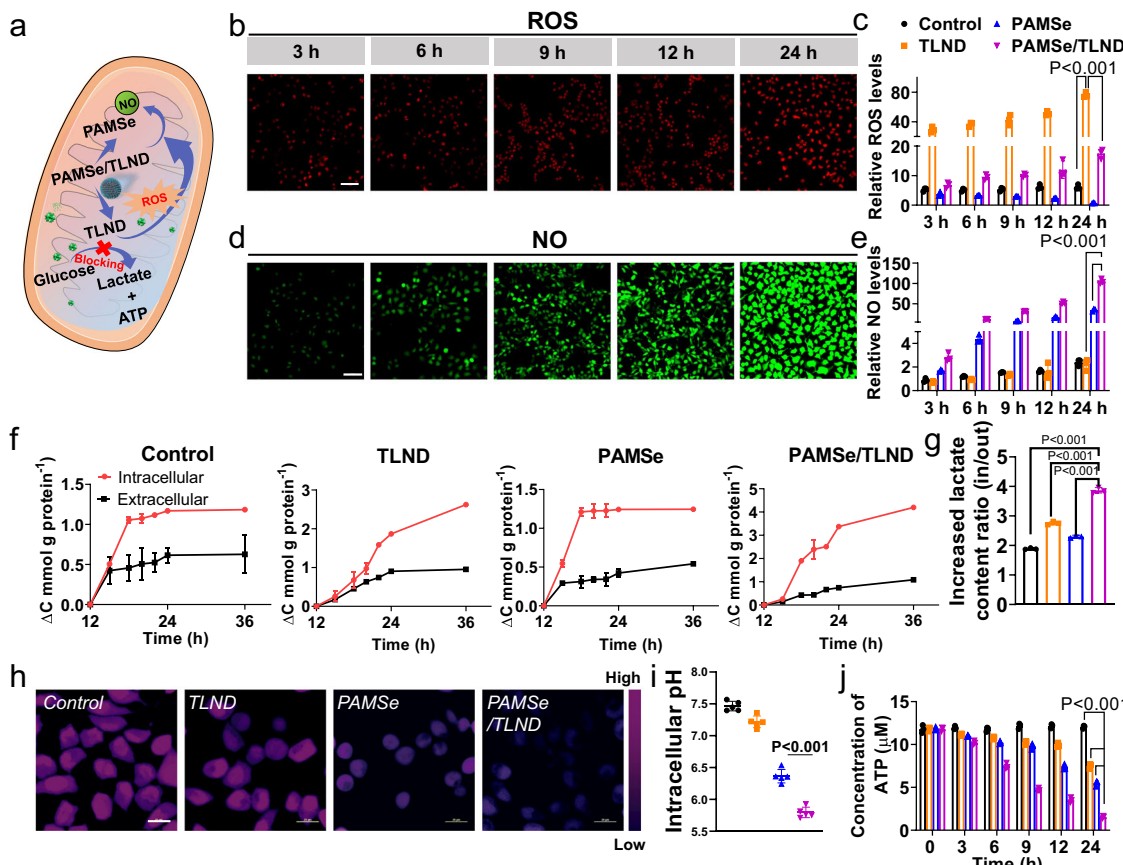

**Fig. 4 | Cascading effect of PAMSe/TLND nanomotors on the regulation of cellular metabolism. a** Schematic illustration of the cellular metabolism regulated by PAMSe/TLND nanomotors. **b** Typical CLSM images of intracellular ROS (labeled with the ROS fluorescent probe, DCFH-DA) in cancer cells after PAMSe/TLND treatment for 24 h ($n = 3$ independent samples, Scale bar: 100 μm). **c** Relative ROS level of cancer cells after different samples treatment for 24 h ($n = 3$ independent samples). **d** Typical CLSM images of intracellular NO (labeled with the NO fluorescent probe, DAF-FM DA) in cancer cells after PAMSe/TLND nanomotors treatment for 24 h ($n = 3$ independent samples, Scale bar: 100 μm). **e** Relative NO level in cancer cells after different samples treatment for 24 h ($n = 3$ independent samples). **f** The intracellular and extracellular increased content of lactic acid after being treated with different samples ($n = 3$ independent samples). **g** The ratio of intracellular and extracellular lactic acid increased content after being treated with different samples ($n = 3$ independent samples). **h** The typical CLSM images with pH fluorescent probes (with pseudo color) ($n = 3$ independent experiment, Scale bar: 20 μm) and **i** corresponding pH values of cancer cells after treatment with different samples for 24 h ($n = 5$ independent samples). **j** The ATP concentration in cancer cells after incubating with different samples for 24 h ($n = 3$ independent samples). The samples in this figure are fresh cell culture medium as control, TLND, PAMSe and PAMSe/TLND. Data in **c**, **e**–**g**, **i**, **j** are mean ± s.d. Statistical significance was assessed by one-way ANOVA with post hoc LSD tests. Source data and exact $p$ values are provided in the Source data file.

this result, Fig. 4g summarizes the ratio of intracellular to extracellular lactate content. After treatment with PAMSe/TLND nanomotors, the ratio of intracellular to extracellular lactate content was 3.9, which was significantly higher than that of free TLND (2.8).

We further assessed whether the accumulation of intracellular lactate induced by PAMSe/TLND nanomotors could lead to acidosis in cancer cells. As shown in Fig. 4h, i, consistent with the trend of intracellular and extracellular lactate accumulation, tumor cells treated with PAMSe/TLND nanomotors exhibited the lowest intracellular pH (~5.9). In addition, we evaluated the effect of PAMSe/TLND nanomotors on the behavior of adenine nucleoside triphosphate (ATP) production in tumor cells. As shown in Fig. 4j, the ATP concentration of both TLND and PAMSe treated cells decreased to some extent over 24 h with the extension of time. The former may be due to the fact that TLND itself, as an inhibitor of hexokinase during anerobic glycolysis of cancer cells, can inhibit the production of ATP, while the latter may be caused by the fact that PAMSe nanomotors produced a certain concentration of NO in the tumor cell environment, which played a certain degree of killing effect on tumor cells. Among the samples, the PAMSe/TLND nanomotors had a significant inhibitory effect on the cellular ATP content with time extension (ATP content decreased from 11.8 to 1.5 μM within 24 h).

## Assessment of the ability of Ang-PAMSe nanomotors to cross the BBB in vitro

The chemotaxis of PAMSe/TLND nanomotors to the sites with higher expression of ROS/iNOS may help nanomotors target TME effectively. For GBM, the first challenge for therapeutic agents is the existence of BBB. Ang is a kind of polypeptide that can bind to low-density lipoprotein receptor-related protein (LRP) and mediate the penetration of BBB. Based on the overexpression of LRP on the surface of cerebral capillary endothelial cells[38], we modified the nanomotors with Ang to help it break through BBB and transport effectively, so that it can play a synergistic role with the chemotaxis of nanomotors to highly expressed ROS/iNOS, thus achieving the accurate targeting of GBM.

We synthesized Ang-modified nanomotors (Ang-PAMSe nanomotors) (Supplementary Fig. 27)[39,40], and the content of Ang in Ang-PAMSe was about 12.0 ± 0.05% (Supplementary Fig. 28). Then, we evaluated the ability of Ang-PAMSe nanomotors to penetrate the BBB model in vitro. Firstly, an in vitro BBB model was established (Fig. 5a). bEnd.3 cells were inoculated in the upper chamber of transwell model to simulate the dense BBB layer (the dense layer was formed after 10 days of culture), and human U87 cells were inoculated in the lower chamber to simulate GBM, and an environment with ROS/iNOS concentration gradient was formed. Then cy5 labeled PMSe, Ang-PMSe

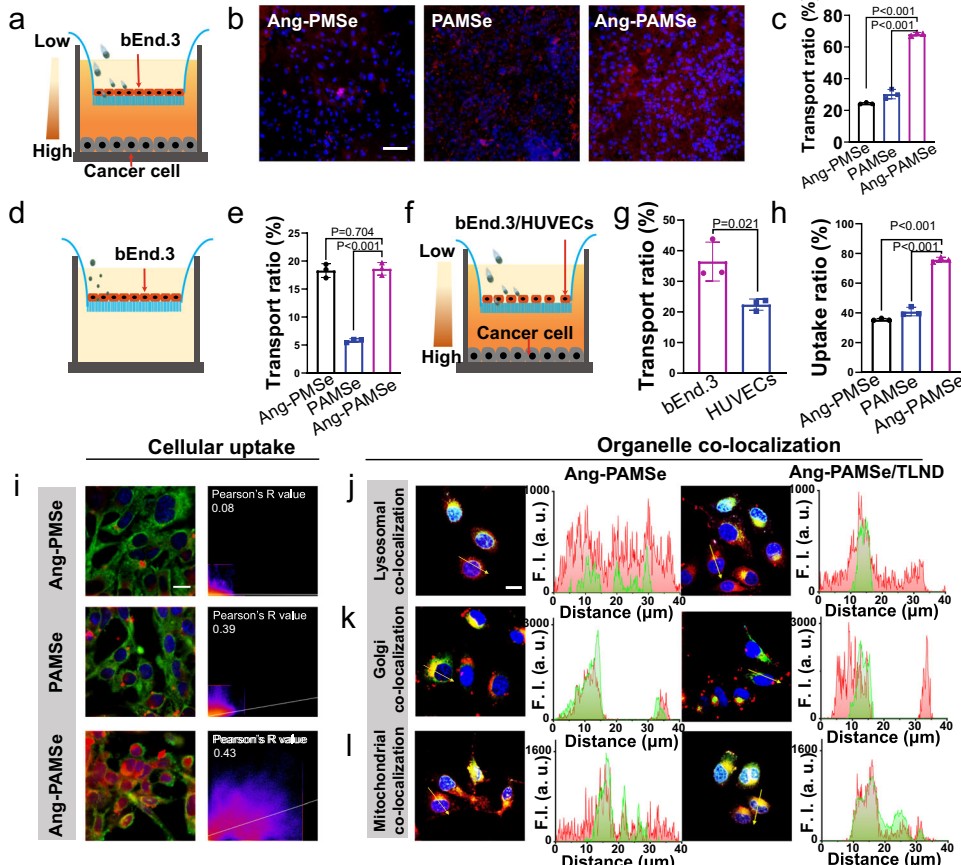

**Fig. 5 | Assessment of the ability of Ang-PAMSe nanomotors to cross the BBB in vitro, and cellular uptake. a** Schematic illustration of the transwell system-based in vitro BBB with ROS/iNOS concentration gradient for evaluating the penetration capability of nanomotor across the endothelial monolayer. **b** Typical CLSM images of the cancer cell layer 24 h after adding the samples) (Scale bar: 100 μm), and **c** its corresponding BBB transport ratio. **d** Schematic illustration of the in vitro BBB model without ROS/iNOS concentration gradient, and **e** BBB transport ratio after different sample treatments. **f** Schematic representation of inoculation of bEnd.3 and HUVECs in the upper chamber. **g** Transport ratio after Ang-PAMSe treatments after 24 h. **h** Cellular uptake ratio for different samples, and **i** its corresponding CLSM images (Green, cell membranes stained with DiO; Red, samples were labeled with cy5; Blue, cell nuclei stained with hochest33342; Scale bar: 20 μm), image J was used to perform a co-localization test on the captured pictures to obtain the firework image and Pearson's R value. **j** CLSM images of co-localization of lysosome with samples **j** after 5 min or **k** after 2 h incubation and corresponding fluorescence curves (Green, lysosome stained with lyso-tracker green/Golgi tracker green; Red, nanomotors were fluorescently labeled with cy5; Blue, cell nuclei stained with hochest33342; Scale bar: 20 μm). **l** CLSM images of co-localization of mitochondrial with samples after 2 h incubation and corresponding fluorescence curves (Green, mitochondria were labeled by mitochondrial probes; Red, samples were fluorescently labeled with cy5; Blue, cell nuclei stained with hochest33342; Scale bar: 20 μm). Data in **b, c, e, g–l** are representative of three independent samples. Data in **c, e, g, h** are mean ± s.d. Statistical significance was assessed by independent samples *t* test (unpaired two-sample *t* test) in **g**. Statistical significance was assessed by one-way ANOVA with post hoc LSD tests. Source data and exact *p* values are provided in the Source data file.

non-nanomotors, PAMSe nanomotors, and Ang-PAMSe nanomotors (200 μg mL$^{-1}$) were added to the upper chamber respectively, and incubated for 24 h. We detected the fluorescence intensity of materials in the lower chamber of transwell model through CLSM observation. Meantime, we collected the liquid and cells in the lower chamber and quantitatively detected the amount of materials in each part. From the CLSM images (Fig. 5b and Supplementary Fig. 29) and the calculation results of transport rate (Fig. 5c and Supplementary Fig. 30), it can be seen that the transport efficiency of PMSe nanoparticles, a passive nanoparticle with neither Ang transporter peptide modification nor motility properties, was only about 7.9%, while Ang-PMSe non-nano-motors (modified with Ang which can mediate the penetration of BBB, but without movement ability) was about 24.4%. The transport efficiency of PAMSe nanomotors (not modified with Ang but with move-ment ability) can reach to 30.3%, The transport rate of Ang-PAMSe nanomotors (modified with Ang receptor and with movement ability) was significantly higher than that of PAMSe nanomotors group, and the transport efficiency was about 68.1%. We speculated that both the autonomous motion performance of nanomotors (chemotaxis to ROS/iNOS) and the transport performance of BBB receptor mediated by Ang play a synergistic role in this process. The integrity of the tight junctions of BBB is particularly important for this study. Therefore, we used CLSM to check the structural integrity of the nanomotors before and after penetrating the BBB. As shown in Supplementary Fig. 31, a dense BBB cell layer had been formed and after treated with Ang-PAMSe nanomotors for 24 h, the BBB cell structure remained dense. Meanwhile, FITC-dextran (FD-4, 4000 Da), a polysaccharide composed of fluorescein isothiocyanate coupled to dextran coupling, was used to identify markers of BBB leakage[41]. As shown in Supplementary Fig. 32, compared with the PBS group, the fluorescence intensity of FD-4 detected in the lower chamber after Ang-PAMSe nanomotor treatment did not increase significantly, which further proved that the Ang-PAMSe nanomotor could penetrate the BBB, and the structural com-pactness of the BBB remained after penetration.

In order to fully verify that the key factors of the high BBB trans-port efficiency of the nanomotors constructed in this work are the microenvironment with chemical inducers and the chemical recogni-tion effect of Ang, we further designed the following experiment. With other conditions remained unchanged, the lower chamber was chan-ged to a DMEM environment without GBM cells, to construct an

environment without ROS/iNOS concentration gradient (Fig. 5d). Then, Ang-PMSe, PAMSe and Ang-PAMSe samples were added respectively. It is worth noting that PAMSe and Ang-PAMSe nanomotors also lost their original chemotaxis, because at this time they were in an environment without ROS/iNOS concentration gradient, and the BBB transport efficiency of PAMSe decreased to 5.8%. Without ROS/iNOS concentration gradient, the transport efficiency of Ang-PAMSe nanomotors was basically the same as that of Ang-PMSe non-nanomotors, which were 18.6 and 18.3%, respectively (Fig. 5e), indicating that Ang mediated receptor transport played a role in this process. However, in the absence of ROS/iNOS concentration gradient, the transport efficiency of Ang-PAMSe nanomotors was significantly lower than that in the presence of ROS/iNOS concentration gradient. This result implied that the chemotaxis of ROS/iNOS played an important role in promoting the transport process of particles across BBB. Then, we inoculated mouse brain endothelial cells bEnd.3 without tight junctions into transwell upper chamber (high expression of LRP) or human umbilical vein endothelial cells (HUVECs, low expression of LRP) (only cultured for 24 h, no dense layer was formed). GBM cells were inoculated in the lower chamber to simulate GBM to form an environment with ROS/iNOS concentration gradient, so as to evaluate the promoting effect of the combination of Ang and LRP on its transport process (Fig. 5f). Ang-PAMSe (200 μg mL$^{-1}$) was added to the above model and incubated for 24 h. Figure 5g showed that the transport efficiency was 36.5% in the bEnd.3 group with LRP receptor overexpression, while that was only 22.4% in HUVECs without LRP receptor overexpression. The above experimental results showed that Ang ligand can bind to LRP overexpressed brain endothelial cells and mediate the BBB transport of nanomotors. Meantime, the autonomous chemotaxis of nanomotors also played an important role in promoting the transport.

In addition, we constructed a transwell model with multilayer cells (bEnd.3 and U-87 MG) to better evaluate the BBB transport performance of nanomotors (Supplementary Fig. 33a). Compared with the single culture of bEnd.3 brain endothelial cell, binary co-culture of bEnd.3 cells with U-87 MG glioma cells can better simulate the microenvironment in vivo, increase tight junctions of BBB, and induce the expression of specific receptors and transporters[42]. As show in Supplementary Fig. 33b, in the multilayer symbiotic BBB model in vitro, the BBB transport rate of Ang-PAMSe nanomotor with both BBB transport performance and motility (61.1%) was significantly higher than that of the passive nanoparticle PMSe (7.0%). The results were similar to that of the monolayer BBB model, which confirmed that the motility effect of the nanomotor and the transport function of Ang can help it penetrate the BBB. After incubation with PMSe and Ang-PAMSe, the fluorescence intensity of FD-4 detected in the transwell lower chamber was not significantly different from that of the control (relative fluorescence intensity was maintained at 764–930 a.u.), indicating that the BBB remained intact after nanomotor penetration (Supplementary Fig. 33c) and the CLSM images of the BBB (Supplementary Fig. 34) also further confirmed this result. Moreover, we constructed an in vitro dynamic 3D BBB model (Supplementary Fig. 35) in order to simulate the flow environment in which the BBB is located in the in vivo environment[43]. After 24 h of circulation of the samples (cy5-PMSe passive nanoparticles and cy5-Ang-PAMSe nanomotors) in liquid media, we detected the $P_{app}$ of the BBB chip. As shown in Supplementary Fig. 36, the ability of Ang-PAMSe nanomotors with both BBB transit function and motility properties to penetrate the vascular system in the BBB channel was 5.7 times that of cy5-PMSe passive nanoparticles.

## Cellular uptake behavior assessment and mitochondria targeting ability

Based on receptor transport and self-chemotaxis, Ang-PAMSe nanomotors have been proved to be able to successfully cross BBB.

Whether it can accurately target GBM and its mitochondria is the focus of this research. Ang-PMSe, PAMSe and Ang-PAMSe (200 μg mL$^{-1}$) were incubated with GBM cells for 24 h, and the uptake of materials by cells was observed qualitatively and quantitatively (Fig. 5h, i). As can be seen from the CLSM images, compared with the Ang-PMSe non-nanomotors with only receptor transport ability and PAMSe nanomotors with only movement ability, the cells incubated with Ang-PAMSe nanomotors with both effects showed more obvious red fluorescence signals, and their fluorescence intensity was nearly twice that of the other two groups (Supplementary Fig. 37). The quantitative detection results of cell uptake rate further confirmed this effect (Fig. 5i). At the same time, the red sample and green cell membrane were co-located. The greater the $R$ value of Pearson's, the higher the degree of overlap between the cell membrane and the sample[44]. Results showed that the $R$ value of Pearson's cells treated with Ang-PAMSe nanomotors was the highest, indicating that their autonomous movement ability can promote the process of cell uptake.

Since the intracellular distribution information is the key factor for the transport and effectiveness of nanomotors, we then tracked the distribution of nanomotors in important organelles in cells after they entered the cells. Because lysosomes can phagocytize nanoparticles that enter cells. In order to play an effective role, escaping lysosomes is an important ability of therapeutic nanoagents. As shown in Fig. 5j, after the Ang-PAMSe nanomotors and drug-loaded Ang-PAMSe/TLND nanomotors were incubated with cells for 5 min. The red fluorescence signal peaks representing nanomotors appeared outside the green area, indicating that these nanomotors were not in lysosome. We also studied the co-localization of nanomotors and Golgi, the main organelles involved in cell transport[45,46]. As shown in Fig. 5k, after the two types of nanomotors were incubated with cells for 2 h, the red fluorescence signal peak representing the nanomotors did not completely coincide with the green region representing the Golgi matrix. As the main way of TLND regulating cell metabolism is to inhibit the production of cellular energy ATP, and the main place of this effect is mitochondria[47], the co-localization of nanomotors and mitochondria was also evaluated. As shown in Fig. 5l, the Ang-PAMSe/TLND nanomotors loaded with TLND had a higher coincidence degree with the green region representing mitochondria, which indicated that it had a better mitochondrial targeting effect. It was conducive to their own and loaded metabolic regulatory drugs to better induce the apoptosis of tumor cells. The above results suggested that Ang-PAMSe/TLND nanomotors can escape lysosomal phagocytosis in a short time after being swallowed by tumor cells, and then target mitochondria and distribute them to Golgi apparatus (conducive to intercellular transport), thus realizing the whole multi-step targeting process of nanomotors.

## Evaluation of NO-induced ICD and in vitro treatment effect

It has been reported in the literature that reactive nitrogen species (RNS) have potent antitumor immune and immunomodulatory functions in TME[48], and high concentration of NO has been proved to act as an ICD inducer and enhance the production of tumor immune antigens[49]. Therefore, we explored the ability of NO to induce ICD in tumor cells and thus activate dendritic cells (DCs) to generate antitumor immune responses by characterizing cell surface exposure to calreticulin (CRT)[50] and extracellular release of high-mobility group box 1 (HMGB1)[51]. From Supplementary Fig. 38, it can be observed that both PAMSe and PAMSe/TLND with NO release function showed significant CRT exposure, where the fluorescence intensity in PAMSe group was about 15.7-fold higher than that in control group. Similarly, PAMSe/TLND significantly increased HMGB1 release by approximately 5-fold compared to the control group (Supplementary Fig. 39), and the above results confirm the ability of NO produced by PAMSe/TLND to induce ICD in tumor cells. We subsequently demonstrated that the raw material of the nanomotor has good biocompatibility (Supplementary

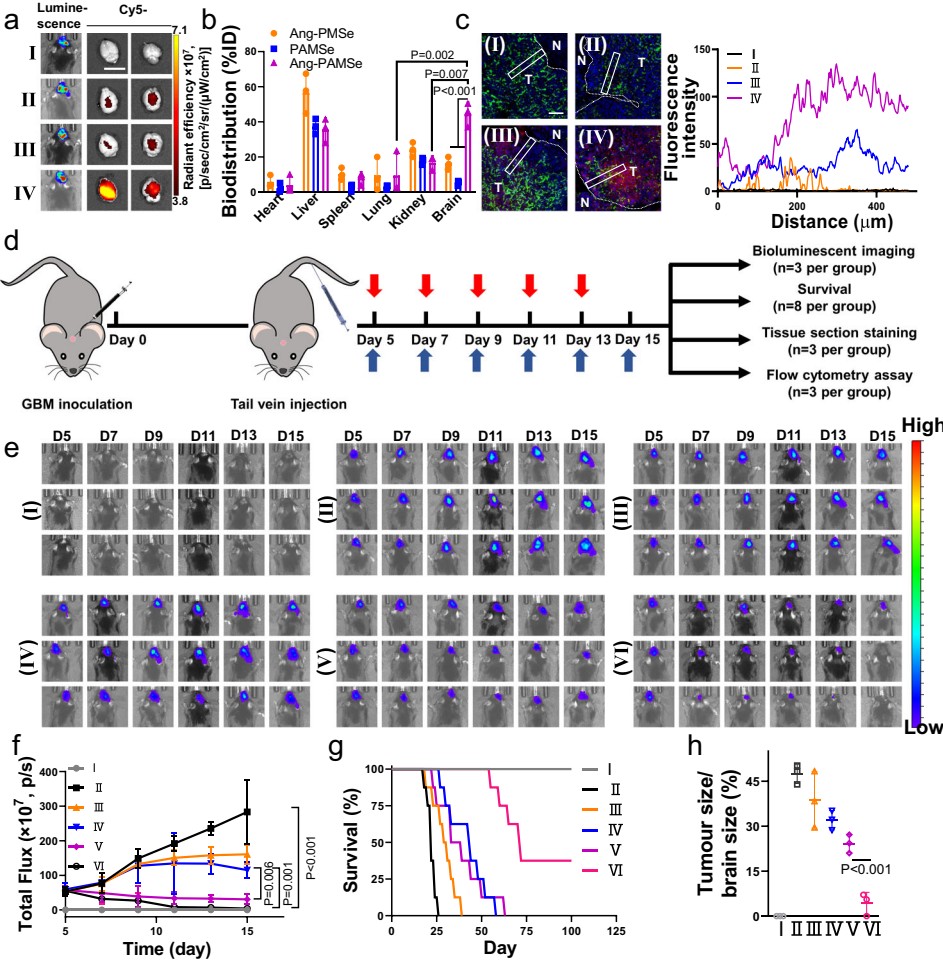

**Fig. 6 | Assessment of targeting and therapeutic efficacy of Ang-PAMSe/TLND nanomotors in vivo. a** Typical chemiluminescence imaging of GBM in C57BL/6 mice before administration and fluorescence images of the brains after injection of different cy5-labeled samples via tail vein injection ($n = 4$ mice per group, Scale bar: 2 cm). **b** Quantitative analysis of sample accumulation in main organs. Cy5-sample levels were determined by fluorescence spectroscopy and expressed as injected dose per gram of tissue (%ID) ($n = 3$ mice per group). **c** Representative CLSM images of brain tumor tissue in mice 24 h after injection of different samples via tail vein and red fluorescence intensity profiles as a function of the distance from a blood vessel in the representative region marked by the white rectangular frames. (Green, blood vessels stained with CD31; Red, samples labeled with red fluorescence by cy5; Blue, cell nuclei stained with DAPI; N: normal brain tissue; T: tumor; $n = 3$ mice,

Scale bar: 100 μm); (The names corresponding to the samples in a and b are (I) PBS, (II) Ang-PMSe, (III) PAMSe, (IV) Ang-PAMSe). **d** Treatment protocols for orthotopic brain-GBM-tumor-bearing models. Where the red arrow is the time point of the drug administration and the blue arrow represents the time point of bioluminescence imaging. **e** Representative IVIS spectrum images and **f** quantified signal intensity ($n = 3$ mice per group). **g** Survival analysis of the mice that loaded with GL261-Luc glioblastoma during treatment process ($n = 8$ mice per group). **h** Quantitative analysis of the tumor size (Start treatment on Day 10) ($n = 3$ mice per group). Data in **b**, **h** are mean ± s.d. Data in **f** are presented as mean ± s.e.m. Statistical significance was assessed by one-way ANOVA with post hoc LSD tests. Source data and exact *p* values are provided in the Source data file.

Fig. 40a), and the PAMSe nanomotor has no significant effect on cell viability in the environment of HUVECs with low ROS/iNOS concentration (Supplementary Fig. 40b), while in cancer cells that can continuously produce ROS/iNOS cell environment, the cellular activity of the material decreased to 49.7% after incubation with cancer cells for 48 h (Supplementary Fig. 40c). Finally, we also evaluated the cellular viability of GBM cells treated with different samples for 24 h. The cellular activity was reduced to 9.5% at a concentration of 200 μg mL$^{-1}$, which may be attributed to the specific cascading reaction (Supplementary Fig. 41).

## Assessment of targeting and therapeutic efficacy of Ang-PAMSe/TLND nanomotors in vivo

Before evaluating the therapeutic effects of nanomotors in vivo, we quantified the TLND level in circulating blood of healthy SD rats at different intervals after intravenous injection of Ang-PAMSe/TLND (3 mg mL$^{-1}$). TLND in Ang-PAMSe/TLND nanomotor exhibited prolonged blood circulation (Supplementary Fig. 42).

Then, we established mice GBM model by orthotopic implantation of GL261-Luc cells (the modeling method of mouse GL261 GBM model was reproducible and able to simulate human GBM)[52]. The tumor-targeting ability of Ang-PAMSe nanomotors, which have both the ability to target BBB transport and the ability to move autonomously, was investigated by tail vein injection of therapeutic agent samples labeled with cy5 (Fig. 6a and Supplementary Fig. 43). Near-infrared fluorescence imaging (IVIS Lumia III) images of mice showed that after 24 h of Ang-PAMSe nanomotor injection, significant red fluorescence was observed in the mouse brain with approximately 1.7 times the intensity of fluorescence after treatment with Ang-PMSe (BBB transport properties only) and PAMSe (motion properties only). Nanomotor biodistribution and organ accumulation were quantitative assessed by intravenously injecting samples via tail vein in orthotopic GL261-bearing mice. Fluorescence quantitation in tumor and other organs showed that the tumor tissue accumulation of Ang-PAMSe group was 3.3 time that of Ang-PMSe group (Fig. 6b), which

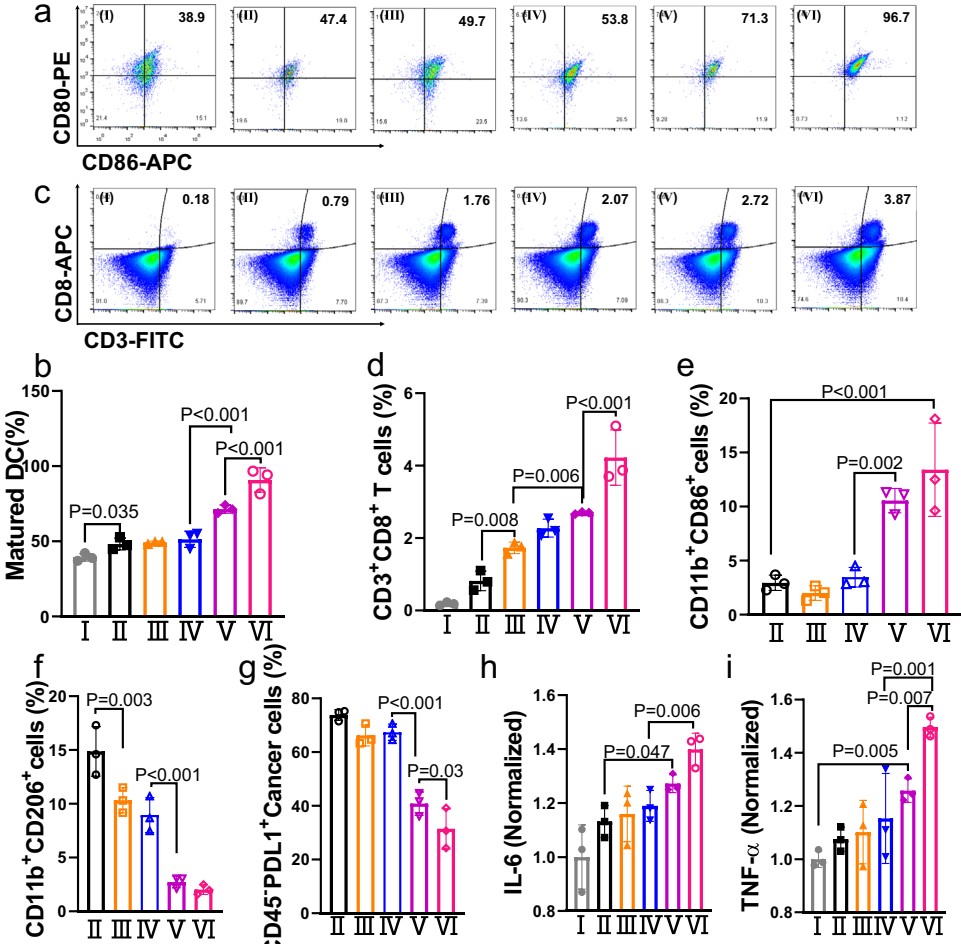

**Fig. 7 | Immune activation of Ang-PAMSe/TLND nanomotors in vivo. a** Flow cytometry analysis and **b** quantitative analysis of DCs maturation level in vivo, the expression of CD80 and CD86 on the surface of DCs extracted from the tumor draining lymph nodes of mice after various treatments. For the gating strategy for DCs maturation analysis refer to Supplementary Fig. 49. **c** Flow cytometry analysis and **d** quantitative analysis of CD8+ T cells in GBM-bearing brain tissue gating on CD3+ cells in each group. For the gating strategy for T cells analysis refer to Supplementary Fig. 52. Quantitative analysis of of **e** M1-like macrophages (CD11b+CD86+) and **f** M2-like macrophages (CD11b+CD206+) in glioma-bearing brain tissue

($n = 3$ mice per group). For the gating strategy analysis refer to Supplementary Fig. 54. **g** Quantitative analysis of CD45-PDL1+ glioma cells in glioma-bearing brain tissue. For the gating strategy analysis, refer to Supplementary Fig. 56. **h** Cytokine levels including IL-6 and **i** TNF-α in the serum of mice at the end of treatment. (The names corresponding to the group are (I) Sham, (II) PBS, (III) TLND, (IV) Ang-PMSe/TLND, (V)) Ang-PAMSe, and (VI) Ang-PAMSe/TLND). Data in **a–i** are representative of three mice per group. Data in **b–i** are mean ± s.d. Statistical significance was assessed by one-way ANOVA with post hoc LSD tests. Source data and exact *p* values are provided in the Source data file.

confirmed that Ang-PAMSe nanomotors showed better BBB penetration and GBM tissue accumulation compared with the three control treatments.

Subsequently, we evaluated the penetration of Ang-PAMSe in solid tumors (Fig. 6c). After 24 h of intravenous injection, the red fluorescence signal of Ang-PAMSe nanomotors almost covered the whole tumor (fluorescence intensity of about 90.2 a.u. was still detectable at about 500 μm from the tumor edge). This can be attributed to the good tumor chemotaxis of Ang-PAMSe nanomotors, which exhibit better tissue penetration ability. In contrast, the fluorescence signal of Ang-PMSe was very weak at about 200 μm from the tumor edge due to the lack of autonomous motility, and the PAMSe nanomotors with only motility could penetrate to about 350 μm from the tumor edge with a fluorescence intensity of 62.8 a.u. These results indicated that the Ang-derived targeting function and the autonomous motility of nanomotors played a key synergistic role in realizing Ang-PAMSe nanomotor to effectively penetrate BBB and GBM.

We also evaluated the therapeutic efficacy of drug-loaded Ang-PAMSe/TLND nanomotors on GBM mice model. The mice successfully inoculated with GBM were randomly divided into groups. Each

treatment agent was injected intravenously on days 5, 7, 9, 11, and 13, respectively, and the bioluminescence imaging of mouse GBM was performed (Fig. 6d). The results showed that the total fluorescence flux of GBM in mice treated with PBS increased to 5 times of the initial value at the end of treatment, and the GBM of mice treated with TLND and Ang-PMSe/TLND nanomotors increased to 3.0 and 2.0 times of the initial value at the end of treatment, respectively. It should be noted that the total GBM fluorescence flux of mice treated with Ang-PAMSe and Ang-PAMSe/TLND nanomotors gradually decreased, and the GBM fluorescence signal of the Ang-PAMSe group decreased to half of the initial value. After treatment with Ang-PAMSe/TLND nanomotors, the fluorescence signal decreased more significantly, reducing to 0.07 times of the initial value (Fig. 6e, f). Meantime, there was no significant change in the body weight of mice (Supplementary Fig. 44). Hematoxylin−eosin (H&E) staining results of main organs showed that no obvious damage systemic toxicity or abnormality after Ang-PAMSe/TLND nanomotors treatment (Supplementary Fig. 45). Moreover, to assess the neurotoxicity of nanomotors more comprehensively, we administered the tail vein to healthy C57BL/6J mice (without tumors) once every 2 days for a total of 5 doses. At the end of administration,

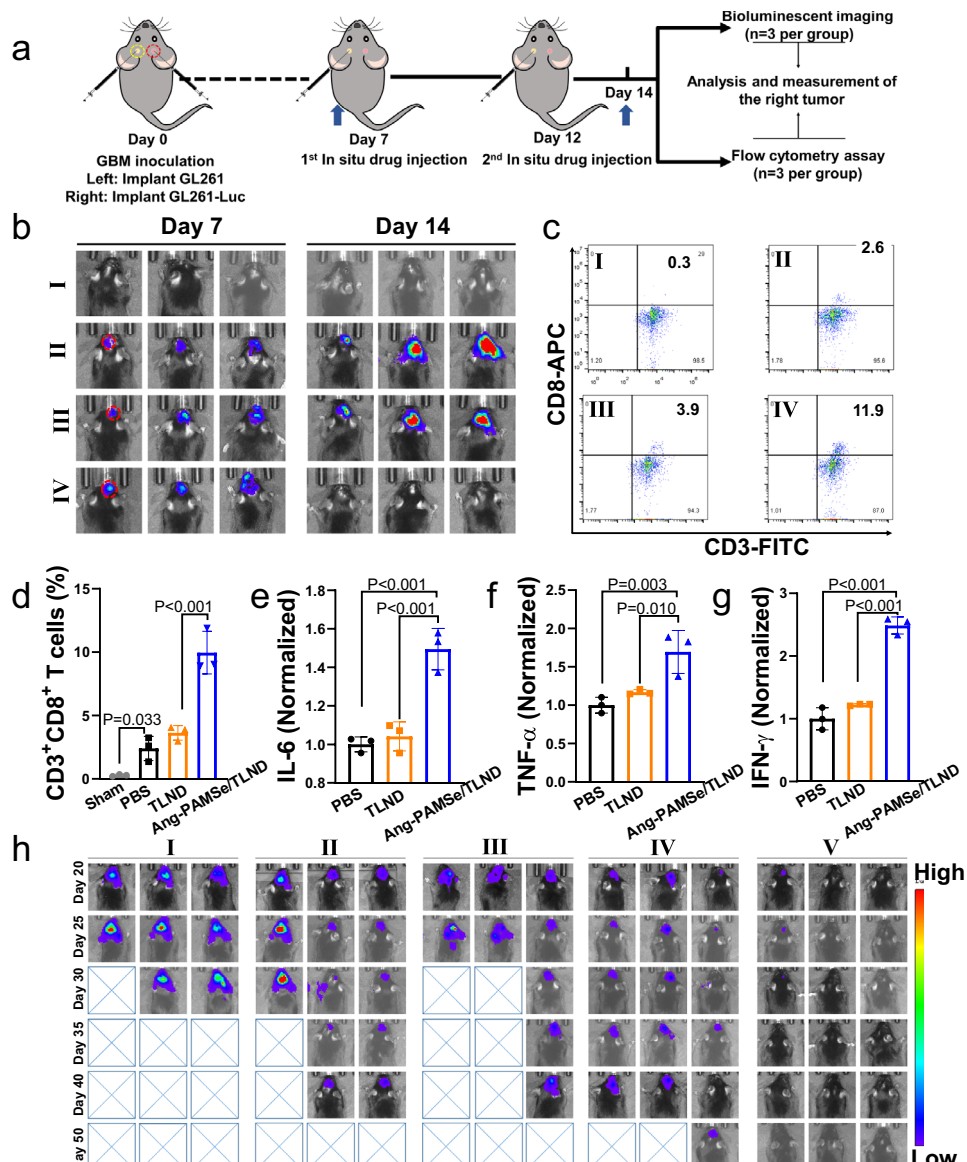

**Fig. 8 | Localized metabolic immunotherapy to achieve distal effect for the GBM model and prevention of GBM recurrence. a** The schematic illustration of the experimental design to determine the resistance to metastasis of the therapeutic agent. The blue arrow represents the time point of imaging. **b** The bioluminescence imaging of the sham-operated group and GBM mice (red circles indicate chemiluminescence imaging) and the bioluminescence imaging was performed on the 7th and 14th day after the treatment process. **c** Representative flow cytometric plots showing different groups of T cells in abscopal GBM (the right tumor). For the gating strategy for T cells analysis, refer to Supplementary Fig. 58. The names corresponding to the samples in **b, c** are (I) Sham, (II) PBS, (III) TLND, and (IV) Ang-PAMSe/TLND. **d** Quantitative analysis of CD3$^+$CD8$^+$ cytotoxic T cells. **e** Cytokine levels including IL-6 and **f** TNF-α in the serum of mice at the end of treatment. **g** Determination of IFN-γ content in mouse serum. **h** Monitoring of GBM recurrence after various treatments for primary orthotopic tumors. (The names corresponding to the samples are (I) PBS, (II) TLND, (III) Ang-PMSe/TLND, (IV) Ang-PAMSe, and (V) Ang-PAMSe/TLND). Data in **b**–**h** are representative of three mice per group. Data in **d**–**g** are mean ± s.d. Statistical significance was assessed by one-way ANOVA with post hoc LSD tests. Source data and exact *p* values are provided in the Source data file.

the cortical tissues were collected separately and stained with H&E to observe the morphological changes of the cerebral cortex to assess the neurotoxicity of the material[53]. As shown in Supplementary Fig. 46, no significant abnormalities of cell shape, nuclei shrinkage or fracture were observed in the cerebral cortex of mice treated with Ang-PAMSe/TLND. In addition, we continue to monitored the survival of mice. Due to the rapid progression of GBM, all control mice (PBS group) died on day 27 of GBM inoculation. Similarly, both TLND and Ang-PMSe/TLND nanomotors prolonged the survival of mice to some extent with all dying at day 39 and day 58, respectively. The median survival of mice treated with Ang-PAMSe/TLND was significantly prolonged to 71 days, significantly longer than the median survival of mice treated with Ang-

PAMSe (33 days) or TLND (29 days), respectively (Fig. 6g). All these results indicated that Ang-PAMSe/TLND nanomotor has the best efficacy in inhibiting the growth of GBM and significantly prolonging the survival of mice.

Further, we established an experiment of administering the treatment after 10 days of GL261 cell inoculation. The procedure of GL261 cells inoculation remained unchanged, and the drug was administered only on day 11, and 5 times every 2 days. On day 20, the whole brains of mice were taken for histological section and H&E staining to observe and compare the size of brain tumor (Supplementary Fig. 47). As shown in Fig. 6h and Supplementary Fig. 48, after treatment by different treatment groups, the tumors were still

suppressed to different degrees. Among them, after treatment by Ang-PAMSe/TLND, the GBM of mice had the most obvious suppression effect.

Then, we further investigated the therapeutic mechanism of Ang-PAMSe/TLND nanomotors in vivo. It has previously been demonstrated at the cellular level that NO (or OONO⁻) produced by Ang-PAMSe/TLND nanomotors can promote cellular CRT expression and HMGB1 release, so we speculated that NO may also be involved in the regulation of ICD in vivo, while dead cell residues can act as tumor-associated antigens (TAAs) to activate immune[54]. DCs is a major antigen-presenting cell, which can collect TAAs and process it to nearby draining lymph nodes, and stimulate activated T cells[55]. During migration, TAAs are modified by DCs to further recognize T cell receptors. During this period, the maturation status of DCs always determines the ultimate success or failure of T cell activation, and the upregulation of the expression of co-stimulatory molecules CD80 and CD86 on the DCs surface is a measure of DCs maturation[56–58]. Tumor-draining lymph nodes and tumor tissue were harvested on day 15 for DCs maturation analysis. As shown in Fig. 7a, b and Supplementary Fig. 49, both the metabolic modulating drug TLND (49.2%) and Ang-PMSe/TLND (51.3%) without NO production capacity showed no significant increase in DCs maturation level compared to the PBS group (48.2%). In contrast, the DCs maturation status of the Ang-PAMSe treatment group was 71.4%, and the performance of Ang-PAMSe/TLND in promoting DCs maturation was even more prominent, up to 90.8%, which was almost twice as high as that of the PBS group. Similarly, the largest proportion of mature DCs detected in GBM tumor tissue also appeared in the Ang-PAMSe/TLND nanomotor group. Compared with PBS treatment group (12.6%), after treated with Ang-PAMSe/TLND with cascade effect, the percentage of mature DCs in tumor tissues increased to 34.1% (Supplementary Figs. 50 and 51).

In addition, in order to explore the potential mechanism of immune activation triggered by Ang-PAMSe/TLND nanomotors, we evaluated the infiltration of immune cells (cytotoxic T cells (CD3⁺CD8⁺) in tumor tissues. Among them, cytotoxic T cells represent cells with direct tumor-killing effect[59]. More CD3⁺CD8⁺ cytotoxic T cells (about 5.2 times more than in the PBS group, Fig. 7c, d and Supplementary Fig. 52) was found in the brain tissue of mice treated with Ang-PAMSe/TLND. The immunosuppressive microenvironment in tumors, such as the polarization of macrophages from M1 phenotype to M2 phenotype, will affect the immune response of T cell, thus promoting the immune escape of tumor cells and tumor progression[60]. The production of NO can reverse the immunosuppressive microenvironment of tumors. For example, it can regulate the polarization of macrophage to M1 phenotype, inhibit the expression of PDL1 in cancer cells[61]. Therefore, we examined these two indicators separately. After being treated with different samples (PBS, TLND, Ang-PMSe/TLND, Ang-PAMSe and Ang-PAMSe/TLND), there was no difference in the proportion of M1-type macrophages (CD11b⁺CD86⁺) in GBM tissues of mice treated with TLND and Ang-PMSe/TLND groups compared with the PBS group (Fig. 7e, f and Supplementary Figs. 53 and 54), while Ang-PAMSe and Ang-PAMSe/TLND treatment groups increased to 10.6% and 13.4%, which were approximately 3.6 and 4.5 times higher than those in the PBS group. We also examined the expression levels of PDL1 of cancer cells in treated GBM tissues. As shown in Fig. 7g and Supplementary Figs. 55 and 56, compared to the PBS group (73.8%), the PDL1 expression level in Ang-PAMSe (40.8%) and Ang-PAMSe/TLND (31.4%) groups decreased a lot, about 0.6 and 0.4 times lower than that of the PBS group. The secretion of immune-related cytokines interleukin-6 (IL-6) and tumor necrosis factor-α (TNF-α) in the serum of mice was also detected[62]. As shown in Fig. 7h, i, the serum levels of IL-6 and TNF-α cytokines were higher in mice treated with Ang-PAMSe/TLND than in other

treatment groups. All these results demonstrated that Ang-PAMSe/TLND nanomotor could achieve enhanced immunotherapy of GBM.

The above results confirmed that NO can fully participate in several steps of the anti-tumor immune cycle in vivo through the motion effect of nanomotors, and achieve efficient immune enhancing effect in cooperation with the metabolic drug TLND. Then, we evaluated whether this therapeutic option that activate the body's own systemic anti-tumor immune response can inhibit tumor metastasis and recurrence. We designed a primary-distal GBM model to investigate whether the proposed PAMSe/TLND nanomotors can achieve systemic anti-tumor response after local GBM treatment, and thus inhibiting tumor metastasis[63]. As shown in Fig. 8a, to better monitor tumor progression, we implanted GL261 cells without luciferase gene transfection markers in the left side of the mouse brain as the "primary tumor" and GL261-Luc GBM in the right side of the mouse brain as the "distal tumor." It is important to note that all treatment procedures give only local treatment of the drug to the primary tumor, while as a distal tumor no drug is given directly. The GBM growth was monitored on both sides using NMR imaging and chemiluminescence imaging, respectively, on day 7 after implantation of the GBM. Subsequently, after 2 topical administrations to the primary tumor (day 7 and day 12), chemiluminescence imaging was performed on day 14. As shown in Supplementary Fig. 57, T₂-weighted NMR images showed no significant black shadows in the brains of mice in the sham-operated group, whereas for mice with loaded GBM there was a high density of black shaded areas (yellow circles) in their left brain. Similarly, the bioluminescence images also showed a brighter GBM fluorescence signal in the right brain region (red circles), and the above results were able to demonstrate the successful construction of the primary and distal tumors (Fig. 8b). As shown in Fig. 8a, b, after the local administrations of the primary tumor twice, it can be clearly observed that the distal tumors of the control (PBS group) mice progressed rapidly, and the distal tumors of the mice in the TLND group also showed a trend of proliferation and growth, while the distal tumors of the mice treated with Ang-PAMSe/TLND nanomotors almost completely disappeared, implying that Ang-PAMSe/TLND nanomotors had a good effect of inhibiting and clearing the distal tumors.

To further investigate the mechanism of distal effects induced by local immunometabolic therapy, we examined the infiltration of T cells in distal tumors[64]. After treatment with different samples, T cells were collected from distal tumor tissues, and cytotoxic T cells (Fig. 8c, d and Supplementary Fig. 58), indicating that Ang-PAMSe/TLND nanomotors was able to enhance anti-tumor immune performance. In addition, the serum levels of cytokines such as IL-6, TNF-α and interferon-γ (IFN-γ) were measured by enzyme-linked immunosorbent assay (ELISA) in mice (Fig. 8e–g), and the serum levels of the above cytokines in mice treated with Ang-PAMSe/TLND nanomotor were higher than those in other treatment groups. All these results demonstrate that Ang-PAMSe/TLND nanomotors can trigger a systemic anti-tumor response in mouse through local treatment of GBM. Finally, we evaluated the metabolic immunomodulatory effect of Ang-PAMSe/TLND nanomotors on recurrence prevention of orthotopic GBM mice. Mice loaded with GBM were treated with different samples for 15 days, then the administration was stopped and tumor recurrence of mice were monitored every 5 days. As shown in Fig. 8h, none of the mice treated with Ang-PAMSe/TLND nanomotors showed any significant signs of recurrence at day 50.

## Discussion

Here, we show a kind of chemotactic nanomotor PAMSe. Compared with the introduction of active sites through the post-loading method (PMSe/A nanomotor), it can introduce active sites through in situ covalent binding method, so that it has more NO release amount (4.7 μM vs. 1.6 μM), longer lasting release time (96 h vs. 24 h), more

significant motility behavior (average velocity $5.2 \pm 1.0 \, \mu m \, s^{-1}$ v.s. $2.1 \pm 0.6 \, \mu m \, s^{-1}$) and easier cellular uptake (about 4.1 times more than the PMSe/A nanomotor). The good chemotactic ability of this kind of nanomotors on tumor cells with high expression of ROS/iNOS are confirmed by both static Y-channel model (chemotactic velocity of $6.7 \, \mu m \, s^{-1}$ and a maximum chemotactic index of 0.6 at a cell density of $5 \times 10^{6}$ cells $mL^{-1}$) and dynamic microfluidic model. The cascading effect constructed within this kind of chemotactic nanomotor and the drug-loaded TLND therein could generate more NO, facilitating its better role in destroying the tumor metabolic symbiosis process through the motion effect, and thus effectively reducing the energy supply of tumor cells (ATP content decreased from 11.8 to $1.5 \, \mu M$ in 24 h).

In this work, we also propose a step-by-step targeting strategy based on the synergistic effect of chemical recognition and TME response, which are validated by in vivo and in vitro models. The importance of synergistic effect for brain endothelial cell recognition, transport, and uptake by tumor cells are confirmed by in vitro monolayer BBB model (brain endothelial cell transport efficiency was about 68.1%), a transwell model with multilayer of cells (the BBB transport rate can reach to 61.1%), and in vitro dynamic 3D BBB model (5.7 times that of cy5-PMSe passive nanoparticles). After being uptaken by tumor cells, nanomotors can reach target mitochondria due to the modification of TPP. This synergy is also demonstrated in mice model, where Ang-PAMSe nanomotors can almost spread throughout the tumor. Furthermore, nanomotors reaching the tumor tissue can interfere with the tumor immune cycle process in multi-step by releasing NO and TLND. It can effectively trigger the ICD of tumor cells, promote the maturation of DCs (mature DCs in tumor tissues from 12.6% in the PBS group to 34.1% in the Ang-PAMSe group), promote macrophages polarization from M2 phenotype to M1 phenotype (the proportion of M1 macrophages is 4.5 times that of PBS group), reduce PDL1 expression on cancer cells (from 73.8% in the PBS group to 31.4% in the Ang-PAMSe group), promote the infiltration of cytotoxic T lymphocytes (about 5.2 times more than in the PBS group), reduce the proportion of Treg cells (from 9.6% of PBS group to 3.5% of Ang-PAMSe/TLND group), activate the body's own systemic anti-tumor immune response (after local administrations of the primary tumor, the distal tumors of the PBS group mice progressed rapidly, while the Ang-PAMSe/TLND group almost completely disappeared), and inhibit tumor metastasis and recurrence (no metastasize or recurrence is found within 50 days, and the median survival is extended to 71 days).

## Methods

### Synthesis of arginine-methacrylamide (Arg-Me)
L-arginine (2 g, 11.5 mmol, Yuanye Biotechnology Co., Ltd., China) was fully dissolved in a mixture of deionized water (20 mL) and 1,4-dioxane (8.5 mL, Sinopharm Chemical Reagent Co., Ltd.), and triethylamine (4.5 mL, 32.3 mmol, Aladdin Chemistry Co., Ltd.) was added while stirring and the solution system was cooled with an ice/water bath. After the solution system was cooled to 0 °C, methacrylic anhydride (3 mL, 18.9 mmol, Aladdin Chemistry Co., Ltd.) was added to the system drop by drop in 10 min. The ice/water bath was removed and the mixture was stirred overnight at room temperature. After the reaction, the reaction system was added drop by drop to an excess of acetone (400 mL, Sinopharm Chemical Reagent Co., Ltd.) solution to obtain the precipitate using the acetone precipitation method. The precipitate was then redissolved in water and precipitated again in an excess of acetone, and this step was repeated twice. Finally, the precipitate was dried under vacuum at room temperature to obtain a white powder as Arg-Me.

### Synthesis of diselenide crosslinker
Selenocysteamine hydrochloride (1.0 g, 3.1 mmol, Shanghai Bide Pharmaceutical Technology Co.) was added to 60 mL of anhydrous

dichloromethane (J&K Scientific Ltd.) solution and triethylamine (2.5 g, 25.1 mmol) was added under magnetic stirring. After cooling the mixture to 0 °C, methacryloyl chloride (1.3 g, 104.5 mmol, Sinopharm Chemical Reagent Co., Ltd.) was slowly added to the mixture. The reaction mixture was then stirred for 24 h at room temperature under the $N_2$ atmosphere. After the reaction, the reaction system was extracted with deionised water to remove impurities, dried overnight over anhydrous sodium sulfate ($Na_2SO_4$, Sinopharm Chemical Reagent Co., Ltd.) and the organic phase was removed by distillation under reduced pressure. The resulting product was purified from the raw material by a silica gel column using ethyl acetate (Jiangsu Yonghua Fine Chemical Co., Ltd.)/petroleum ether (Jiangsu Yonghua Fine Chemical Co., Ltd.) (2:1 by volume).

### Synthesis of PAMSe nanomotors
Arg-Me (121 mg, 0.5 mmol), diselenide crosslinker (20 mg, 0.05 mmol) and azo diisobutyronitrile (AIBN) were dissolved in anhydrous dimethyl sulfoxide (DMSO, 20 mL, Sinopharm Chemical Reagent Co., Ltd.) in a $N_2$ atmosphere, heated at reflux, and stirred continuously for 3 h. After the reaction, the precipitate obtained by centrifugation of the product was washed three times with water and then freeze-dried to obtain a white powder, which was PAMSe. To prepare arginine-loaded nanomotors as a control, the above Arg-Me monomer was replaced by methacrylic acid (Sinopharm Chemical Reagent Co., Ltd.) and the rest of the synthesis procedure was kept unchanged to obtain polymer nanoparticles as PMSe. PMSe was then mixed with L-arginine (mass ratio = 10:1) for post-loading of L-arginine to obtain the post-loading type of PMSe/A nanomotor.

### Synthesis of Ang-PAMSe nanomotors
In all, 100 mg of PAMSe and 4 mL of anhydrous DMSO were added in a 25 mL three-necked flask and the solid were dissolved by sonication, then 54 μL of sulfoxide dichloride ($SOCl_2$, Sinopharm Chemical Reagent Co., Ltd.) were added drop by drop and stirred at room temperature for 2 h. The system was transferred to an oil bath at 90 °C and stirred while continuously pumping with a vacuum pump for 30 min to remove the unreacted $SOCl_2$. Next, 5 mg of N-(4-aminophenyl) maleimide (APM) was added to the flask and stirred overnight. The reaction solution was collected and transferred to a dialysis bag (MW = 3500 Da) for 3 days in an aqueous environment and then freeze-dried to obtain PAMSe-APM. PAMSe-APM mixed with purchased sulfhydryl groups modified Angiopep-2 (Ang-SH, Hefei Guo peptide Biotechnology Co., Ltd.) was added to phosphate buffer solution (PBS) at a ratio of 1:9 by mass and stirred overnight at room temperature. The resulting reaction system was transferred to a dialysis bag (MW = 3500 Da) and dialyzed in 5% glucose aqueous solution to remove the free peptides. After dialysis, the product was freeze-dried, which was Ang-PAMSe. The content of Ang in Ang-PAMSe was quantitatively detected by UV-vis spectrophotometer.

### Synthesis of Cy5-fluorescently labeled nanomotors
The nanomotors to be grafted for fluorescence, such as PAMSe, PMSe/A, and Ang-PAMSe, were placed in 1 mL of aqueous solution (200 μg $mL^{-1}$) of Tris(2-carboxyethyl) phosphine (TCEP, Aladdin Chemistry Co., Ltd.) and stirred for 12 h. Next, 50 μL of Cy5-maleimide (Apexbio ApexBio Technology LLC, Lot NO. AB941337769) was added to the system and stirred for 24 h in $N_2$ atmosphere. At the end of the reaction, the precipitate was obtained by centrifugation ($5800 \times g$, 8 min) and washed three times with water, and the final product was freeze-dried to obtain a blue powder.

### Synthesis of triphenylphosphine-modified Lonidamine (TLND)
$BrCH_2CH_2NH_3^+Br^-$ (2.0 g, 10 mmol, Aladdin Chemistry Co., Ltd.) and TPP (2.6 g, 10 mmol, Energy Chemical Co., Ltd. China) were refluxed in acetonitrile (40 mL, Sinopharm Chemical Reagent Co., Ltd.) with

stirring for 24 h. The reaction mixture was allowed to evaporate by rotation and the remaining crystals were dissolved in a small amount of water and the aqueous solution was adjusted to pH 11.0 with NaOH (2 mol $L^{-1}$, Sinopharm Chemical Reagent Co., Ltd.). After that, the water was evaporated by rotation and extracted with methanol to get TPP-$NH_2$. Lonidamine (LND, 0.4 g, 1.1 mmol, Aladdin Chemistry Co., Ltd.), oxalyl chloride (2 mL, Sinopharm Chemical Reagent Co., Ltd.) and N, N-dimethylformamide (DMF, 0.1 mL, Sinopharm Chemical Reagent Co., Ltd.) were added to dichloromethane (20 mL) and refluxed at 50 °C for 2 h. The yellow solid was obtained by spin evaporation under reduced pressure. Then TPP-$NH_2$ (0.5 g, 1.1 mmol), triethylamine (160 μL, 1.1 mmol) and the above yellow solid were dissolved in dichloromethane (20 mL), stirred at room temperature for 12 h and extracted by ultrapure water. The organic phase was dried with anhydrous $Na_2SO_4$ (Sinopharm Chemical Reagent Co., Ltd.) and left to stand overnight. The liquid was finally collected and the solvent was removed by rotary evaporation and dried under vacuum to give a yellow-brown solid as TLND.

## Characterization of materials
The surface morphology, dimensions and elemental mapping analysis of NPs were studied with a JEM-2100 transmission electron microscope and a 200 kV field emission transmission electron microscope (JEOL JEM-2100F). Zeta potential and particle size were measured by the Zetasizer (Nano-Z, Malvern, UK). The NanoSight NS 300 (Malvern Instruments) high-resolution, real-time dynamic nanoparticle detection technique was used for particle number measurement. $^1$H-NMR spectra of samples were recorded on a Bruker Avance 400 spectrometer. Liquid chromatography-mass spectrometry (6460 QQQ MS) coupling test using electrospray ionization source (ESI) mass spectrometer with liquid chromatography (Agilent 1290 Infinity LC). Fourier transform infrared (FTIR) spectra were captured by the Cary 5000 FTIR spectrophotometer (Varian, USA). Ultraviolet (UV) absorption spectra was obtained by using a PerkinElmer 650 spectrophotometer (PerkinElmer Ltd., US). All fluorescence spectra of samples measurements were accomplished on a F-4600 spectrophotometer (Hitachi, Japan). The molecular weight of the polymer after degradation was detected by gel permeation chromatography (GPC, Agilent 1260 InfinityII). The absorbance of the plates was read at corresponding wavelength using a microplate reader (Multiskan FC, Thermo Fisher instruments Co., Ltd., US). The chemotactic movement of nanomotors in the Y model and the tracking of movement in the cellular environment are captured by fluorescence microscopy (MF53-N, Guangzhou Micro-shot Technology Co., Ltd., Guangzhou, China). The microfluidic device in the chemotaxis performance assessment was fixed above the lens of the confocal laser scanning microscopy (CLSM, HP Apo TIRF 100X N.A. 1.49, Nikon, Ti-E-A1R, Japan) for continuous video recording. All the fluorescence images of the cells were taken by CLSM. Both in vivo fluorescence imaging of the material and chemiluminescence imaging of mouse brain tumors were imaged using a small animal imager (IVIS Lumina III, Living image system 4.5.5). The cell flow data were all collected by flow cytometry (BD Accuri C6 Plus flow cytometer). The chemical structure formula is drawn with ChemDraw (Ultra 7.0) software.

## TLND drug loading and drug release performance of nanomotors
In all, 50 mg of each of the prepared PAMSe, PMSe and Ang-PAMSe and 10 mg of TLND were weighed and mixed in 5 mL solution ($V_{methanol:water} = 2:3$), then mixed on a silent mixer for 24 h, washed several times by centrifugation and dried in a freeze dryer. Five mg of each TLND-loaded material was dispersed in PBS solution or PBS (5 mL) containing 0.002% $H_2O_2$, placed on a silent mixer and the supernatant was centrifuged at a specific time point to determine the cumulative release concentration of the drug using UV at 300 nm

loading (DL) and encapsulation efficiency (EE) of the nanomotor are calculated by formula 1 and formula 2.

$$\text{LC}\,(\%) = [m\,(\text{loaded drug})/m\,(\text{nanomotors})] \times 100\% \quad (1)$$

$$\text{EE}\,(\%) = [m\,(\text{loaded drug})/m\,(\text{initial total drug})] \times 100\% \quad (2)$$

## Cell culture
The mouse GL261 (catalog: SAc0135) and GL261-Luc (catalog: IML-083) glioma cell lines were purchased from Shanghai Fusheng Industrial Co.; Mouse-derived brain endothelial cells (bEnd.3, cell NO. CRL-2299) and human-derived glioma cells (U87, CL-0238) were purchased from Wuhan Procell Life Science&Technology Co., Ltd.; Human umbilical vein endothelial cells (HUVECs) were purchased from American type culture collection (ATCC, U.S., cell NO. CRL-1730). Michigan cancer foundation-7 cells (MCF-7) were obtained from Shanghai Meixuan Biotechnology Co., Ltd (cell NO. 115). Cells were cultured in Roswell Park Memorial Institute-1640 (RPMI-1640) supplemented with 10% fetal bovine serum, 100 U $mL^{-1}$ of penicillin G sodium and 100 μg $mL^{-1}$ of streptomycin sulfate in a humidified atmosphere that contained 5% $CO_2$ at 37 °C. All cell lines were negative by mycoplasma testing.

## Assessment of NO release performance
The cancer cells were inoculated into 96-well plates and incubated overnight, the original medium was discarded and DMEM solution (200 μg $mL^{-1}$) of fresh amphoteric nanomotors (PAMSe) and post-loaded nanomotors (PMSe/A) was added, as a control, the two samples were added to the wells without cancer cells and incubated for 96 h. During this period, the solution in the wells was taken at a fixed time point and the NO content of the solution was measured using the NO assay kit (Beyotime Institute of Biotechnology, China).

## Analysis of the movement behavior of nanomotors in the cellular environment
Cancer cells ($5 \times 10^4$ cells $mL^{-1}$) were inoculated in a culture dish and incubated overnight. Cy5-labeled PAMSe nanomotors (200 μg $mL^{-1}$) and cy5-PMSe/A nanomotors (200 μg $mL^{-1}$) were added to the system, and as a control, the above two types of nanomotors were added separately to a DMEM medium-only environment, and a assembled inverted fluorescence microscope (×100 objective) was used to track the nanomotor motion trajectory. Image sequences were tracked and analyzed manually using the Image J (Version 2.0.0) plugin. The average velocity of the particles was obtained by averaging the velocity of the particles over different time intervals. Each group of 10 nanoparticles was tracked and their velocities were measured to obtain the final average velocity and calculate the MSD value. The MSD was then fitted to determine the type of motion of the different types of particles.

## Collective chemotactic behavior of nanomotors in Y-shaped channels
The dimensional parameters of the Y model channel are: 1 cm long and 0.4 cm wide for the channel body and 1 cm long and 0.3 cm wide for the Y-shaped glass substrate channel of the two branched channels. Preparation of gels to mimic chemokine gradients: 5 mg of agarose was added to 500 μL of PBS and then heated to 90 °C to dissolve. When cooled to room temperature, 200 μL of $5 \times 10^6$ cells $mL^{-1}$ of GL261-Luc cell lysate was added and transferred to the 4 °C gel. As a control, the remaining conditions were unchanged and the same concentration of bEnd.3 cell lysate was added to the gel to simulate an environment without a chemokine gradient. Agarose gels (~8 mm³) containing different cell lysates were placed in storage slots (II) and storage slots (III) on

the branch channels, and then 400 µL of PBS was added to the Y-shaped channels, keeping the water level and covering the entire channel. The mixtures were solidified at room temperature to form agarose gels that containing cell lysates. Over time, ROS/iNOS diffused in the Y-shaped channel to form a chemoattractant concentration gradient. The concentration gradient of ROS (in the case of superoxide anion, $O_2 \cdot^-$) and iNOS at five locations from near to far from the center of the lysate of cancer cells were detected using the superoxide assay kit (Beyotime, S0060) and human iNOS ELISA kit (MEIMIAN, MM-1514H1, 96 T), respectively. GL261 cell lysates with different densities ($10^3$, $10^4$, $10^5$, and $10^6$) were placed in Y channel (III), and the intracellular iNOS and ROS concentrations were quantified at the corresponding cell densities. Meanwhile, the fluorescence changes of nanomotors within the Y-channel (III) is tracked within 90 min. Then, 50 µL of Cy5-PAMSe and Cy5-PMSe nanoparticle solutions were lightly dropped into the reservoir (I) respectively, and an inverted fluorescence microscope was used to continuously track the convergence direction of the particles at the mouth of the trigonal channel, and to take fluorescence images of the reservoir (II) and reservoir (III) at certain time points (0 min, 30 min, 60 min and 90 min). Similarly, Cy5-PAMSe nanomotors were loaded into agarose gels (no chemotaxis gradient) with bEnd.3 cell lysate at both storage tanks (II) and storage tanks (III) to observe the above indicators. Finally, the trajectories of $n = 50$ granules were traced for each sample and (100 s) normalized to a common origin and analyzed for chemotactic velocity and chemotactic index. The fluorescence intensity pseudo-color images were processed using ImageJ (Version 2.0.0) and the gray scale values were corrected to a range of 0–1 using a linear function.

### Dynamic chemotactic behavior of nanomotors in microfluidic channels

A three-inlet and one-outlet glass substrate microfluidic channel with dimensions of 2.2 cm (length) × 1.5 mm (width) × 300 µm (height) was used to evaluate the dynamic tropism of the motors under confocal scanning laser microscopy. Cy5-PAMSe nanomotor or Cy5-PMSe non-nanomotor solution was injected into the middle channel at a flow rate of 0.6 mL h$^{-1}$, cancer cell lysates were injected into the lower channel at the same flow rate, and buffer solution was injected into the upper channel. After the flow rate was stabilized, a CLSM was used to start scanning near the outlet tangent to the lower side channel, and the position was continuously recorded on video for 1 min (30 frames per second). The fluorescence intensity of each material was measured by Image J (Version 2.0.0) perpendicular to the flow direction, and a normalized fluorescence intensity curve was plotted based on the average fluorescence intensity of each frame. As a control, buffer solution was injected at the same flow rate in both the upper and lower channels, and Cy5-PAMSe nanomotor or Cy5-PMSe non-nanomotor solution was injected into the middle channel at a flow rate of 0.6 mL h$^{-1}$ to monitor the corresponding index.

### NO release and ROS depletion from PAMSe/TLND nanomotors

Cancer cells ($8 \times 10^5$ cells mL$^{-1}$) were inoculated in a culture dish and incubated overnight in an apposed culture. Blank medium, PAMSe (200 µg mL$^{-1}$), TLND, PAMSe/TLND (200 µg mL$^{-1}$) were added to each system after incubation with cells for different times (3 h, 6 h, 9 h, 12 h and 24 h) respectively, and 3-Amino,4-aminomethyl-2′,7′-difluorescein, diacetate (DAF-FM DA, NO fluorescent probe, 5 µM, Beyotime Institute of Biotechnology, China) was incubated with the cells for 60 min and images were taken using CLSM (Ex: 495 nm; Em: 515 nm), their fluorescence being quantified using Image J. Similarly, at the end of material treatment of cells, 2′,7′-Dichlorofluorescin diacetate (DCFH-DA, ROS fluorescent probe, 10 µM, Beijing Solarbio Science & Technology Co., Ltd.) was incubated with cells for 60 min and images were taken using CLSM (Ex: 504 nm; Em: 515 nm), their fluorescence being quantified using Image J (Version 2.0.0).

### Intracellular/extracellular lactate content assay

Cancer cells (cell density $5 \times 10^5$ cells mL$^{-1}$) were inoculated in 24-well plates and incubated overnight for 12 h. Blank medium, PAMSe (200 µg mL$^{-1}$), TLND, PAMSe/TLND (200 µL, 200 µg mL$^{-1}$) were then added to the corresponding wells and incubated until 15, 18, 20, 22, 24 and 36 h respectively. The medium was collected at 15, 18, 20, 22, 24, and 36 h and the cells were gently washed with PBS, and all removed fluid was mixed for the next step. 100 µL of trypsin digest was added to each well, incubated for 1 min, then centrifuged at 90 g for 10 min and the bottom cells were collected. All wash buffer was collected for the next step of analysis. The lower cells were digested by 100 µL trypsin for 1 min and 100 µL culture solution was used to terminate the digestion. The cells were then centrifuged, the supernatant removed and the cell sediment left. The cell sediment was washed with 200 µL PBS, centrifuged, the supernatant removed and the cell sediment left. Subsequently, 250 µL of cell lysate was added to fully lyse the obtained cells for 15 min, centrifuged at $810 \times g$ for 10 min, and the supernatant was taken and the intracellular lactate content of the different groups was determined using a lactate kit (Nanjing Jiancheng Institute of Biological Engineering), while the protein content of the cells was determined in each group using a BSA kit (Nanjing Jiancheng Institute of Biological Engineering), and the lactate content Units were converted to nmol µg$^{-1}$ protein.

### Intracellular pH assay

Cancer cells ($5 \times 10^5$ cells mL$^{-1}$) were inoculated in 24-well plates or in confocal dishes and cultured overnight. Then, 200 µL of different materials (200 µg mL$^{-1}$) were added to the well plates to treat the cells for 24 h. Bis(acetoxymethyl) 3,3′-(3′,6′-bis(acetoxymethoxy)−5-((acetoxymethoxy)carbonyl)−3-oxo-3H-spiro[isobenzofuran-1,9′-xanthene]−2′,7′-diyl) dipropanoate (BCECF AM, pH fluorescent probe, 5 µM, Beyotime Institute of Biotechnology, China) was incubated with cells for 15 min, at the same time, pH 5.8 and pH 8.0 standards were set up to incubate with cells to obtain standard curves, and photos were taken using CLSM (Ex: 450 nm; Em: 450 nm) and pseudo-colored using Image J (Version 2.0.0) was used to obtain a positive correlation between the magnitude of pH and the shade of purple and quantify their contribution to fluorescence intensity.

### Detection of intracellular adenosine triphosphate (ATP) content

Cancer cells ($5 \times 10^6$ cells mL$^{-1}$) were inoculated in 6-well plates and incubated against the wall overnight, after treatment of cells with 200 µL of different samples (200 µg mL$^{-1}$) for 0, 3, 6, 9, 12, and 24 h. The material was removed and washed well. The trypsin-digested cell suspension was centrifuged ($12,000 \times g$, 10 min) and the lower cell layer was collected. In all, $10^6$ cells mL$^{-1}$ cell suspension was prepared in PBS, sonicated at constant intervals for 3 min (200 W, 2 s sonication, 1 s pause) and the supernatant was collected by centrifugation ($9060 \times g$, 10 min) at 4 °C. ATP content was determined by referring to the procedures of the ATP kit (Beyotime Institute of Biotechnology, China).

### Construction of monolayer-layer transwell model and evaluation of the performance of nanomotor penetrating BBB

(1) 200 µL, bEnd.3 cells (cell density of $1.5 \times 10^5$ cells mL$^{-1}$) were added to the nest of the upper chamber as Day 0 and cultured for 10 days. 500 µL of U87 (cell density of $2.5 \times 10^5$ cells mL$^{-1}$) were inoculated in the corresponding lower chamber at Day 8. After Day 10, the original medium in the upper chamber was discarded and a concentration of 200 µg mL$^{-1}$ of material (PMSe, Ang-PMSe, PAMSe, and Ang-PAMSe) was added and transferred to the wells inoculated with U87 and incubated for 24 h. The rest of the conditions remained unchanged, and the lower chamber was used as a control group without any cells

inoculated. The amount of supernatant from the lower chamber and the material contained in the cells after lysis were collected and recorded as the material transfer volume.

$$\text{Transport efficiency}(\%) = ([m(\text{materials transported}))/ \\ (m(\text{total materials added})] \times 100\%) \quad (3)$$

(2) The bEnd.3 cells with high expression of low-density lipoprotein (LRP) and HUVECs with low expression of LRP were used as experimental controls. 200 μL of bEnd.3 and HUVECs cells with a cell density of $1.5 \times 10^5$ mL$^{-1}$ were added separately to the nest of the upper chamber, and 500 μL of U87 with a cell density of $2.5 \times 10^5$ mL$^{-1}$ were inoculated in the corresponding lower chamber for 24 h. Afterwards, the original medium in the upper chamber was discarded and 200 μg mL$^{-1}$ of material (Ang-PAMSe) was added and transferred to the wells inoculated with U87. After 24 h of incubation, the supernatant of the lower chamber and the amount of material contained in the cells after lysis were collected and the transfer rate was calculated. Meanwhile, the structural compactness of the BBB layer before and after incubation with the material was observed by CLSM.

(3) The liquid in the lower chamber was collected and the fluorescence intensity of FD-4 was measured using a fluorescence spectrophotometer (Ex: 492 nm and Em: 520 nm).

### Construction of multilayer symbiotic BBB model and evaluation of the performance of nanomotor penetrating BBB

(1) At Day 0, 100 μL, U-87 MG cancer cells (cell density of cell density of $1.0 \times 10^6$ cells mL$^{-1}$) were inoculated into the inverted transwell chamber bottom surface and incubated continuously to form a tumor cell layer.

(2) On Day 1, the chambers from step 1 were orthotopically placed and 200 μL bEnd.3 cells (cell density of $1.5 \times 10^5$ cells mL$^{-1}$) were added to their interior for continuous incubation to build the BBB layer.

(3) On Day 8, 500 μL of U-87 MG (cell density of $2.5 \times 10^5$ cells mL$^{-1}$) was inoculated in the transwell lower chamber after continuous incubation to create ROS/iNOS concentration gradient.

(4) At Day10, the original medium in the chambers was discarded, the control group was replaced with fresh cell culture medium, and 200 μg mL$^{-1}$ of fluorescently labeled Cy5-PMSe and Cy5-Ang-PAMSe nanomotors were added to the experimental group, respectively, and the above chambers were transferred to the wells inoculated with cancer cells and incubated for 24 h.

(5) At Day11, liquid in the chambers, liquid and cells in the well plates were collected separately and the fluorescence intensities of these materials were measured using a fluorescence spectrophotometer (Ex: 650 nm, Em: 670 nm) to calculate the transport rate.

### Construction of in vitro dynamic 3D BBB model and evaluation of the performance of nanomotor penetrating BBB

(1) The bEnd.2 cells (100 μL, $2.3 \times 10^7$ cells mL$^{-1}$) were added to the lower channel through the inlet and the chip was immediately turned over to allow bEnd.2 cells to grow against the side of the porous membrane. 2 days later, the chip was placed on the front side, bEnd.2 cells were inoculated again to spread the cells across the bottom of the channel, and incubation was continued for 2 days to build a vascular-like structure in the lower channel.

(2) GL261 cells (100 μL, $2.3 \times 10^7$ cells mL$^{-1}$) were inoculated into the upper channel of the chip and incubated in the incubator for 3 days.

(3) The above constructed BBB chip was connected to the syringe pump via a connector and the cell culture medium was flowed through the lower channel to acclimatize the BBB chip to flow conditions for 24 h.

(4) The cell culture medium containing the material (cy5-PMSe passive nanoparticles and cy5-Ang-PAMSe nanomotors) was circulated through the lower channel (BBB), and the liquid in the upper channel and the liquid circulating in the lower channel were collected after 24 h. The concentration of the material in the solution was detected, and the apparent permeability ($P_{app}$) was calculated according to the calculation formula.

$$P_{app} = \frac{V_{upper} \times C_{upper}}{A \times t \times \frac{C_{lower} \times V_{lower} + V_{upper} \times C_{upper}}{V_{lower} + V_{upper}}} \quad (4)$$

$V_{upper}$ and $V_{lower}$ are the volumes of the liquid medium in the upper and lower channels at moment $t$, respectively; $A$ is area of membrane which is 0.69 cm$^2$ in our chip model; $C_{upper}$ and $C_{lower}$ are the material concentrations in the liquid media collected in the upper and lower channels, respectively.

### Cellular uptake performance assessment

GL261-Luc cells ($1 \times 10^5$ cells mL$^{-1}$) were placed in culture dishes and incubated for 12 h. 200 μL, 200 μg mL$^{-1}$ of different samples were added and incubated with the cells. The cells were stained with 200 μL DiO (10 μM in DMSO) and 200 μL Hoechst 33342 (10 μg mL$^{-1}$) for 15 min to color the cell membrane and nucleus respectively, and the uptake of the material by the cells was observed by CLSM. GL261-Luc cells ($1 \times 10^5$ cells mL$^{-1}$) were inoculated in 6-well plates and cultured overnight. 1 mL of different samples was added and treated for 24 h. The supernatant was collected, the cells were lysed with cell lysis solution and the cell suspension was collected. The fluorescence intensity of the samples was measured at 650 nm (Ex: 650 nm, Em: 667 nm). The fluorescence intensity of the cell suspension was recorded as the uptake value and the fluorescence intensity of the mixture of supernatant and cells was recorded as the total value.

$$\text{Cell uptake efficiency}(\%) = [m(\text{uptake})/m(\text{total})] \times 100\% \quad (5)$$

### Cellular organelle localization of nanomotors

GL261-Luc cells ($1 \times 10^5$ cells mL$^{-1}$) were placed in culture dishes and incubated for 12 h. Cells were incubated with Cy5-labeled Ang-PAMSe and Ang-PAMSe/TLND (200 μL, 200 μg mL$^{-1}$) respectively for 5 min, aiming to assess co-localization of material with lysosomes, in which lysosomes were labeled green by LysoGreen (1 μM in DMSO, Keygen Bio. labeled green by LysoGreen (1 μM in DMSO, Keygen Bio. Tech.), and nucleus were stained blue by Hoechst 33342 (10 μg mL$^{-1}$); after incubation with cells for 2 h, the material was observed separately with Golgi tracker green (200 μL, 250 μg mL$^{-1}$ in Hank's/10 mM N-2-hydroxyethylpiperazine-N-2-ethane sulfonic acid (HEPES), Keygen Bio. Tech.) and mitochondria (mitochondrial fluorescent dye, 200 μL, 1 nM in DMEM) respectively.

### Expression of cytosolic calreticulin (CRT)

GL261-Luc cells were inoculated into culture dishes and incubated overnight. 200 μL of different samples (200 μg mL$^{-1}$) were added to treat the cells for 24 h. The cells were washed thoroughly with PBS and stained with primary Anti-calreticulin antibody (dilution 1:500, catalog number: 27298-1-AP, Proteintech) for 30 min, followed by three washes of PBS for 5 min each, and then incubated with secondary antibody goat anti-rabbit. The cells were then incubated with the secondary antibody CoraLite488-conjugated Goat Anti-Rabbit IgG(H + L) (dilution of 1:50, Cat: SA00013-2, Clone: A0428, Lot: 20000127, Proteintech) (Ex 495 nm, Em 505–550 nm) for 15 min. Next, the nuclei were stained with Hoechst 33342 (200 μL, 10 μg mL$^{-1}$, Keygen Bio. Tech.) and CRT expression was observed with CLSM and quantified using Image J.

### Extracellular release of high-mobility group box 1 (HMGB1)

GL261 cells were inoculated in 12-well plates and incubated overnight. 200 µL of different samples (200 µg mL$^{-1}$) were added to treat the cells for 24 h. The cell culture supernatant was collected to detect HMGB1 content using an enzyme-linked immunoassay kit (MeiMian ELISA kit).

### Cell viability assessment

Evaluation of the biocompatibility of PAMSe nanomotor raw materials. After sufficient degradation of PAMSe in H$_2$O$_2$ (0.002%) for 48 h, the lower dissociates were obtained by centrifugation and freeze-dried. The dissociates were incubated with tumor cells (MCF-7) at different concentrations (20, 50, 100, 150, 200 µg mL$^{-1}$) for 48 h and cell activity was assayed using MTT (Apexbio ApexBio Technology LLC, Cat No: B7777) reagent. To evaluate the effect of PAMSe nanomotors on cellular activity in HUVECs (with low ROS concentrations) and in the cancer cell environment (with high ROS concentrations). Different concentrations of PAMSe (20, 50, 100, 150, 200 µg mL$^{-1}$) were incubated with HUVECs and MCF-7 cells ($1 \times 10^5$ cells mL$^{-1}$) for 48 h, and cell activity was detected. To assess the effect of different samples on the activity of glioblastoma (GBM) cells, different samples (200 µg mL$^{-1}$) were added to 96-well plates with GBM cells and incubated for 24 h. Cell activity was measured by MTT.

### Pharmacokinetic (PK) study of Ang-PAMSe/TLND nanomotor

Healthy female SD rats (6-8 weeks) were injected with Ang-PAMSe/ TLND (200 µL, 3 mg mL$^{-1}$) via tail vein and blood was collected from the orbit using capillary tubes at 1 h, 2 h, 4 h, 8 h, 12 h, 24 h and 48 h, respectively. The blood was centrifuged ($560 \times g$, 10 min) and the serum was collected. 100 µL of serum was fixed to 1 mL with PBS and the TLND concentration was measured using a UV spectrophotometer.

### Animals and tumor models

C57BL/6J mice (6–8 weeks, 18–20 g, female) and Sprague-Dawley (SD, 6–8 weeks, female) rats were purchased from Jiangsu Alingfei Biotechnology Co. All animals were bred in a pathogen-free facility with a 12 h light/dark cycle at $20 \pm 3$ °C and 40–50% humidity and had ad libitum access to food and water. All animal experimental operations were in accordance with the specifications of the Guide for the Care and Use of Laboratory Animals, and all experimental procedures and protocols were approved by the Animal Experimentation Ethics Committee of Nanjing Normal University (approval number: IACUC-20200802-1).

### Establishment of a GBM model in mice

C57BL/6J mice (6–8 weeks, 18–20 g, female) were first anesthetized by inhalation of 1–5% isoflurane mixed with oxygen, and GL261-Luc cells (8 µL, 10$^6$ cells per mouse) suspended in a mixture of PBS and Matrigel (1:1, v-v) were slowly injected into the brain using a brain stereotaxic instrument (RWD Life Science Co., Shenzhen), positioned to (1.8 mm, 0.6 mm, 2 mm depth) using the fontanel point of origin, where the craniotomy operation was kept consistent provided that the mice without cell injection were referred to as the sham-operated group as a control. To establish a primary and distal GBM models, GL261 cell suspension (PBS and Matrigel (1:1, v-v)) was injected (1.8 mm, −0.6 mm, 2 mm depth) into the left side of the brain of C57BL/6J mice to treat them as primary tumors, and GL261-Luc cell suspension (PBS/Matrigel = 1:1) was positioned (1.8 mm, 0.6 mm, 2 mm depth) orthotopic and injected into the right side of the brains of C57BL/6J mice as distal metastases. Mice were injected with Gd-DTPA (0.25 mmol kg$^{-1}$, Shanghai Macklin Biochemical Co., Ltd) intraperitoneally for 5 min and then underwent T$_2$-weighted MRI (MesoMR23-060 H-I). All mice undergoing chemiluminescence imaging required intraperitoneal injection of 100 µL, 150 mg kg$^{-1}$ luciferase substrate D-luciferin potassium (ApexBio Technology LLC) before imaging, and the growth of gliomas was monitored 10 min later using a small animal imager (IVIS

Lumina III, living image system 4.5.5) bioluminescence system, with the total amount of fluorescence intensity used as a measure of tumor size. The maximum volume of the mouse tumor did not exceed 1 cm$^3$, which was approved by the animal care committee of Nanjing Normal University. The anesthetic mice were euthanized by cervical dislocation.

### Evaluation of in vivo targeting performance of nanomotors

Mice that had established intracranial GBM were subjected to bioluminescence imaging on day 5 or day 11 after molding. Mice with tumors of close size were randomly divided into 4 groups. PBS and Cy5-labeled materials (Ang-PMSe, PAMSe and Ang-PAMSe) were injected into the mice via the tail vein respectively, and 24 h later, the mice were anesthetized by inhaling 1–5% isoflurane mixed with oxygen and the fluorescent images were scanned at Ex: 650 nm, Em: 667 nm using a near-infrared fluorescence imaging system (IVIS Lumina III, living image system). Next, the mouse's brain, heart, liver, spleen, lungs and kidneys were removed and imaged again. Images were acquired and analyzed using IVIS Lumina III, living image system 4.5.5 software.

### Biological distribution of nanomotor

The tumor-bearing mice were randomly divided into 4 groups, and Cy5-Ang-PMSe (chemical recognition targeting), Cy5-PAMSe (chemokine targeting) and Cy5-Ang-PAMSe were injected via tail vein respectively, where PBS was injected into tail vein as blank control. After 24 h, the major organs of the mice including heart, liver, spleen, lung, kidney, brain and whole brain were collected, the surfaces were rinsed with PBS, blotted with filter paper and weighed. The tissues were then cut into grinding tubes and added to RIPA lysate at a ratio of 1 mL:100 mg tissue, and the tissues were ground with preset parameters on a tissue grinder to prepare a homogenate, centrifuged at $810 \times g$ for 5 min, the supernatant was removed and fixed to 2 mL with RIPA lysate, and the fluorescence intensity of the material was measured using a fluorescence spectrophotometer (Ex: 650 nm, Em: 667 nm). The amount of material in each tissue was calculated based on the fluorescence intensity–concentration standard curve. The percent injected dose rate (%ID) was then calculated according to the following formula: ID% = (mass of sample in organ tissue/total mass of injected sample) × 100%.

$$ID\% = [m\,(\textbf{sample in organ tissue})/m\,(\textbf{total injected sample})] \times 100\% \quad (6)$$

### Tumor penetration performance assessment of nanomotors

The mouse brains were removed after imaging treatment for in vivo targeting experiments, and frozen sections were prepared after freezing. Subsequently, tumor tissues, blood vessels and cell nuclei were immunofluorescently stained with anti-CD31 antibody (dilution 1:100, catalog number: ab222783, clone: EPR17260-263, Abcam), coraLite488-conjugated Goat Anti-Rabbit IgG(H + L) (dilution of 1:50, Cat: SA00013-2, Clone: A0428, Lot: 20000127, Proteintech) (Ex 495 nm, Em 505–550 nm), and DAPI (Ex 365 nm, Em 420–470 nm) seperately. Sections were scanned by the tissue scanner pannoramic MIDI (3D HISTECH, Hungary) and processed by pannoramic viewer software. Fluorescence intensity was analyzed by Image J (Version 2.0.0) software.

### In vivo antitumor effects of nanomotors in GBM model

Mice that had established intracranial GBM as described above were subjected to bioluminescence imaging on day 5 post-modeling. Mice with tumors of similar size were randomly divided into 5 groups of 17 mice each, while another group was set up as a sham-operated group and left untreated for the subsequent treatment. Tumor-bearing mice were given intravenous injections of each preparation (PBS, TLND, Ang-PMSe/TLND, Ang-PAMSe and Ang-PAMSe/TLND) on days 5, 7, 9,

11, and 13, respectively.) for five times, corresponding to days 5, 7, 9, 11, 13, and 15 bioluminescence imaging of mouse brain tumors was performed using the in vivo bioluminescence IVIS III system (Fig. 5c) to monitor the magnitude of fluorescence intensity of brain tumors and changes in mouse body weight (three mice were taken from each group). After termination of treatment on day 15, three mice from each group were euthanized. Blood was taken from the orbits and the centrifuged serum assayed for IL-6 and TNF-α (MeiMian ELISA kit) cytokines for both data sets were available.

In addition, the procedure of inoculating GL261 cells was unchanged, and only the administration was changed to take place on day 11, and the drug was administered every 2 days for 5 times. The whole brains of mice were taken on day 20, and tissue sections and H&E staining were performed to observe and compare the brain tumor sizes.

### Biosafety evaluation of nanomotor in vivo
The H&E staining of each organ (heart, liver, spleen, lung and kidney) in the orthotopic GL261 glioblastoma tumor-bearing C57BL/6J at the end of the treatment. In addition, healthy C57BL/6J mice (without tumors) were injected with various materials (PBS, TLND, Ang-PMSe/TLND, Ang-PAMSe and Ang-PAMSe/TLND). At the end of administration, cortical tissues were collected separately and stained with H&E to observe the morphological changes in the cerebral cortex to assess the neurotoxicity of the material.

### Flow cytometry
The anti-mouse antibodies used in the experiment are the following, primary antibodies: Anti-calreticulin antibody (dilution 1:500, catalog number: 27298-1-AP, Proteintech), Anti-CD31 antibody (dilution 1:100, catalog number: ab222783, clone: EPR17260-263, Abcam), FITC-anti-mouse CD11c antibody (dilution 1:200, catalog number: 117305, clone: B277031, Lot: N418, Biolegend), PE-anti-mouse CD80 antibody (dilution of 1:40, catalog number: 104707, clone: 16-10A1, Lot: B340153, Biolegend), APC-anti-mouse CD86 antibody (dilution of 1:80, catalog number: 105071, clone: GL-1, Lot: B323580, Biolegend), FITC-anti-mouse CD3 antibody (dilution of 1:50, catalog number: 100203, clone: 17A2, Lot: N418 Biolegend), APC anti-mouse CD8 antibody (dilution of 1:80, catalog number: 100711, clone: 53-6.7, Lot: B329662, Biolegend), Percp-cy5.5-anti-mouse CD11b antibody (dilution of 1:50, catalog number: 101228, clone: M1/70, Lot: B353753, Biolegend), FITC-anti-mouse CD206 antibody (dilution of 1:100, catalog number: 141704, clone: C068C2, Lot: B350306, Biolegend), FITC-anti-mouse CD45 antibody (dilution of 1:100, catalog number:103108, clone: 30-F11, Lot: B363202, Biolegend), APC-anti-mouse PDL1 antibody (dilution of 1:50, catalog number: 124312, clone: 10F.9G2, Lot: B277024, Biolegend). Secondary antibodies: CoraLite488-conjugated Goat Anti-Rabbit IgG(H + L) (dilution of 1:50, Cat: SA00013-2, Clone: A0428, Lot: 20000127, Proteintech).

### Detection of mature dendritic cells (DCs) in lymph nodes and tumor tissue
Orthotopic tumor-bearing mice (3 per group) were taken at the end of the above treatment and euthanised. Mouse lymph nodes and tumor tissue were taken, ground, and placed in staining buffer (Multisciences (Lianke) Biotech, Co., Ltd. Lot. A10752) through a 70 μm pore size filter (Thermo Fisher Scientific). The above cell suspension was centrifuged (200 × g, 5 min) and the supernatant was removed. Staining buffer was added to blow the cells well. First 10 μL of FC blocking (1 μg/10⁶ cells, Miltenyibiotec, Lot. 5210508639) was added to each sample tube, and they were shaken and mixed well to avoid non-specific antibody binding. After incubation on ice for 30 min, the supernatant was discarded by centrifugation (200 g, 5 min), and 100 μL of staining buffer was added for staining. Mature DCs in lymph nodes were detected by staining on ice for 45 min using FITC-anti-mouse CD11c antibody (dilution 1:200, catalog number: 117305, clone: B277031, Lot: N418, Biolegend), PE-anti-mouse CD80 antibody (dilution of 1:40, catalog

number: 104707, clone: 16-10A1, Lot: B340153, Biolegend) and APC-anti-mouse CD86 antibody (dilution of 1:80, catalog number: 105071, clone: GL-1, Lot: B323580, Biolegend). The cells were resuspended with 100 μL of staining buffer, filtered through non-woven fabric, and placed in a flow cytometer for analysis.

### Polarization of macrophages and expression of PDL1 in tumor cells
The brain tissues were obtained from mice after treatment. The brain tissues were cut with scissors, digested with collagenase type IV containing 0.1%, and then prepared into cell suspensions by sieve grinding, and the samples were resuspended by 40% and 70% percoll and centrifuged to collect non-lymphocytes and lymphocytes for use. Lymphocytes were labeled with Percp-cy5.5-anti-mouse CD11b antibody (dilution of 1:50, catalog number: 101228, clone: M1/70, Lot: B353753, Biolegend), FITC-anti-mouse CD206 antibody (dilution of 1:100, catalog number: 141704, clone: C068C2, Lot: B350306, Biolegend) and APC-anti-mouse CD86 antibody (dilution of 1:80, catalog number: 105071, clone: GL-1, Lot: B323580, Biolegend) to detect the polarization of macrophages. The expression level of PDL1 in tumor cells was detected with FITC-anti-mouse CD45 antibody (dilution of 1:100, catalog number:103108, clone: 30-F11, Lot: B363202, Biolegend) and APC-anti-mouse PDL1 antibody (dilution of 1:50, catalog number: 124312, clone: 10F.9G2, Lot: B277024, Biolegend).

### Detection of T cell infiltration in mouse brain tumor
The perfused mouse brains were rinsed in Hank's buffer solution to remove the surface foreign matter, and the brain tissues were cut to the size of rice grains with surgical scissors and added with 1 mg mL⁻¹ collagenase IV (Sigma-Aldrich) and 50 U mL⁻¹ DNase I (Sigma-Aldrich, 15 kU) in 1640 medium in 5 mL of tissue digest, placed on a 37 °C shaker, and fully digested for 1 h. The entire digested tissue homogenate was poured into a 70 μm nylon filter, and the tissue residue was filtered by grinding the tissue with the flat end of a syringe, the above cell suspension centrifuged (200 × g, 5 min) to remove the supernatant. The cells were blown well by adding staining buffer. CD3⁺CD8⁺ T cells were stained with FITC-anti-mouse CD3 antibody (dilution of 1:50, catalog number: 100203, clone: 17A2, Lot: N418 Biolegend) and APC anti-mouse CD8 antibody (dilution of 1:80, catalog number: 100711, clone: 53-6.7, Lot: B329662, Biolegend). Flow cytometry data was obtained from flow cytometry (BD Accuri C6 Plus flow cytometer) and analyzed with CytExpert (2.3) software and FlowJo (Version 10).

### Activation of systemic immune system performance assessment
In this model, the sham-operated group was still set up, and mice with established bilateral gliomas as described above were randomly divided into 3 groups of 6 mice each. All treatment procedures were performed by giving only local treatment of drugs to the in-situ tumors, while as distal tumors were not directly administered. That is, 200 μg mL⁻¹ (PBS, TLND and Ang-PAMSe/TLND) was administered locally to the in-situ tumors on day 7 and day 12 after implantation respectively, and the growth of the distal tumors was observed by MRI and chemiluminescence imaging on day 7 and chemiluminescence imaging on day 14. At day 14, three mice from each group were euthanized. To accurately assess the infiltration of T cells in the distal tumors, we selected the brain region where the distal tumors were located (right brain region) for detection of CD3⁺CD8⁺ T cells using flow cytometric assays (staining and assay as above).

### Assessment of immune memory capacity
The tumors were continuously observed after day 20 treatment, and thereafter the mice were bioluminescently imaged every 5 days to continuously observe the fluorescence time point of tumor reappearance in the Ang-PAMSe/TLND group.

## Statistical analysis

Microsoft Excel (2016), GraphPad Prism (Version 8), Origin 2018 software are used for numerical processing and drawing. Figure 6f values were expressed as the mean ± standard error (s.e.m.). All other values are expressed as mean ± standard deviation (s.d.) unless indicated otherwise. An unpaired two-tailed $t$ test was performed to compare the statistical significance between the two groups. For multiple comparisons, one-way ANOVA was performed by SPSS13.0. The $p$ values and statistical tests performed are indicated in the figure and associated legend, respectively. A $p$ value of ≤0.05 was considered statistically significant.

## Reporting summary

Further information on research design is available in the Nature Portfolio Reporting Summary linked to this article.

## Data availability

All data that support the findings of this study are available within the article and the Supplementary Information. Source data are provided with this paper.

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

## Acknowledgements

The work was supported by National Natural Science Foundation of China (No. 22175096, 22275095), Jiangsu Key Laboratory of Biofunctional Materials, Jiangsu Collaborative Innovation Center of Biomedical Functional Materials, the Qinglan Project Foundation of Colleges and Universities of Jiangsu Province, the Postgraduate Research & Practice Innovation Program of Jiangsu Province (KYCX22_1545) and the Priority Academic Program Development of Jiangsu Higher Education Institution. We thank Mr. Songyuan Zhang for his valuable advice on English writing.

## Author contributions

C.M., J.S., M.W., and H.C. conceived the project and designed the experiments. H.C., T.L., Z.L., Y.T., and Z.Z. performed the experiments. H.C., S.T., Z.L., J.T., and N.L. analyzed and interpreted the data in this study. H.C., M.W., and C.M. wrote the manuscript draft. The final draft of the manuscript was approved by all the co-authors.

## Competing interests

The authors declare no competing interests.
