## [Peer Review File · Nature Communications]

A nitric-oxide driven chemotactic nanomotor for enhanced immunotherapy of glioblastomaREVIEWER COMMENTS

Reviewer #1 (Remarks to the Author): with expertise in nanoparticles, glioblastoma

This manuscript reports on an innovative strategy based on NO-propelled nanomotors for the immunotherapy against glioblastoma multiforme. Overall, this is a nice work, and experiments detailed described and reported. However, some general concerns need to be addressed before its reconsideration for publication.

- 1) First of all, the language and the style need a strong revision. In many points the reading of the paper is indeed difficult
- 2) Secondly, the structure and the logic flow of the manuscript should be improved as well for clarity
- 3) In vitro experiments concerning BBB crossing are quite oversimplified and performed on a traditional static 2D transwell system. Tests on more complex in vitro models are suggested
- 4) Appropriate controls should be showed and clearly described for each performed experiment

Reviewer #2 (Remarks to the Author): with expertise in glioblastoma

The authors are to be commended for delivering a manuscript covering a very comprehensive, innovative, fresh and elegant nanomedicine-based technology. In essence, the authors developed, described and tested a 'nanomotor' drug, able to cross the blood-brain barrier, and concentrating in the tumor areas due to a diffusion according to a chemotactic gradient, to deliver a drug as payload that is to induce immunogenic cell death in glioblastoma cells. Without any doubt this is a fresh and innovative idea and concept with large theoretical potentials for GBM therapy. There are however some major concerns as detailed hereunder.

1. First and foremost, the authors need to substantially improve the English language: for this, the authors are advised to sit together with a native speaker to explain in detail which messages they exactly want to convey: the grammar, the syntaxis, the articles used or not used, verb conjugations etc really are important to make the reader understand the message. As it is, the poor language substantially detract from (and sometimes makes it impossible to understand)the original message and research results the authors want to communicate. It would be a pity if all this would be refuted based on communication issues.

2. It might be good to include team members or authors who are familiar with the disease. Glioblastoma is not just a paradigm malignancy in the brain. The major problems to overcome might not be the blood-brain barrier, which often is partially disrupted anyway in this disease, but the widespread infiltration in the normal brain and the cellular heterogeneity. Exactly those points are not recapitulated by the model chosen (GL261 in mice), which in addition is far more immunogenic than the real human glioblastoma environment that should be tackled. Therefore, it is necessary to at least show comparable results in other models e.g. human xenograft models since none of them alone, is fully able to mimic the complex human situation. GL261 as model for human GBM unfortunately is way too unconvincing for these types of read-out. In the manuscript itself, immune checkpoint therapy is mentioned several times as if it were a relevant standard in GBM. Immune checkpoint blockers in GBM at this stage are completely irrelevant and should not be referred to as the golden standard of immunotherapy just because they take that position in non-neuro-oncology settings.

3. In addition to the stepwise efficacy data of the approach, it would be good to generate and show at least some minor toxicity/safety data: liver, lung, neurotox ?

4. The manuscript would benefit from a more rigid structure in which from the beginning the different objectives of the research are clearly and stepwise communicated. Therefore, the authors could use the frame/structure they refer to at the end of the manuscript i.e. the different 'links' 1-7.

5. In fig.3j, the ATP concentration goes down in the experimental condition. In the light of the envisioned ICD, this is a bit harder to understand since ATP could/should (?) be considered as a

read-out for ICD as well ?

6. in fig.5c, it is shown that IV therapy starts at day 5 after tumorinoculation. Even in the GL261 model, often 10 days are required for proper grafting of the tumor cells in the brain. Therefore, apart from the remarks made about the model itself, it is required to perform the same experiment but with a start of therapy at earliest 10 days after tumorinoculation.

Reviewer #3 (Remarks to the Author): with expertise in nanomotors

In this manuscript a nanoparticle is developed which contains arginine and TLND and which is modified with angiopeptide. Arginine is converted under ROS exposure with the enzyme iNOS into NO. Both NO and TLND contribute to an immunological response leading to tumor cell killing. These latter aspects are known and also demonstrated in this paper. The additional claim is that the particles developed are nanomotors and as such improve the therapeutic efficacy. This claim however is not really substantiated and more quantitative validation is required for this paper to be acceptable for publication.

First of all, the authors show that the accumulated release of NO after 96 h in the mimicked tumor cell environment is 4.7 μM . This means that the release per hour is very low and I wonder how this could lead to the kinetic plots observed. Secondly, the particles are spherical and as such have no intrinsic motility behavior. The entire motility has thus to be created from the gradient created around the particles. The authors should quantify this gradient. They should for example determine the ROS production in the microfluidic channels and the gradient created by the cell lysate.

The authors should calculate the force necessary to drive the motors and indicate if there is sufficient NO present/produced to achieve this kind of force over prolonged times.

The video s5 shows that all particles have directional motion and not mere Brownian as you would expect from non-motor systems. This indicates that there is an additional hydrodynamic flow process in the system. Interpretation of these data is therefore inconclusive.

The reason why the arginine loaded nanomotors behave worse than the covalent ones is not explained. Is this a result of loading efficiency? What is the loading capacity of the covalent arginine systems? Quantitative information is missing.

The quality of the colocalization studies in fig 4l-n is poor and should be improved to allow any conclusions to be drawn.

Have the authors established that the BBB is intact in their mouse model? If not, then this is merely an EPR effect that allows particles to accumulate.

Fig S38 what is the physiological explanation that the particles without arginine end up in the lungs?

How does the accumulation in the brain compare to the other organs?

What is the reason for the 40% death rate in the mice in VI, as the tumor has been depleted?

Other technical questions/comments

The labeling with angiopeptide is not quantified.

What are the iNOS levels of the cancer cells used in fig 3?

Video S2: it is unclear what the 10 frames are and what we are looking at. A better explanation should be provided.

From video S4 the cell uptake efficiency is not clear.

Regarding the BBB in vitro model. The observed transcytosis transfer of the controls is very high (20%), which could indicate that there is no tight cell layer. The authors should have done controls with different particles and for example dextran to establish the integrity of the cell layer.

Supp fig 8 doesn't provide any useful information.

Fig S36 cell viability is surprisingly high, in comparison with the later in vivo studies. Can the authors provide an explanation?

Fig S37, the X axis is unspecified, It is not clear what is compared.

Fig 5 d please specify groups V and VI in the figure caption.

Reviewer #4 (Remarks to the Author): with expertise in glioblastoma, cancer nanotechnology

Chen et al present the design of a complex nanoparticle with multiple functions. The overall premise is intriguing but not all aspects are well supported, especially the antitumor immune response. First, the nanoparticle has interesting release profiles especially in different media. Second, the in vitro data are supportive of the chemotactic spread of the nanomotors across an iNOS gradient. Further, the in vivo data seem to corroborate the in vitro findings. However, the data supporting the main claim of the work, that is activation of the innate arm of immunity within the tumor, are not convincing. Therefore, this work requires new essential experimentation and significant revisions.

Major issues

- The immunological aspects of the work are not well supported. The issue in GBM is not necessarily the absence of neoantigens but the rather immunosuppressed nature of the tumor immune microenvironment. It is not clear why DCs and TAMs increase their cross-presentation while no direct function of the NO is targeting the activation and proper reprogramming of the antigen-presenting cells. Since this is the main key function of the particle, the paper is missing important data supporting this function.
- The authors should clarify which are the tumor-draining lymph nodes for a GBM. This is not trivial or standard. Why didn't the authors collect and analyze DCs and macrophages from the tumor itself?
- The main hypothesis of the paper is based on the activity of NPs in cancer cells. This assumes the NPs were taken up by cancer cells. The in vivo studies should include flow cytometry studies analyzing the cellular uptake of NPs in the tumor. Many studies show NPs are predominantly taken up by phagocytic cells and not glioma cells.
- Are DC activated due to the immunogenicity of the NPs (NP being the antigen) or due to cancer-associated antigens?
- It is not clear why or how Tregs were decreased.

Minor issues

- A detailed biodistribution and PK study should be performed providing the % ID in each organ.
- iNOS inhibition has been implicated in the reduction of glioma stem cell population. The authors should investigate whether the antitumor effect is due to decrease of the stem cell content of the GBM and not directly linked to an antitumor immune response.
- Unacceptable language is used: "Mice were executed..."

Response to reviewers

Reviewer #1 (Remarks to the Author): with expertise in nanoparticles, glioblastoma

This manuscript reports on an innovative strategy based on NO-propelled nanomotors for the immunotherapy against glioblastoma multiforme. Overall, this is a nice work, and experiments detailed described and reported. However, some general concerns need to be addressed before its reconsideration for publication.

Answer: We thank the reviewer so much for his/her comment on the novelty of our work. We are also grateful for this opportunity to revise these problems in the manuscript.

1-1) First of all, the language and the style need a strong revision. In many points the reading of the paper is indeed difficult

Answer: We apologize for the poor language and the style of our manuscript. We have now rewritten the manuscript. We hope that the writing style and language level can meet the publishing requirements of your journal.

1-2) Secondly, the structure and the logic flow of the manuscript should be improved as well for clarity

Answer: Thanks so much for the reviewer's kind suggestion.

According to the suggestions of the reviewers, we have improved the structure and logical flow of the manuscript, so that readers can better understand the research ideas of the manuscript. Thanks again for the kind reminding of the reviewer, we have marked the revised part in red in the revised manuscript.

1-3) In vitro experiments concerning BBB crossing are quite oversimplified and performed on a traditional static 2D transwell system. Tests on more complex in vitro models are suggested.

Answer: We thank the reviewer a lot for this valuable advice.

Based on the reviewers' suggestions, we constructed more in vitro complex static and dynamic BBB models to further evaluate the ability of nanomotors to penetrate the BBB. The experimental models of in vitro experiments concerning BBB crossing in the revised manuscript can be summarized as Fig. R1, including monolayer and multilayer cell models (corresponding to a and b), and static and dynamic models (corresponding to c), respectively.

Fig. R1 In vitro BBB penetration model. **a** Static monolayer BBB model in vitro. **b** Static multicellular BBB model in vitro. **c** Dynamic microfluidic multicellular BBB model in vitro.

In addition to the transwell monolayer cell penetration model available in the preliminary manuscript, we constructed a transwell model with multiple layers of cells (bEnd.3 and U-87 MG). Compared with brain endothelial cell (bEnd.3) monoculture, multicellular co-culture of bEnd.3 cells with glioma cells (U-87 MG) can better mimic in vivo microenvironment, increase tight junctions of BBB, and induce expression of specific receptors and transporters (*Acta Neurobiol. Exp. (Wars)*, 2011, 71, 113; *Neurosci. Methods* 2013, 212, 211).

As shown in Supplementary Fig. 33a, we constructed an in vitro multilayer cell symbiotic BBB model. The specific steps are as follows.

(1) At Day 0, 100 μL , U-87 MG cancer cells (cell density of $1.0 \times 10^6 \text{ cell mL}^{-1}$) were inoculated into the inverted transwell chamber bottom surface and incubated continuously to form a tumor cell layer.

(2) On Day 1, the chambers from step 1 were orthotopically placed and 200 μL bEnd.3 cells (cell density of $1.5 \times 10^5 \text{ cell mL}^{-1}$) were added to their interior for continuous incubation to build the BBB layer.

(3) On Day 8, 500 μL of U-87 MG (cell density of $2.5 \times 10^5 \text{ cell mL}^{-1}$) was inoculated in the transwell lower chamber after continuous incubation to create a ROS/iNOS concentration gradient.

(4) At Day 10, the original medium in the chambers was discarded, the control group was replaced with fresh cell culture medium, and 200 $\mu\text{g mL}^{-1}$ of fluorescently labeled Cy5-PMSe and Cy5-Ang-PAMSe nanomotors were added to the experimental group, respectively, and the above chambers were transferred to the wells inoculated with cancer cells and incubated for 24 h.

(5) At Day 11, liquid in the chambers, liquid and cells in the well plates were collected separately and the fluorescence intensities of these materials were measured using a fluorescence spectrophotometer (Ex: 650 nm, Em: 670 nm) to calculate the transport rate.

As shown in Supplementary Fig. 33b, in the in vitro multilayer symbiotic BBB model, the BBB transport rate (61.1%) of Ang-PAMSe nanomotor with both BBB transport performance and motility was significantly higher than that of the passive nanoparticle PMSe (7.0%). The results were similar to that of the monolayer BBB model, which confirmed that the motility effect of the nanomotor and the transport function of Ang can help its penetrate the BBB.

In this process, the integrity of the tight junctions of BBB model cells is particularly important. It is

well known that dextran is a polysaccharide composed of branching glucose molecules of different lengths. FITC-dextran (FD-4, 4000 Da) is a polysaccharide composed of fluorescein isothiocyanate coupled to dextran coupling, which can be used to identify markers of BBB leakage (*Curr. Protoc. Neurosci.* 2017, 9.58.1; *Toxicol. Lett.* 2015, 237, 79; *Food Funct.* 2017, 8, 1166). Therefore, we used FD-4 (Glpbio) to assess the integrity of BBB after the material penetrated the BBB. The main method is as follows.

On Day 11, the incubated chambers were transferred to a well plate with fresh medium and 200 μL of a solution containing 1 mg mL^{-1} FD-4/HBSS was added to the chambers and incubated for 2 h. Next, the liquid in the well plate was collected and the fluorescence intensity of FD-4 was measured using a fluorescence spectrophotometer (Ex: 492 nm, Em: 520 nm). After incubation with PMSe and Ang-PAMSe, the fluorescence intensity of FD-4 detected in the transwell lower chamber was not significantly different from that of the control (relative fluorescence intensity was maintained at 764~930 a.u.), indicating that the BBB remained intact after nanomotor penetration (Supplementary Fig. 33c) and the confocal laser microscope (CLSM) images of the BBB (Supplementary Fig. 34) also further confirmed this result.

Supplementary Fig. 33 **a** Schematic diagram of in vitro multilayer symbiotic BBB model preparation. **b** Cellular transport rates of PMSe and Ang-PAMSe in a multilayer symbiotic BBB model. **c** Paracellular permeability of FD-4 after 24 h treatment with PMSe and Ang-PAMSe (The control group was fresh cell culture medium) (n=3). Asterisk (*) denotes statistical significance between bars (* $p < 0.05$) using one-way ANOVA analysis.

Supplementary Fig. 34 CLSM plots and fluorescence quantification of the upper co-cultured BBB and cancer cell layers in an in vitro multilayer symbiotic BBB model after adding fresh medium (control), PMSe and Ang-PAMSe nanomotors ($200 \mu\text{g mL}^{-1}$) and incubating for 24 h (after) (Green, cell membranes stained with DiO; blue, cell nuclei stained with DAPI; Scale bar: 100 μm) (n=3).

In addition, we constructed an in vitro dynamic 3D BBB model in order to simulate the flow environment in which the BBB is located in the in vivo environment (*Nat. Commun.* 2019, 10, 2621; *Nat. Methods* 2016, 13, 152). That is, we used a customized microfluidic chip (Biomimetic Environment On Chip, Height: 375 μm , Width: 1.5 mm, Length: 46 mm, Total volume: 31.2 μL) that can simulate the physiological fluid flow and microcirculation of tangential stress. The structure is composed of two chambers (upper and lower), and the two chamber channels are connected by a microporous polyester membrane (pore size: 1 μm), and a connector to connect the fluid system of the syringe pump into the chip. The in vitro dynamic 3D model established in this experiment consists of bEnd.3 and mouse glioblastoma cells (GL261) in a tubular structure, which takes into account the variable factor of tangential stress between capillary microbleeds and bEnd.2 cells in the cerebral cortex, and also provides the gradient of chemokine ROS/iNOS concentration in the tumor microenvironment, it more closely simulates the physiological fluid flow and BBB microenvironment in vivo under tangential stress.

As shown in Supplementary Fig. 35, bEnd.2 cells were spread over all surfaces in the lower chamber of the chip as basal vascular channels; GL261 cells were inoculated on the porous membrane surface of the upper chamber channels of the chip. The specific steps are as follows:

(1) The bEnd.2 cells (100 μL , 2.3×10^7 cell mL^{-1}) were added to the lower channel through the inlet and the chip was immediately turned over to allow bEnd.2 cells to grow against the side of the porous membrane. 2 days later, the chip was placed on the front side, bEnd.2 cells were inoculated again to spread the cells across the bottom of the channel, and incubation was continued for 2 days to build a vascular-like structure in the lower channel.

(2) GL261 cells (100 μL , 2.3×10^7 cell mL^{-1}) were inoculated into the upper channel of the chip and incubated in the incubator for 3 days.

(3) The above constructed BBB chip was connected to the syringe pump via a connector and the cell culture medium was flowed through the lower channel to acclimatize the BBB chip to flow conditions for 24 h.

(4) The cell culture medium containing the material (cy5-PMSe passive nanoparticles and cy5-Ang-PAMSe nanomotors) was circulated through the lower channel (BBB), and the liquid in the upper channel and the liquid circulating in the lower channel were collected after 24 h. The concentration of the material in the solution was detected, and the apparent permeability (P_{app}) was calculated according to the calculation formula.

$$P_{app} = \frac{V_{upper} \times C_{upper}}{A \times t \times (C_{lower} \times V_{lower} + V_{upper} \times C_{upper}) / (V_{lower} + V_{upper})}$$

V_{upper} and V_{lower} are the volumes of the liquid medium in the upper and lower channels at moment t , respectively; A is area of membrane which is 0.69 cm^2 in our chip model; C_{upper} and C_{lower} are the material concentrations in the liquid media collected in the upper and lower channels, respectively.

After 24 h of circulation of the materials (cy5-PMSe passive nanoparticles and cy5-Ang-PAMSe nanomotors) in liquid media, the P_{app} of the BBB chip was detected and calculated to be 2.6×10^{-8} cm s^{-1} and 15.0×10^{-8} cm s^{-1} . As shown in Supplementary Fig. 36, the ability of Ang-PAMSe nanomotors with both BBB transit function and motility properties to penetrate the vascular system in the BBB channel was 5.7 times that of cy5-PMSe passive nanoparticles.

Supplementary Fig. 35 Schematic diagram of constructing an in vitro BBB model under dynamic conditions. Photographs of the microfluidic chip (**left**), schematic diagram inside the channel (**middle**), with bEnd.2 cells cultured on all surfaces of the lower channel and glioblastoma cells (GL261) inoculated on the porous membrane surface of the upper channel. Schematic diagram of the passage of fluid through the syringe pump (**Right**).

Supplementary Fig. 36 Transcytosis of PMSe and Ang-PAMSe nanomotors, measured by quantifying the ratio of P_{app} of Ang-PAMSe nanomotors versus PMSe passive nanoparticles in BBB chips ($n=3$). Asterisk (*) denotes statistical significance between bars ($*p < 0.05$) using one-way ANOVA analysis.

1-4) Appropriate controls should be showed and clearly described for each performed experiment **Answer:** Thanks so much for the reviewer's kind suggestion.

Based on the reviewer's suggestion, we have marked the sample groups of each experiment in detail, especially the specifics of the blank control designation, and the description of each performed experiment has been improved and highlighted in red. The specific improvements are as follows.

Fig. 3 | Cascade effect of PAMSe/TLND nanomotors on the regulation of cellular metabolism. a Schematic illustration of the cellular metabolism regulated by PAMSe/TLND nanomotors. **b** CLSM images of intracellular ROS (labeled with the ROS fluorescent probe, DCFH-DA) in cancer cells after PAMSe/TLND treatment for 24 h (Scale bar: 100 μ m). **c** Fluorescence intensity (F. I.) of intracellular ROS in cancer cells after different samples treatment for 24 h (n=3). **d** CLSM images of intracellular NO (labeled using the NO fluorescent probe, DAF-FM DA) in cancer cells after PAMSe/TLND nanomotors treatment for 24 h (Scale bar: 100 μ m). **e** Fluorescence quantification of intracellular NO in cancer cells after different samples treatment for 24 h (n=3). **f** The intracellular and extracellular increased content of lactic acid after being treated with different samples (n=3). **g** The ratio of intracellular and extracellular lactic acid increased content after being treated with different samples (n=3). **h** The CLSM images with pH fluorescent probes (with pseudo color) (Scale bar: 20 μ m) and **(i)** corresponding pH values of cancer cells after treatment with different samples for 24 h (n=3). **j** The ATP concentration in cancer cells after incubating with different samples for 24 h (n=3). (The samples in this figure are fresh cell culture medium as control, TLND, PAMSe and PAMSe/TLND) Asterisk (*) denotes statistical significance between bars (* $p < 0.05$) using one-way ANOVA analysis.

Fig. 4 | Assessment of the ability of Ang-PAMSe nanomotors to cross the BBB in vitro, and cellular uptake. **a** Schematic illustration of the classic transwell system based in vitro BBB with ROS/iNOS concentration gradient for evaluating the penetration capability of Ang-PAMSe nanomotor across the endothelial monolayer. **b** CLSM images of the cancer cell layer 24 h after adding the sample (Ang-PMSe, PAMSe and Ang-PAMSe, Scale bar: 100 μ m), and (c) its corresponding BBB transport ratio ($n=3$). **d** Schematic illustration of the in vitro BBB model without ROS/iNOS concentration gradient, and (e) BBB transport ratio after different sample treatments ($n=3$). **f** Schematic representation of inoculation of bEnd.3 and Human umbilical vein endothelial cells (HUVECs) in the upper chamber, respectively, to mimic the differential expression of LRP. **g** BBB or HUVECs transport ratio after Ang-PAMSe treatments after 24 h ($n=3$). **h** Cellular uptake ratio for different samples (Ang-PMSe, PAMSe and Ang-PAMSe) by GL261-Luc cells (transfection of luciferase gene) for 24 h ($n=3$), and (i) its corresponding CLSM images (Green, cell membranes stained with DiO; red, samples were labeled with cy5; blue, cell nuclei stained with hoche333342; Scale bar: 20 μ m), image J was used to perform a co-localization test on the captured pictures to obtain the firework image and Pearson's R value. **j** CLSM images of co-localization of lysosome with Ang-PAMSe and Ang-PAMSe/TLND after 5 min incubation and corresponding fluorescence curves (Green, lysosome stained with lyso-tracker green; red, nanomotors were fluorescently labeled with cy5; blue, cell nuclei stained with hoche333342; Scale bar: 20 μ m). **k** CLSM images of co-localization of Golgi with Ang-PAMSe and Ang-PAMSe/TLND after 2 h incubation and corresponding fluorescence curves (Green, Golgi was labeled with Golgi tracker green; red, nanomotors were fluorescently labeled with cy5; blue, cell nuclei stained with hoche333342; Scale bar: 20 μ m). **l** CLSM images of co-localization of mitochondrial with Ang-PAMSe and Ang-PAMSe/TLND after 2 h incubation and corresponding fluorescence curves (Green, Mitochondria were labeled by mitochondrial probes; red, samples were fluorescently labeled with cy5; blue, cell nuclei stained with hoche333342; Scale bar: 20 μ m). Asterisk (*) denotes statistical significance between bars (* $p < 0.05$) using one-way ANOVA analysis.

Reviewer #2 (Remarks to the Author): with expertise in glioblastoma

The authors are to be commended for delivering a manuscript covering a very comprehensive, innovative, fresh and elegant nanomedicine-based technology. In essence, the authors developed, described and tested a 'anomotor'drug, able to cross the blood-brain barrier, and concentrating in the tumor areas due to a diffusion according to a chemotactic gradient, to deliver a drug as payload that is to induce immunogenic cell death in glioblastoma cells. Without any doubt this is a fresh and innovative idea and concept with large theoretical potentials for GBM therapy. There are however some major concerns as detailed hereunder.

Answer: We are pleased that the reviewer made positive comments on our manuscript and many thanks for these constructive criticism and comments.

2-1. First and foremost, the authors need to substantially improve the English language: for this, the authors are advised to sit together with a native speaker to explain in detail which messages they exactly want to convey: the grammar, the syntax, the articles used or not used, verb conjugations etc really are important to make the reader understand the message. As it is, the poor language substantially detract from (and sometimes makes it impossible to understand)the original message and research results the authors want to communicate. It would be a pity if all this would be refuted based on communication issues.

Answer: Many thanks for the reviewer's kind suggestion. We specially sought the help of a senior professor in the biomedical field to revise the manuscript word by word, hoping that the revised manuscript could meet the requirements of the journal.

2-2. It might be good to include team members or authors who are familiar with the disease. Glioblastoma is not just a paradigm malignancy in the brain. The major problems to overcome might not be the blood-brain barrier, which often is partially disrupted anyway in this disease, but the widespread infiltration in the normal brain and the cellular heterogeneity.

Answer: We thank the reviewer a lot for these valuable comments.

As stated by the reviewers, glioblastoma (GBM) is a primary brain tumor of extremely high malignancy. Conventional treatments in the clinic include surgical resection, radiotherapy, and pharmacotherapy (typically chemotherapy with temozolomide), but have not produced significant improvements in median survival of patients. Reasons for the lack of effective progression in treatment efficacy may include the presence of the blood-brain barrier (BBB) limiting drug infiltration at the tumor site, the aggressive growth and extensive infiltration of GBM, and extensive inter- and intra-tumor heterogeneity (*J. Neurol Neurosur. Ps.* 2021, 92, 1103; *Adv. Sci.* 2021, 8, 2100978; *Nat. Rev. Clin. Oncol.* 2018, 15, 422.; *Science*, 2008, 321, 1807; *Genes Dev.* 2007, 21, 2683). Among them, the BBB is the primary factor affecting the therapeutic effect of the therapeutic agent and an important barrier that needs to be overcome for the therapeutic agent to enter the brain. In our work, BBB was overcome by means of a combined chemical recognition-microenvironmental response combination, i.e., by modified peptide Angiopep-2 (Ang) that specifically chemically recognizes low-density lipoproteins highly expressed by brain endothelial cells and GBM cells at the BBB site thereby enabling trans-BBB transport. In addition, the presence of ROS/iNOS concentration gradient between BBB and brain tumor tissues was used as a chemical induction to enable chemotactic behavior of nanomotors towards ROS/iNOS, thus synergistically achieving BBB penetration.

As the reviewers pointed out, to address the more important reasons affecting the therapeutic efficacy of GBM (i.e., the aggressive growth and extensive infiltration of GBM, extensive inter- and intra-tumor heterogeneity), we used the chemotactic properties of PAMSe, released NO and combined action with TLND to activate the complete immune cycle in vivo, instead of using traditional chemotherapeutic drugs, and exploited the killing ability of immune cells on tumor cells, which is expected to overcome the therapeutic difficulties caused by tumor heterogeneity. The activated immune cells not only work in the tumor tissue, but also can effectively kill tumor cells infiltrating other parts of the brain and other parts of the body, which is an important reason why the therapeutic agents involved in this paper can prevent tumor metastasis (Fig. 6a-6i can confirm this).

We are very grateful to the reviewers for their reminders, and we have reorganized the ideas of the article according to the reviewers' suggestions, sorting out the problems faced in GBM treatment and focusing on the problems in the treatment process. In order to better explain the above situation, we also cited some research literature in this field and **specifically consulted** some doctors of our city's professional hospitals.

“Exactly those points are not recapitulated by the model chosen (GL261 in mice), which in addition is far more immunogenic than the real human glioblastoma environment that should be tackled. Therefore, it is necessary to at least show comparable results in other models e.g. human xenograft models since none of them alone, is fully able to mimic the complex human situation. GL261 as model for human GBM unfortunately is way too unconvincing for these types of read-out.”

Answer: Thousands of thanks for the valuable question of the reviewer.

The anticancer effect of the nanomotors constructed in this study is based on their combination with chemotactic properties, NO release properties and TLND metabolic regulation to activate the complete immune cycle in vivo. The use of mouse-derived GL261 cells on C57 mice to construct a GBM model containing immune circulation was determined after consulting a large body of literature (*Nat. Nanotechnol.* 2021, 16, 538; *Plos One* 2015, 10, e0134715; *Cancer Immunol. Res.* 2016, 4, 124; *Clin. Cancer Res.* 2014, 20, 5290).

The main methods commonly used to construct brain tumor models are (1) Syngeneic mouse model, derived by transplanting mouse tumor cell lines into strain-matched mice; (2) Patient-derived xenografts model, derived by transplanting human tumor explants into immunodeficient mice, and (3) Genetically engineered mouse tumor model, created by introducing genetic modifications that result in spontaneous tumor development (*Cells* 2021, 10, 265). Among them, the Syngeneic mouse model, which has a more complete immune system, is often chosen as a model for brain tumor construction in immunotherapy (*Front. Oncol.* 2020; *Cancer Discov.* 2018, 8, 1358), while the patient-derived xenografts model, which has only a limited immune component, is mostly used for non-immunotherapy related tumor treatment (*CNS Drugs.* 2020, 34, 127; *Cell Rep.* 2013, 3, 260). Specifically, the GBM models involved in this work are human-derived U-87 MG cells with balb/c nude mice model and mouse-derived GL261 cells with C57BL/6J mice model. However, balb/c nude mice have an innate immune deficiency, as evidenced by a suppressed thymic immune system that is unable to produce mature T cells, as well as a lack of close contact with T cells, and the development of antigen-presenting cells such as B cells in nude mice is severely compromised (*Genet. Res.* 1996, 8, 295; *Curr. Protoc. Immunol. Chapter 15, Unit 15, 21, 2008*). Therefore, orthotopic U-87 MG glioblastoma tumor-bearing balb/c nude mice are unable to provide the complete immune circulatory system required for this experiment. This is ultimately the reason why this work chose to use mouse-derived GL261 cells to construct GBM on C57BL/6J mice, as in many similar studies (*Nat. Nanotechnol.* 2021, 16, 538; *Plos One* 2015, 10, e0134715; *Cancer Immunol. Res.* 2016, 4,

124; *Clin. Cancer Res.* 2014, 20, 5290). We hope to get the understanding and permission of the reviewer.

Further, we also adopted the reviewer's suggestion and supplemented with Mouse xenograft experiments to establish a mouse intracranial GBM xenograft model using human brain astrocytoma cells U-87 MG to evaluate the antitumor activity of Ang-PAMSe/TLND nanomotors *in vivo*. The specific experiments are as follows.

(1) BALB/c nude mice (female, 6-8 weeks old) purchased from Alingfei Biotechnology Co. (Jiangsu, China) and human brain astrocytoma U-87 MG cell lines (glioblastoma; Cat. No. CL-0238) were purchased from Procell Life Science & Technology to establish an intracranial GBM xenograft model. U-87 MG cells (10^6 cells per mouse in 8 μ L of PBS) were injected directionally into the brain of mice anesthetized with 1-5% isoflurane and oxygen (1.8 mm lateral, 0.6 mm anterior to the bregma; 2 mm depth). Mice that were craniotomized consistently but not injected with U-87 MG cells were referred to as the sham-operated group.

(2) The sham-operated group consisted of 3 mice, and GBM-bearing mice were randomly assigned into 5 groups of three mice each, and samples (PBS, TLND, Ang-PMSe/TLND, Ang-PAMSe and Ang-PAMSe/TLND) were administered by tail vein starting on day 11 after implantation, every two days, for a total of five doses. Sections were prepared from whole brain tissue on day 20 and scanned after staining with hematoxylin and eosin (H&E) stain (Fig. R2).

As shown in Fig. R3, the tumors of mice were still suppressed to different degrees after treatment by different groups, among which the GBM of mice had a more obvious suppression effect after treatment by Ang-PAMSe/TLND nanomotor, indicating that Ang-PAMSe/TLND still showed good therapeutic effects in this model, which might be attributed to the synergistic therapeutic effect of NO released from nanomotors with TLND.

Mouse xenograft model

Fig. R2 Schematic illustration of the mouse xenograft experimental (orthotopic U-87 MG glioblastoma tumor-bearing balb/c nude mice) design.

Fig. R3 Histological analyses of brain tumor collected from orthotopic U-87 MG glioblastoma tumor-bearing balb/c nude mice following different samples treatment (Scale bar: 4 mm) and quantitative analysis of the tumor size (n= 3 biologically independent animals per group). Asterisk (*) denotes statistical significance between bars (*p < 0.05) using one-way ANOVA analysis.

“In the manuscript itself, immune checkpoint therapy is mentioned several times as if it were a relevant standard in GBM. Immune checkpoint blockers in GBM at this stage are completely irrelevant and should not be referred to as the golden standard of immunotherapy just because they take that position in non-neuro-oncology settings.”

Answer: Many thanks for the helpful comment from the reviewer.

According to the reviewer's guidance, we learned that the standard treatment for diagnosed GBM includes temozolomide (TMZ) (*Neuro-Oncology* 2019, 21, 730). temozolomide is the most commonly used chemotherapeutic agent for GBM and was treated by intraperitoneal injection of standard dose (SD) temozolomide ($50 \text{ mg kg}^{-1} \times 5 \text{ days}$) in the GL261 mouse glioma model (*Oncotargets Ther.* 2017, 10, 265). Therefore, we also added the intraperitoneal injection of standard dose of TMZ as a control group in the subsequent experiments in treatment.

We also improved the construction method of the GL261 glioblastoma tumor-bearing C57BL/6J mice model. The procedure of inoculating GL261 cells was unchanged, and only the administration was changed to take place on day 11, and the drug was administered every 2 days for 5 times. The whole brains of mice were taken on day 20, and tissue sections and H&E staining were performed to observe and compare the brain tumor sizes (Supplementary Fig. 47). As shown in Supplementary Fig. 48, Ang-PAMSe/TLND had a significant tumor suppressive effect compared to the standard treatment modality of TMZ.

Supplementary Fig. 47 Schematic illustration of the orthotopic GL261 glioblastoma tumor-bearing C57BL/6J mice experimental design.

Supplementary Fig. 48 Histological analyses of brain tumor collected from orthotopic GL261 glioblastoma tumor-bearing C57BL/6J mice following different samples treatment (Scale bar: 4 mm) and quantitative analysis of the tumor size (n= 3 biologically independent animals per group). Asterisk (*) denotes statistical significance between bars (*p < 0.05) using one-way ANOVA analysis.

2-3. In addition to the stepwise efficacy data of the approach, it would be good to generate and show at least some minor toxicity/safety data: liver, lung, neurotox?

Answer: Many thanks for the helpful suggestion of the reviewer.

According to the reviewers' suggestion, we supplemented the H&E staining of each organ in the orthotopic GL261 glioblastoma tumor-bearing C57BL/6J at the end of the treatment. As shown in Supplementary Fig. 45, no obvious damage to major organs (heart, liver, spleen, lung and kidney) was observed after treatment with Ang-PAMSe/TLNI) compared to the sham-operated group.

In addition, to more comprehensively assess the neurotoxicity of each sample, we administered the tail vein to healthy C57BL/6J mice (without tumors) once every 2 days for a total of 5 doses. At the end of administration, cortical tissues were collected separately and stained with H&E to observe the morphological changes in the cerebral cortex to assess the neurotoxicity of the material (*Int. J. Nanomedicine* 2020, 15, 5299). As shown in Supplementary Fig. 46, no significant abnormalities in cell shape, nuclei shrinkage or fracture were observed in the cerebral cortex of mice treated with Ang-PAMSe/TLNI) no significant abnormalities in cell shape, nuclei shrinkage or fracture were observed in the cerebral cortex of mice. All these results showed that the nanomotor did not cause damage to major organs and did not cause neurotoxicity in the cerebral cortex, and its biosafety was good.

Supplementary Fig. 45 Histological analyses of different main organs (heart, liver, spleen, lung, and kidney) collected from orthotopic GL261 glioblastoma tumor-bearing C57 mice following different samples treatment (Scale bar: 500 μ m).

Supplementary Fig. 46 Histological observation of cerebral cortex tissue collected from healthy C57BL/6J mice after treated with PBS, TLND, Ang-PMSe/TLND, Ang-PAMSe and Ang-PAMSe/TLND (Scale bar: 3 mm).

2- 4. The manuscript would benefit from a more rigid structure in which from the beginning the different objectives of the research are clearly and stepwise communicated. Therefore, the authors could use the frame/structure they refer to at the end of the manuscript i.e. the different 'Links'1-7 ?

Answer: Thanks so much for the reviewer's kind suggestion. According to the suggestions of the reviewers, we reorganized the framework/structure of the manuscript and we tried our best to modify our manuscript and made many changes in the revised manuscript, which are marked with red color.

2-5. In fig.3j, the ATP concentration goes down in the experimental condition. In the light of the envisioned ICD, this is a bit harder to understand since ATP could/should (?) be considered as a read-out for ICD as well ?

Answer: Thanks so much for the question of the reviewer.

As stated by the reviewer, after the ICD, the cancer cell releases ATP outside the cell (ATP_{out}) to release the "find-me" signal to the outside world, and this released ATP can promote phagocytosis of apoptotic cells and stimulate the specific anti-tumor immunocidal effect. In other words, the release of extracellular ATP is one of the markers of ICD death in tumor cells (*Cell Death & Dis.* 2020, 11, 1013; *Sci. Bull.* 2017, 62, 1427; *Angew. Chem. Int. Ed.* 2020, 59, 2; *Adv. Mater.* 2021, 33, 2102188).

In general, the amount of total ATP produced by tumor cells (ATP_{total}) is the sum of ATP intracellular (ATP_{in}) and ATP extracellular (ATP_{out}), while the amount of total ATP produced during apoptosis tends to decrease, especially intracellular ATP, which tends to decrease more significantly (*Nat. Commun.* 2021, 12, 1345; *Angew. Chem. Int. Ed.* 2021, 60, 16139). It is the intracellular ATP, not the extracellular ATP content, that is measured in this paper and therefore shows a decreasing trend. The analysis on the results is specified as follows.

The ATP content in tumor cells (ATP_{in}) treated with PAMSe/TLND nanomotors in Fig. 3j had a significant decreasing trend, mainly due to the cascade effect of PAMSe and TLND on the regulation of cellular metabolism (Fig.3a). As the mitochondria-targeted triphenylphosphine-modified lonidamine (TLND) is a metabolic regulator drug that mainly acts on the energy metabolism of cells, by altering the mitochondrial ultrastructure of tumor cells, inhibiting the activity of hexokinase that binds to mitochondria and thus reducing the glycolysis of tumor cells, the ability of cells to produce ATP is significantly reduced, and the purpose of tumor cell inhibition is achieved. PAMSe in turn can be catalyzed by the large amount of ROS produced during TLND-regulated cell metabolism to produce NO with tumor-killing properties, and in summary, the ATP production capacity of tumor cells will be greatly reduced.

Fig. 3 | Cascade effect of PAMSe/TLND nanomotors on the regulation of cellular metabolism. a Schematic illustration of the cellular metabolism regulated by PAMSe/TLND nanomotors. **b** CLSM images of intracellular ROS (labeled with the ROS fluorescent probe, DCFH-DA) in cancer cells after PAMSe/TLND treatment for 24 h (Scale bar: 100 μ m). **c** Fluorescence intensity (F. I.) of intracellular ROS in cancer cells after different samples treatment for 24 h (n=3). **d** CLSM images of intracellular NO (labeled using the NO fluorescent probe, DAF-FM DA) in cancer cells after PAMSe/TLND nanomotors treatment for 24 h (Scale bar: 100 μ m). **e** Fluorescence quantification of intracellular NO in cancer cells after different samples treatment for 24 h (n=3). **f** The intracellular and extracellular increased content of lactic acid after being treated with different samples (n=3). **g** The ratio of intracellular and extracellular lactic acid increased content after being treated with different samples (n=3). **h** The CLSM images with pH fluorescent probes (with pseudo color) (Scale bar: 20 μ m) and **i** corresponding pH values of cancer cells after treatment with different samples for 24 h (n=3). **j** The ATP concentration in cancer cells after incubating with different samples for 24 h (n=3). (The samples in this figure are fresh cell culture medium as control, TLND, PAMSe and PAMSe/TLND) Asterisk (*) denotes statistical significance between bars (*p < 0.05) using one-way ANOVA analysis.

2-6. in fig.5c, it is shown that IV therapy starts at day 5 after tumor inoculation. Even in the GL261 model, often 10 days are required for proper grafting of the tumor cells in the brain. Therefore, apart from the remarks made about the model itself, it is required to perform the same experiment but with a start of therapy at earliest 10 days after tumor inoculation.

Answer: We thank the reviewer a lot for his/her valuable suggestions.

In the experimental protocol for in vivo evaluation of nanomotor treatment efficacy, the initiation of treatment on day 5 after GL261 cells inoculation was determined based on thorough literature research and careful pre-experiments. We found that the number of days to construct the brain tumor model was distributed between 3-10 days, such as after 3 days of inoculation of tumor cells (*J. Control. Release* 2020, 320, 63; *Adv. Mater.* 2019, 31, 1805697), after 5 days of inoculation (*J. Control. Release* 2021, 338, 583; *Sci. Adv.* 2020, 6, eaaz4204), after 6 days of inoculation (*ACS Nano* 2020, 14, 8, 10127.) and

after 7 days of inoculation (*Adv. Funct. Mater.* 2020, 30, 1909369), and also after 10 days of inoculation (*ACS Nano* 2019, 13, 5591), when treatment was started. This difference in timing may be related to the specific number of cells inoculated when constructing the tumor model (Table R1), and the final dosing time is also determined by imaging monitoring of brain tumors or by observing tumor size in brain sections. In this work, the brain tumor size of mice was likewise evaluated after inoculation of tumor cells and before administration of drug treatment. As shown in Fig. 5e, 5 days after inoculation with GL261 cells, there was a significant tumor fluorescence signal in the brain of mice in the modeling group (II)–(VI) compared with the sham-operated group (I), which proved the successful modeling of GBM, and therefore the time of administration was determined as the 5th day after inoculation with tumor cells.

Table R1 Reported GBM model construction parameters

Cell line	Total number of cells/ mouse	Incubation time (day)	Ref.
Luci+ GL261	1.5×10^5	12	Nat. Nanotechnol. 2021, 16, 538.
U87-Luc	5×10^5	10	Adv. Funct. Mater. 2016, 26, 4201.
U-87 MG- 2 mg of minced Luc brain tumor tissue		10	ACS Nano 2018, 12, 11070.
U87	6×10^5	10	ACS Nano 2019, 13, 5591.
U87MG-Luc	5×10^5	7	Adv. Funct. Mater. 2020, 30, 1909369.
GL261	1×10^6	5	This work

Fig. 5 | Assessment of targeting and therapeutic efficacy of Ang-PAMSE/TLND nanomotors in vivo. **a** Chemiluminescence imaging of GBM in C57BL/6 mice before administration and fluorescence images of the brains after injection of different cy5-labeled samples via tail vein injection (Scale bar: 2 cm). **b**

Quantitative analysis of sample accumulation in main organs. Cy5-sample levels were determined by fluorescence spectroscopy and expressed as injected dose per gram of tissue (%ID g⁻¹). **c** Representative CLSM images of brain tumor tissue in mice 24 h after injection of different samples via tail vein and red fluorescence intensity profiles as a function of the distance from a blood vessel in the representative region marked by the white rectangular frames. (Green, blood vessels are stained with CD31; red, samples are labeled with red fluorescence by cy5; blue, cell nuclei stained with DAPI; N: normal brain tissue; T: tumor; Scale bar: 100 μm); (The names corresponding to the samples in a-c are (I) PBS, (II) Ang-PMSe, (III) PAMSe, (IV) Ang-PAMSe). **d** Treatment protocols for orthotopic brain-GBM-tumor-bearing models. Where the red arrow is the time point of the drug administration and the blue arrow represents the time point of bioluminescence imaging. **e** Representative IVIS spectrum images and **(f)** quantified signal intensity (n = 3 biologically independent animals per group). **g** Survival analysis of the mice that loaded with GL261-Luc glioblastoma during treatment process (n=8, biologically independent animals per group). **h** Quantitative analysis of the DCs maturation level in vivo and **(i)** flow cytometry analysis of the expression of CD80 and CD86 on the surface of DCs extracted from the tumor draining lymph nodes of mice after various treatments. For the gating strategy for DCs maturation analysis refer to Supplementary Fig. 49. (n=3 biologically independent experiments). **j** Quantitative analysis of CD3⁺CD8⁺ cytotoxic T cells, and **(k)** representative flow cytometric plots of CD8⁺ T cells in GBM-bearing brain tissue gating on CD3⁺ cells in each group. For the gating strategy for T cells analysis refer to Supplementary Fig. 51 (n=3 biologically independent experiments per group). **l** Quantitative analysis of CD3⁺CD4⁺ T cells. **m** The quantitative analysis of CD3⁺CD4⁺Foxp3⁺ Tregs in the brain tissue of each group according to flow cytometry in each group. **n** Cytokine levels including IL-6 and **(o)** TNF-α in the serum of mice at the end of treatment (n=3). (The names corresponding to the group in e-o are (I) Sham, (II) PBS, (III) TLND, (IV) Ang-PMSe/TLND, (V) Ang-PAMSe and (VI) Ang-PAMSe/TLND). Asterisk (*) denotes statistical significance between bars (*p < 0.05) using one-way ANOVA analysis.

According to the reviewers' suggestion, we still added the experiment of administering the treatment after 10 days of GL261 cell inoculation. We improved the construction method of the GL261 glioblastoma tumor-bearing C57BL/6J mice model. The procedure of inoculating GL261 cells was unchanged, and only the administration was changed to take place on day 11, and the drug was administered every 2 days for 5 times. The whole brains of mice were taken on day 20, and tissue sections and H&E staining were performed to observe and compare the brain tumor sizes (Supplementary Fig. 47). As shown in Supplementary Fig. 48, the tumors were still suppressed to different degrees after treatment by different treatment groups, among which the GBM of mice had the most obvious suppression effect after treatment by Ang-PAMSe/TLND.

Supplementary Fig. 47 Schematic illustration of the orthotopic GL261 glioblastoma tumor-bearing C57BL/6J mice experimental design.

Supplementary Fig. 48 Histological analyses of brain tumor collected from orthotopic GL261 glioblastoma tumor-bearing C57BL/6J mice following different samples treatment (Scale bar: 4 mm) and quantitative analysis of the tumor size (n= 3 biologically independent animals per group). Asterisk (*) denotes statistical significance between bars (* $p < 0.05$) using one-way ANOVA analysis.

Reviewer #3 (Remarks to the Author): with expertise in nanomotors

In this manuscript a nanoparticle is developed which contains arginine and TLND and which is modified with angiopeptide. Arginine is converted under ROS exposure with the enzyme iNOS into NO. Both NO and TLND contribute to an immunological response leading to tumor cell killing. These latter aspects are known and also demonstrated in this paper. The additional claim is that the particles developed are nanomotors and as such improve the therapeutic efficacy. This claim however is not really substantiated and more quantitative validation is required for this paper to be acceptable for publication.

Answer: We are pleased that the referee finds our work of general interest. We are also grateful to have this opportunity to further clarify and substantiate the therapeutic mechanism of nanomotors.

As suggested by the reviewers, we supplemented the quantitative detection of concentration gradients of reactive oxygen species (ROS) and inducible nitric oxide synthase (iNOS) in the environment where the nanomotors are located and the theoretical simulation of the chemotaxis mechanism of the nanomotors; The quality of co-localization pictures of materials in various organelles was improved, and the content of materials in various organelles was also measured quantitatively; Quantitative detection of different materials in various organs of GBM bearing mice to evaluate their biological distribution; Quantitative detection of Angiopeptide-2 content in Ang-PAMSe; Quantitative detection of iNOS concentration in the cell environment set in Fig. 3; Glucan was used to evaluate the integrity of the BBB cell layer constructed in vitro. On this basis, an in vitro double-layer symbiotic BBB model was also constructed to evaluate the BBB permeability of Ang PAMSe nano motor and the BBB integrity before and after penetration. We hope that our reply and amendment can meet the requirements of the reviewer.

3-1. First of all, the authors show that the accumulated release of NO after 96 h in the mimicked tumor cell environment is 4.7 μM . This means that the release per hour is very low and I wonder how this could lead to the kinetic plots observed.

Answer: Thanks so much for the question of the reviewer.

The cumulative NO release curves are plotted in Fig. 2c by setting up independent sample wells corresponding to different time points ($t=0, 1, 3, 6, 9, 12, 24, 36, 72$ and 96 h) instead of sampling multiple times in the same well. For example, at $t=1$ h, the supernatant and cells in the corresponding sample wells were collected to detect the NO content, and at $t=6$ h, another batch of sample wells were collected to detect the cumulative NO release from the nanomotor in the cellular environment in this way. Because NO as a gas has a very short half-life in the cellular environment, it mostly exists as nitrate/nitrite. We use the classical Griess reagent method to detect the formation of nitrate and nitrite of NO in the cellular environment, rather than directly detecting the generated NO gas, thus allowing us to detect the cumulative release of NO over a longer period of time.

In addition, the NO concentration in the normal cellular environment is at the nM level, as reported by the Wink group, and is about 10-30 nM in endothelial cells and vascular smooth muscle cells (*World J. Gastroenterol.* 2006, 12, 4660); In colorectal cancer, the NO level is about 100 nM (*Free Radical Biol. Med.* 2008, 45, 18). It has also been reported that NO may promote tumor growth by stimulating cancer cell progression and enhancing angiogenesis and metastasis when the cellular level of NO is 1-30 nM, while when its level is high enough to induce p53 phosphorylation (>400 nM) and nitrosative stress (>1 μM), NO may exert its anticancer activity through a variety of mechanisms, including upregulation of p53 to stimulate apoptosis, degradation of anti-apoptotic proteasome molecules, cytochrome C release and increased mitochondrial permeability to produce cell necrosis and cytotoxicity, which are important

reasons for the anticancer effect of NO (*Nat. Rev. Cancer* 2006, 6, 521; *Adv. Therap.* 2022, 5, 2100227; *J. Med. Chem.* 2017, 60, 7617). From this perspective, an NO release of 4.7 μM is an amount sufficient for the cells to be apoptotic, which is also comparable to the amount of NO release involved in the killing of cancer cells by NO in much of the current literature. As shown in Table R2, the currently commonly used NO donors include diazeniumdiolates (NONOates), S-nitrosothiols (SNOs), arginine (Arg) and N-nitrosoamine (BNN6) and their release of NO (1.8 μM ~ 17.5 μM) during cancer treatment. Therefore, the amount of NO released by the nanomotors proposed in this work is enough to exert antitumor effect.

Table R2 Summary of the amount of NO released from typical NO release platforms for tumor therapy.

NO donor	Means of controlling NO release	NO production (μM)	Total time	Ref.
Arg	Cancer cells	~1.8	12 h	J. Am. Chem. Soc. 2021, 143, 12025.
Arg	Cancer cells	~2.2	2 h	ACS Nano 2021, 15, 4845.
Arg	Cancer cells	~2.3	12 h	Adv. Sci. 2021, 8, 2002525.
GSNO	US-triggered	~10	5 min	Biomaterials 2020, 230, 119636.
NONOate	GST-responsive	~13, content of nitrate/nitrite	24 h	Adv. Mater. 2018, 30, 1704490.
SNO	X-ray-triggered	~14	24 h	Angew. Chem. Int. Ed. 2015, 54, 14026.
BNN6	808 nm laser-triggered	~17.5	1 h	Acta Biomater. 2019, 100, 365.
PAMSe		~4.7	96 h	This work

Fig. 2 | Characterizations of the structure and movement behavior of PAMSe nanomotors. **a** TEM image of PAMSe nanomotors (Scale bar: 500 nm). **b** Zeta potential and nanoparticle size of PAMSe nanomotors. **c** NO release profiles from PAMSe nanomotors and post-loaded nanomotors (PMSe/A) in different environments, (I) PAMSe nanomotors and (II) PMSe/A nanomotors in cancer cellular environment, (III) PAMSe nanomotors and (IV) PMSe/A nanomotors in dulbecco's modified eagle medium (DMEM) solution (n=3). Representative trajectories of the **(d)** PAMSe and **(e)** PMSe/A nanomotors under cancer cellular environment over 10 s (Supplementary Movies 1-3, samples in a representative experiment (n=10)). **f** Schematic of chemotactic motion of PAMSe nanomotors along the concentration gradient of chemokine. To simulate chemokine concentration gradients (ROS/iNOS), (II) containing bEnd.3 cell and (III) containing GL261 cell lysate agarose gels were loaded onto the Y-shaped chamber; while agarose gels containing bEnd.3 lysate were loaded on the other side of the channel (II) as a control. **g** Fluorescent images (Scale bar: 400 μm) of the different chambers and **(h)** the corresponding fluorescence quantification values at 0, 30, 60, and 90 min, by adding nanomotors to the left side (I) of the Y-shaped device (n=10). Trajectories of **(i)** PAMSe nanomotors, **(j)** PMSe non-nanomotors on chemokine concentration gradients and **(k)** PAMSe nanomotors on a chemokine-free concentration gradient (the video is intercepted for 100 s, n=50) (Supplementary Movie 5). **l** Chemotaxis velocity and **(m)** chemotactic index (ratio of distance to path length) of samples in (i-k) (n = 50). **n** The schematic illustration of three-inlet one-outlet microfluidic device. Confocal laser scanning microscopy (CLSM) images of the microfluidic channels for **(o)** PMSe non-nanomotors or **(p)** PAMSe nanomotors passing through the middle channel and the substrate (lysate of cancer cells)/buffer pass through the

channels on each side (Supplementary Movie 6, 60 s). **q** Fluorescence intensity distribution graph with the normalized fluorescence intensity to a value between 0~1. **r** The schematic illustration and CLSM images of the microfluidic channels for **(s)** PMSe non-nanomotors or **(t)** PAMSe nanomotors passing through the middle channel and the buffer pass through both of the branch channels. **u** Fluorescence intensity distribution graph with the normalized fluorescence intensity to a value between 0~1. Typical the volume flow rate of 0.6 mL h^{-1} through each inlet was used to maintain the interaction time. Asterisk (*) denotes statistical significance between bars (* $p < 0.05$) using one-way ANOVA analysis.

“Secondly, the particles are spherical and as such have no intrinsic motility behavior. The entire motility has thus to be created from the gradient created around the particles. The authors should quantify this gradient. They should for example determine the ROS production in the microfluidic channels and the gradient created by the cell lysate.”

Answer: A lot of thanks for the valuable suggestion of the reviewer.

As the reviewers pointed out, the quantitative detection of concentration gradients of chemoattractants is important for the investigation of the mechanism of motility of chemotactic nanomotors. Again, we thank the reviewers for their guidance here.

The PAMSe nanomotor is based on an endogenous reaction-driven mechanism, whereby the arginine unit of the nanomotor serves as a substrate for NO production catalyzed by ROS and iNOS, which are highly expressed at the tumor site. Due to the specific affinity of the substrate arginine unit for the enzyme ROS/iNOS, the nanomotor can achieve positive chemotaxis of the ROS/iNOS concentration gradient and thus target the tumor site.

Under the guidance of the reviewers, we quantified the concentration gradients of fluid iNOS and ROS using the Y-channel model as an example. Among them, Cy5 labeled PAMSe nanomotors was placed in reservoir (I), while reservoir (II) and (III) contain agarose gel containing lysate of brain endothelial cell (bEnd.3) or GBM cells (GL-261) (Fig. 2f). Specifically, bEnd.3 and GL-261 cells with density of $5 \times 10^6 \text{ cells mL}^{-1}$ were lysed, and the cell contents were extracted, which was then mixed with the equal volume of agarose solution in reservoir (II) and (III), respectively. The mixtures were solidified at room temperature to form agarose gels that containing cell lysates. Over time, ROS/iNOS diffused in the Y-shaped channel to form a chemoattractant concentration gradient.

As shown in Supplementary Fig. 17, the concentrations of ROS (in the case of superoxide anion, $\text{O}_2^{\cdot-}$) and iNOS were detected at five locations from near to far from the substrate. The results showed that the concentration of the chemoattractant ROS/iNOS was higher at the location containing GL261 lysate (location 1), where the concentration of iNOS was about $2.5 \mu\text{M}$ and the concentration of $\text{O}_2^{\cdot-}$ was about 58.2 nM ; the concentration of iNOS at location 5 was about $0.2 \mu\text{M}$ and the concentration of $\text{O}_2^{\cdot-}$ was about 6.8 nM . A concentration gradient was formed by ROS/iNOS diffusion outward along the channel geometry at positions 1 to 5. In addition, the ROS/iNOS concentration at the bEnd.3 lysate (position 1') is low, with a concentration of approximately $0.3 \mu\text{M}$ for iNOS and 12.2 nM for $\text{O}_2^{\cdot-}$. Since iNOS expression is characteristic of immune stimulation induced in a non-calcium-dependent manner, this is not normally present in normal resting cells (*Trends Immunol.* 2015, 36, 161). Therefore, the iNOS content detected in the reservoir (II) may diffuse out of the reservoir (I).

Supplementary Fig. 17 The concentration gradient of iNOS and ROS ($O_2^{\cdot-}$) in the Y-shaped channel. a The cell lysate of bEnd.3 or GL261 cells at initial density of 5×10^6 cells mL^{-1} was mixed with the equal volume of agarose solution in reservoir (II) and (III) respectively to form the network gels. After the agarose had solidified and stabilized for 30 min, the concentrations of (b) iNOS and (c) $O_2^{\cdot-}$ were detected at the corresponding positions, respectively (n=3).

3-2. The authors should calculate the force necessary to drive the motors and indicate if there is sufficient NO present/produced to achieve this kind of force over prolonged times.

Answer: Many thanks to reviewer for the valuable question.

The nanomotor constructed in this study is based on chemotaxis induced by enzyme-substrate specific binding, using high concentration of ROS/iNOS in the tumor microenvironment as a chemoattractant. For the chemotaxis system, the concentration of the chemoattractant is the key to determine whether the chemotaxis of the nanomotor can occur, and we have studied this in detail.

We first demonstrated in review comment 3-1 that the chemoattractant ROS/iNOS in the Y-channel model containing GL261 lysate (position 1) can diffuse outward along the channel geometry forming a concentration gradient from positions 1 to 5 (Supplementary Fig. 17). Next, we evaluated the effect of setting different chemotactic gradients (different initial cell densities) in the Y-channel on nanomotor chemotaxis. We placed GL261 cell lysates of different densities (10^3 , 10^4 , 10^5 and 10^6) to the Y-channel (III) and the concentrations of intracellular iNOS and ROS at the corresponding cell densities were quantified (Supplementary Fig. 19). Meanwhile, we tracked the fluorescence changes of nanomotors within the Y-channel (III) over 90 min (Supplementary Fig. 20).

As shown in Supplementary Fig. 19, the intracellular iNOS concentrations of GL261 cancer cells at densities of 10^3 , 10^4 , 10^5 and 10^6 cell mL^{-1} were approximately 2.0, 4.1, 9.8 and 17.3 μM , and the intracellular intracellular $O_2^{\cdot-}$ were approximately 4.4, 11.6, 36.0 and 63.9 nM, respectively, similar to the concentrations reported in the literature (DOI:10.1002/adma.202206654). When the original density of GL261 cells decreased from 10^6 cell mL^{-1} to 10^3 cell mL^{-1} (**intracellular iNOS concentration decreased from -17.3 μM to 2.0 μM and intracellular $O_2^{\cdot-}$ concentration from -63.9 nM to 4.4 nM**), chemotactic behavior of PAMSe nanomotors was no longer observed within 90 min (Supplementary Fig. 20), probably due to 10^3 cell mL^{-1} GL261 cells did not provide enough chemotactic agent to induce sufficient force chemotactic movement of the nanomotors. Thus, sufficient chemical inducers in the environment, and good NO release properties of the nanomotors are the key factors to achieve nanomotor actuation.

Supplementary Fig. 19 The intracellular concentration of iNOS and O₂^{·-} in GL261 cancer cells with different cell densities.

Supplementary Fig. 20 Representative fluorescence images of PAMSe nanomotors in reservoir (I) designated position of Y-shaped channel at 30min, 60 min and 90 min. Among them, the agarose gels in reservoir (I) and (II) containing equal volume of agarose solution and GL261 or bEnd.3 cells lysate with different cell densities (10³, 10⁴, 10⁵ and 10⁶ cells mL⁻¹).

Further, we also examined the concentrations of iNOS and ROS (represented by O₂^{·-} and H₂O₂) in non-tumor and tumor tissues of tumor model mice to predict whether the nanomotors constructed in this work can sense the high expression of iNOS and ROS in tumor tissues in the *in vivo* environment. As shown in Fig. R4, the concentrations of iNOS in the tumor tissues was detected to be 29.7 μM, the concentration of O₂^{·-} was 98.9 nM, and the concentration of H₂O₂ was 36.2 μM, which were approximate accorded with literature reports (*Anal. Chem.* 2019, 91, 7774-7781; *Adv. Mater.* 2021, 33, e2005562; *J. Am. Chem. Soc.* 2012, 134, 15758-15764). In comparison, iNOS was barely detectable in normal tissue of the mice. And the concentrations of O₂^{·-} in normal tissue of the mice was about 18.3 nM, while the concentration of H₂O₂ was about 2.9 μM, respectively. Compared to the highest concentrations of chemoattractant ROS/iNOS in *in vitro* chemotaxis assays (17.3 μM of iNOS and 63.9 nM of O₂^{·-} in reservoir (I) in the Y-shaped channel, Supplementary Fig. 19), the local concentration provided by the tumor tissue can be considered sufficient to induce directional propulsion of the nanomotors.

Fig. R4 Quantification of (a) iNOS and ROS (represented by (b) O₂·⁻ and (c) H₂O₂) in healthy tissue or tumor tissue of tumor-bearing mice.

Furthermore, in Fig. 2c we monitored the NO release from the nanomotors in the cellular environment for up to 96 h. It is evident that the NO release rate from the nanomotors is increasing significantly for at least the first 36 h and the cumulative release is still increasing over 96 h, which demonstrates the ability of the nanomotors to continuously produce NO. On this basis the corresponding driving force of the nanomotor was estimated using the Stokes equation: $F_d = 6\pi\eta RU$, where η refers to the water viscosity (7×10^{-4} Pa·s at 37 °C), R to the radius of the nanomotor (100 nm), and U to the chemotaxis velocity (*Proc. Natl. Acad. Sci. U.S.A. 2021, 118, e2104481118*). The driving force $F_d = 7.3 \times 10^{-15}$ N was calculated to drive the nanomotor at a cell density of 10^6 cell mL⁻¹. More importantly, we conduct the finite element simulation (COMSOL Multiphysics software) to guide our experiments (Fig. R5). In a typical simulation, the PAMSe colloidal particle was placed at the center of the box. The diffusions module, electrostatics module and creeping flow module are coupled on the sphere surface through an electroosmotic boundary condition. In the simulation, the fluid velocity near the edge of the frame is used as the particle velocity, the speed of colloidal particle is thus estimated to be $5.0 \mu\text{m s}^{-1}$, which is roughly closer to our experimental results ($5.5 \mu\text{m s}^{-1}$), thus further confirming our proposal mechanism (gradient dominated locomotion). Note that the fluid profile obtained from the simulation only provides a qualitative understanding of the propulsion mechanisms, which may quantitatively differ from the experimental one.

Thanks again to the reviewers for their questions, which allowed us to probe more deeply into the mechanisms and conditions of movement of chemotactic-like nanomotors.

Fig. R5 Fluid speed magnitude (color-coded) and flow field lines (black arrows) in the reference frame of a colloidal particle. The movement direction of the PAMSe colloidal particle is represented by white arrow.

3-3. The video s5 shows that all particles have directional motion and not mere Brownian as you would expect from non-motor systems. This indicates that there is an additional hydrodynamic flow process in the system. Interpretation of these data is therefore inconclusive.

Answer: We thank the reviewer a lot for his/her valuable question.

We apologize for the misunderstanding caused to the reviewers due to the unclear image presentation. We showed the corresponding condition settings in the video as a cartoon diagram (Fig. R6a), where the bEnd.3 cell lysate gel, which does not produce ROS/iNOS, was used to simulate a chemotaxis-free environment; and the GL261 cell lysate gel, which produces large amounts of ROS/iNOS, was used to simulate a chemotaxis-producing environment. Based on the above, we described the details of the sample setup at the time of video shooting in Table R3 and Fig. R6a. After adding the sample solution from the reservoir (I), the chemotactic motion behavior of the nanomotor was photographed (Fig. R6b) using a fluorescence microscope at the bifurcation (circles in Fig. R6a) after 30 min of stabilization. We normalized the trajectories of the 50 tracked nanoparticles to the same starting point in order to observe the direction of chemotaxis (Fig. R6c) and to calculate the displacement, velocity and chemotaxis factor of their motion (Fig. R6d and Fig. R6e). To visualize the chemotaxis behavior of the nanomotors in different environments as shown in Supplementary movie 5, we extracted the last frame of the video and depicted the boundary of the Y-channel with a white dashed line.

As shown in Fig. R6b and Fig. R6c, the PAMSe nanomotor mostly moved toward the upper right side of the screen (the direction with the ROS/iNOS concentration gradient) with an average velocity of about $6.7 \mu\text{m s}^{-1}$. In the same environment, the passive nanoparticles PMSe without motility also showed a certain moving displacement, with an average velocity of about $2.4 \mu\text{m s}^{-1}$, which was much smaller than the moving displacement and average velocity of PAMSe nanomotors under the same conditions, but the direction of motion was non-selective toward the two reservoirs on the right. Similarly, when both reservoirs (II) and (III) were placed with bEnd.3 cell lysate gels (concentration gradient without chemical elicitor), PAMSe nanomotors were unable to exhibit significant chemotactic behavior, with an average velocity of about $1.3 \mu\text{m s}^{-1}$. In addition, PAMSe nanomotors+chemokine gradient had the largest chemotactic index (Fig. R6e). All of the above suggested that the appearance of the more obvious displacement in Fig. R6b and Fig. R6c was due to the chemotactic motion behavior of the nanomotor. The possible reason for the PMSe nanoparticles+chemokine gradient and PAMSe nanomotors+no-chemokine gradient showing some displacement was due to the flow induced when the fluid was added, causing the passive particles to gain velocity with the fluid and the direction of motion was non-selective towards the two reservoirs on the right.

Table R3 Specific information on the different condition settings in Supplementary movie 5.

Name of the sample	Reservoir (I)	Reservoir (II)	Reservoir (III)
PAMSe nanomotors+chemokine gradient	PAMSe nanomotors	bEnd.3 cell lysate gel	GL261 cell lysate gel
PMSe nanoparticles +chemokine gradient	PMSe nanoparticles	bEnd.3 cell lysate gel	GL261 cell lysate gel
PAMSe nanomotors+no-chemokine gradient	PAMSe nanomotors	bEnd.3 cell lysate gel	bEnd.3 cell lysate gel

Fig. R6 Characterizations movement behavior of PAMSe nanomotors. **a** Cartoon diagram of the corresponding condition settings in supplementary movie 5. **b** Screenshot corresponding to the supplementary movie 5, the white dotted line is the outline of the Y channel. Motion analysis of the samples in supplementary movie 5, including (c) motion trajectory, (d) motion velocity and (e) chemotactic index (the video is intercepted for 100 s from Movie S5, n=50).

3-4. The reason why the arginine loaded nanomotors behave worse than the covalent ones is not explained. Is this a result of loading efficiency? What is the loading capacity of the covalent arginine systems? Quantitative information is missing.

Answer: Thousands of thanks for the valuable question of the reviewer.

Theoretically, the NO-driven nanomotor constructed in this work, in which the active group (guanidine group in L-arginine) is introduced into the nanomotor substrate by in-situ covalent bonding method, can have more effective, stable and durable NO release performance than the NO-driven nanomotors which L-arginine was loaded by post-loading method. In general, the amount of L-arginine can be detected by using naphthol-diacetyl chromogenic quantification method (*J. Am. Chem. Soc.* 2021, 143, 12025; *Adv. Sci.* 2021, 8, 2002525), which is more suitable for the detection of free L-arginine. While during the synthesis process of PAMSe nanomotor, L-arginine was first modified to obtain arginine-methacryloylamide (Arg-Me) (Supplementary Fig. 1) and then used to covalently bind with diselenide cross-linker (Supplementary Fig. 6). Thus, PAMSe nanomotor does not have free arginine, only active guanidine. Therefore, it is difficult to directly compare the differences in L-arginine content between PAMSe and PMSe/A.

In this work, the difference of the active groups between PAMSe and PMSe/A was reflected the difference of NO release ability, motility and cell uptake ability. We first examined the NO release performance of the two types of nanomotors in the tumor cell environment. As shown in Fig. 2c, the PAMSe nanomotor was able to release NO continuously for at least 96 h in the tumor cell environment with a release of 4.7 μM , while the PMSe/A nanomotor was basically released as early as 24 h, and the cumulative concentration was only 1.6 μM . Higher NO release also endowed PAMSe nanomotor to exhibit better motion ability, and we tracked the trajectory of PAMSe and PMSe/A nanomotors in the tumor cell environment (with high concentration of ROS/iNOS) (Fig. 2d,e and Supplementary Fig.14, Supplementary Movie 1-3). Upon analysis of the motion trajectories of both PAMSe and PMS/A nanomotor, the average motion velocity of PAMSe nanomotors in the tumor cell environment was $5.2 \pm 1.0 \mu\text{m s}^{-1}$, which was about 1.5 times higher than that of PMS/A nanomotor ($2.1 \pm 0.6 \mu\text{m s}^{-1}$). In addition, the cellular uptake (Supplementary Fig. 15) and real-time live cell imaging (Supplementary Fig. 16 and Supplementary Movie 4) results demonstrated that the PAMSe nanomotor was more efficiently taken up by cells after 1 h incubation with cells (about 9 times more than the PMSe/A nanomotor).

All the above results indicate that L-arginine can maintain more active groups when it is added in the synthesis of nanomotor by covalent bonding, thus showing better NO release ability, movement speed and more uptake by cells.

Supplementary Fig. 1. Synthesis process of arginine-methacryloylamide (Arg-Me).

Supplementary Fig. 6. Synthesis route of PAMSe nanomotors.

Supplementary Fig. 14 Trajectory of PAMSe nanomotors and PMSe/A nanomotors in DMEM solution (Supplementary Movie 3, 10 s.). Samples in a representative experiment, $n = 10$.

Supplementary Fig. 15 Representative CLSM images and uptake ratio of cancer cells after uptaking the cy5- PMSe/A or cy5-PAMSe nanomotors (Green, cell membranes stained with Dio; red, nanomotors were fluorescently labeled with cy5; blue, cell nuclei stained with hochest33342; Scale bar: 20 μm , n=3). Asterisk (*) denotes statistical significance between bars (* $p < 0.05$) using one-way ANOVA analysis.

Supplementary Fig. 16 Real-time live-cell images (Supplementary Movie 4) of cancer cell treatment with PMSe/A or PAMSe nanomotors and the corresponding fluorescence intensity (F. I.) of nanomotors at t=1 h. (Bright, cells; red, nanomotors were fluorescently labeled with cy5; scale bar: 100 μm). Asterisk (*) denotes statistical significance between bars (* $p < 0.05$) using one-way ANOVA analysis.

Fig. 2 | Characterizations of the structure and movement behavior of PAMSe nanomotors. **a** TEM image of PAMSe nanomotors (Scale bar: 500 nm). **b** Zeta potential and nanoparticle size of PAMSe nanomotors. **c** NO release profiles from PAMSe nanomotors and post-loaded nanomotors (PMS/A) in different environments, (I) PAMSe nanomotors and (II) PMS/A nanomotors in cancer cellular environment, (III) PAMSe nanomotors and (IV) PMS/A nanomotors in dulbecco's modified eagle medium (DMEM) solution (n=3). Representative trajectories of the **(d)** PAMSe and **(e)** PMS/A nanomotors under cancer cellular environment over 10 s (Supplementary Movies 1-3, samples in a representative experiment (n=10)). **f** Schematic of chemotactic motion of PAMSe nanomotors along the concentration gradient of chemokine. To simulate chemokine concentration gradients (ROS/iNOS), (II) containing bEnd.3 cell and (III) containing GL261 cell lysate agarose gels were loaded onto the Y-shaped chamber; while agarose gels containing bEnd.3 lysate were loaded on the other side of the channel (II) as a control. **g** Fluorescent images (Scale bar: 400 µm) of the different chambers and **(h)** the corresponding fluorescence quantification values at 0, 30, 60, and 90 min, by adding nanomotors to the left side (I) of the Y-shaped device (n=10). Trajectories of **(i)** PAMSe nanomotors, **(j)** PMS non-nanomotors on chemokine concentration gradients and **(k)** PAMSe nanomotors on a chemokine-free concentration gradient (the video is intercepted for 100 s, n=50) (Supplementary Movie 5). **l** Chemotaxis velocity and **(m)** chemotactic index (ratio of distance to path length) of samples in (i-k) (n = 50). **n** The schematic illustration of three-inlet one-outlet microfluidic device. Confocal laser scanning microscopy (CLSM) images of the microfluidic channels for **(o)** PMS non-nanomotors or **(p)** PAMSe nanomotors passing through the middle channel and the substrate (lysate of cancer cells)/buffer pass through the

channels on each side (Supplementary Movie 6, 60 s). **q** Fluorescence intensity distribution graph with the normalized fluorescence intensity to a value between 0-1. **r** The schematic illustration and CLSM images of the microfluidic channels for **(s)** PMSe non-nanomotors or **(t)** PAMSe nanomotors passing through the middle channel and the buffer pass through both of the branch channels. **u** Fluorescence intensity distribution graph with the normalized fluorescence intensity to a value between 0-1. Typical the volume flow rate of 0.6 mL h^{-1} through each inlet was used to maintain the interaction time. Asterisk (*) denotes statistical significance between bars (* $p < 0.05$) using one-way ANOVA analysis.

3-5. The quality of the colocalization studies in fig 4l-n is poor and should be improved to allow any conclusions to be drawn.

Answer: Thanks a lot for the helpful comments from the reviewer.

Based on the reviewer's suggestion, we improved the quality of organelle co-localization images to better observe the co-localization of different samples with organelles. It is clearly observed from Fig. 4j-4l that both Ang-PAMSe with motility function only and Ang-PAMSe/TLND with both motility and mitochondrial targeting functions exhibit distribution in lysosomes and other regions of the cell; Ang-PAMSe has a higher overlap with Golgi apparatus, while Ang-PAMSe/TLND has a significantly higher overlap with mitochondria, which implies that Ang-PAMSe/TLND has a better mitochondrial targeting function, which should be attributed to the mitochondrial targeting motif triphenylphosphine (TPP) in TLND.

In addition, we further used organelle extraction kits, such as lysosome extraction kit (BB-3603, BestBio, shanghai, China), Golgi extraction kit (BB-3604, BestBio, shanghai, China) and mitochondrial extraction kit (SM0020, mitochondrial extraction kit, Beijing Solarbio Science & Technology Co., Ltd.) to extract the corresponding organelles to quantify the content of materials in each organelle. GL261 cells (cell density of $1 \times 10^6 \text{ cells mL}^{-1}$) were treated with Cy5-labeled different samples (Cy5-Ang-PAMSe and Cy5-Ang-PAMSe/TLND, final concentration of $200 \mu\text{g mL}^{-1}$) for 24 h. Next, organelles were extracted and detected using a fluorescence spectrophotometer (Ex: 650 nm, Em: 667 nm) The content of material in each organelle was measured. Uptake rate of material in the organelle = amount of material in the organelle / amount of material taken up by the cell $\times 100\%$. As shown in Fig. R7, the lysosomal uptake efficiency of both Ang-PAMSe and Ang-PAMSe/TLND is very low (up to only about 1.3%); Ang-PAMSe is more located in the Golgi (Golgi uptake rate of 7.3%), while Ang-PAMSe/TLND is more distributed in the mitochondria (mitochondrial uptake rate of about 8.1%). This again quantitatively demonstrates that Ang-PAMSe/TLND has a better mitochondrial targeting ability.

Fig. 4 | Assessment of the ability of Ang-PAMSe nanomotors to cross the BBB in vitro, and cellular uptake. **a** Schematic illustration of the classic transwell system based in vitro BBB with ROS/iNOS concentration gradient for evaluating the penetration capability of Ang-PAMSe nanomotor across the endothelial monolayer. **b** CLSM images of the cancer cell layer 24 h after adding the sample (Ang-PAMSe, PAMSe and Ang-PAMSe, Scale bar: 100 μ m), and **(c)** its corresponding BBB transport ratio ($n=3$). **d** Schematic illustration of the in vitro BBB model without ROS/iNOS concentration gradient, and **(e)** BBB transport ratio after different sample treatments ($n=3$). **f** Schematic representation of inoculation of bEnd.3 and Human umbilical vein endothelial cells (HUVECs) in the upper chamber, respectively, to mimic the differential expression of LRP. **g** BBB or HUVECs transport ratio after Ang-PAMSe treatments after 24 h ($n=3$). **h** Cellular uptake ratio for different samples (Ang-PAMSe, PAMSe and Ang-PAMSe) by GL261-Luc cells (transfection of luciferase gene) for 24 h ($n=3$), and **(i)** its corresponding CLSM images (Green, cell membranes stained with DiO; red, samples were labeled with cy5; blue, cell nuclei stained with hoche333342; Scale bar: 20 μ m), image J was used to perform a co-localization test on the captured pictures to obtain the firework image and Pearson's R value. **j** CLSM images of co-localization of lysosome with Ang-PAMSe and Ang-PAMSe/TLND after 5 min incubation and corresponding fluorescence curves (Green, lysosome stained with lyso-tracker green; red, nanomotors were fluorescently labeled with cy5; blue, cell nuclei stained with hoche333342; Scale bar: 20 μ m). **k** CLSM images of co-localization of Golgi with Ang-PAMSe and Ang-PAMSe/TLND after 2 h incubation and corresponding fluorescence curves (Green, Golgi was labeled with Golgi tracker green; red, nanomotors were fluorescently labeled with cy5; blue, cell nuclei stained with hoche333342; Scale bar: 20 μ m). **l** CLSM images of co-localization of mitochondrial with Ang-PAMSe and Ang-PAMSe/TLND after 2 h incubation and corresponding fluorescence curves (Green, Mitochondria were labeled by mitochondrial probes; red, samples were fluorescently labeled with cy5; blue, cell nuclei stained with hoche333342; Scale bar: 20 μ m). Asterisk (*) denotes statistical significance between bars (* $p < 0.05$) using one-way ANOVA analysis.

Fig. R7. Quantitative detection of the uptake efficiency of samples in various organelles (lysosome, Golgi and mitochondrion).

3-6. Have the authors established that the BBB is intact in their mouse model? If not, then this is merely an EPR effect that allows particles to accumulate.

Answer: Many thanks for the valuable question of the reviewer.

The integrity of the BBB is disrupted to varying degrees during the progression of gliomas (*Nat. Rev. Clin. Oncol.* 2021, 18, 696). A number of assays including magnetic resonance imaging and positron emission tomography (PET), Evans blue staining, and histopathological methods have been performed to study BBB integrity at different stages of progression in a GBM-bearing mouse model of orthotopic GL261 GBM, showing that GL261 tumors display varying degrees of BBB disruption during progression (*J. Neurooncol* 2014, 119, 297; *Proc. Natl. Acad. Sci. U.S.A.* 2013, 110, 832). For the early orthotopic GBM model, GL261 cells inoculated for 7 days showed only slight disruption of the BBB, preserving the presence of almost intact BBB (*Nat. Rev. Mater.* 2016, 1, 16014). For late stage orthotopic GBM models, it is believed that the tight junctions of the BBB are disrupted after 14 days of GL261 cell inoculation, but even under this condition, the drug concentration at the tumor site relying only on the EPR effect of the tumor is still insufficient to reach therapeutic levels (*Chem. Soc. Rev.* 2019, 48, 2967). The time to construct the GBM model in this work is 5-10 days, which is an early stage, so efficient penetration of the BBB remains one of the key and primary factors limiting effective treatment (*Adv. Funct. Mater.* 2020, 30, 1909369).

The Ang-PAMSe nanomotor constructed in this work is a combined targeting strategy to achieve penetration of the BBB and thus complete tumor infiltration. First, endothelial cells and GBM cells at the BBB site are recognized by the modified peptide Angiopep-2 (Ang), in which case the targeting ligand (Ang) binds to the endogenously expressed BBB/BTB receptor low-density lipoprotein receptor-related protein 1 (LRP1), triggering the active transport function. Furthermore, based on an endogenous reaction-driven mechanism, the arginine unit of the nanomotor as a substrate can be reacted with ROS and iNOS highly expressed at the tumor site to produce L-citrulline and nitric oxide (NO). Due to the specific affinity of the substrate arginine unit for the enzyme ROS/iNOS, the nanomotor can achieve positive chemotaxis of the ROS/iNOS concentration gradient to target the tumor site. Thus, the nanomotor constructed in this work achieves deep penetration into tumor tissues through combined chemical ligand/receptor recognition transport and chemotactic movement.

Based on this, we evaluated the distribution of different samples in GBM tumor tissues. As shown in Fig. 5a and Supplementary Fig. 43, the distribution of the materials (red fluorescence) in the GBM

tumor tissues of the GBM-bearing mice after 24 h by tail vein injection of (I) PBS, and cy5 labeled (II) cy5-Ang-PMSe, (III) cy5-PAMSe, and (IV) cy5-Ang-PAMSe, respectively. Near-infrared fluorescence imaging (IVIS Lumina III) images of mice showed that after 24 h of Ang-PAMSe nanomotor injection, significant red fluorescence was observed in the mouse brain tumor. In addition, quantitation of fluorescence in the brain and other organs showed that the brain has a maximum accumulation of 39% of the injected dose (ID) for Ang-PAMSe (Fig. 5b), which confirmed that Ang-PAMSe nanomotors showed better BBB penetration and GBM tissue accumulation compared to the three control treatments. Subsequently, we evaluated the penetration of Ang-PAMSe in solid tumors (Fig. 5c). After 24 h of intravenous injection, the red fluorescence signal of Ang-PAMSe nanomotors was almost all over the tumor (about 90.2 a.u. fluorescence intensity was still detected at about 500 μm from the tumor edge). This can be attributed to the fact that Ang-PAMSe nanomotors have good tumor chemotaxis, which exhibit better tissue permeability. In contrast, because of the absence of autonomous motility, the fluorescence signal of Ang-PMSe at about 200 μm from the tumor edge was very weak, and the PAMSe nanomotors with only motility could penetrate about 350 μm from the tumor edge with a fluorescence intensity of 62.8 a.u.. These results suggested that the Ang-derived targeting function and the autonomous motility of nanomotors played a key synergistic role in achieving the effective penetration of Ang-PAMSe nanomotors into the BBB and the GBM efficiently.

Supplementary Fig. 43 Total radiant efficiency of different cy5-labeled samples (I) PBS, (II) Ang-PMSe, (III) PAMSe, (IV) Ang-PAMSe in GBM (n=3). Asterisk (*) denotes statistical significance between bars (*p < 0.05) using one-way ANOVA analysis.

Fig. 5 | Assessment of targeting and therapeutic efficacy of Ang-PAMSe/TLND nanomotors in vivo. **a** Chemiluminescence imaging of GBM in C57BL/6 mice before administration and fluorescence images of the brains after injection of different cy5-labeled samples via tail vein injection (Scale bar: 2 cm). **b**

Quantitative analysis of sample accumulation in main organs. Cy5-sample levels were determined by fluorescence spectroscopy and expressed as injected dose per gram of tissue (%ID g⁻¹). **c** Representative CLSM images of brain tumor tissue in mice 24 h after injection of different samples via tail vein and red fluorescence intensity profiles as a function of the distance from a blood vessel in the representative region marked by the white rectangular frames. (Green, blood vessels are stained with CD31; red, samples are labeled with red fluorescence by cy5; blue, cell nuclei stained with DAPI; N: normal brain tissue; T: tumor; Scale bar: 100 μm); (The names corresponding to the samples in a-c are (I) PBS, (II) Ang-PMSe, (III) PAMSe, (IV) Ang-PAMSe). **d** Treatment protocols for orthotopic brain-GBM-tumor-bearing models. Where the red arrow is the time point of the drug administration and the blue arrow represents the time point of bioluminescence imaging. **e** Representative IVIS spectrum images and **(f)** quantified signal intensity (n = 3 biologically independent animals per group). **g** Survival analysis of the mice that loaded with GL261-Luc glioblastoma during treatment process (n=8, biologically independent animals per group). **h** Quantitative analysis of the DCs maturation level in vivo and **(i)** flow cytometry analysis of the expression of CD80 and CD86 on the surface of DCs extracted from the tumor draining lymph nodes of mice after various treatments. For the gating strategy for DCs maturation analysis refer to Supplementary Fig. 49. (n=3 biologically independent experiments). **j** Quantitative analysis of CD3⁺CD8⁺ cytotoxic T cells, and **(k)** representative flow cytometric plots of CD8⁺ T cells in GBM-bearing brain tissue gating on CD3⁺ cells in each group. For the gating strategy for T cells analysis refer to Supplementary Fig. 51 (n=3 biologically independent experiments per group). **l** Quantitative analysis of CD3⁺CD4⁺ T cells. **m** The quantitative analysis of CD3⁺CD4⁺Foxp3⁺ Tregs in the brain tissue of each group according to flow cytometry in each group. **n** Cytokine levels including IL-6 and **(o)** TNF-α in the serum of mice at the end of treatment (n=3). (The names corresponding to the group in e-o are (I) Sham, (II) PBS, (III) TLND, (IV) Ang-PMSe/TLND, (V) Ang-PAMSe and (VI) Ang-PAMSe/TLND). Asterisk (*) denotes statistical significance between bars (*p < 0.05) using one-way ANOVA analysis.

3-7. Fig S38 what is the physiological explanation that the particles without arginine end up in the lungs?
How does the accumulation in the brain compare to the other organs?

Answer: Thanks a lot for the helpful question from the reviewer.

In order to more accurately assess the distribution of the different samples in the various organs of the GBM tumor-bearing mice, we performed a quantitative assay of the material biodistribution in the following steps:

(1) The tumor-bearing mice were randomly divided into 4 groups, and Cy5-Ang-PMSe (chemical recognition targeting), Cy5-PAMSe (chemokine targeting) and Cy5-Ang-PAMSe (Table R4) were injected via tail vein respectively, where PBS was injected into tail vein as blank control.

(2) After 24 h, the major organs of the mice including heart, liver, spleen, lung, kidney, brain and whole brain were collected, the surfaces were rinsed with PBS, blotted with filter paper and weighed. The tissues were then cut into grinding tubes and added to RIPA lysate at a ratio of 1 mL:100 mg tissue, and the tissues were ground with preset parameters on a tissue grinder to prepare a homogenate, centrifuged at 3000 rpm for 5 min, the supernatant was removed and fixed to 2 mL with RIPA lysate, and the fluorescence intensity of the material was measured using a fluorescence spectrophotometer (Ex: 650 nm, Em: 667 nm). The amount of material in each tissue was calculated based on the fluorescence intensity-concentration standard curve. The percent injected dose rate (%ID) was then calculated according to the following formula: ID% = (mass of sample in organ tissue/total mass of injected sample) × 100%. As shown in Fig. 5b, compared to other samples, Ang-PAMSe accumulated significantly in the

brain of mice. Moreover, quantitation of fluorescence in the brain and other organs showed that the brain has a maximum accumulation of 39% of the injected dose (ID) for Ang-PAMSe (Fig. 5b), which confirmed that Ang-PAMSe nanomotors showed better BBB penetration and GBM tissue accumulation compared to the three control treatments.

Table R4 Targeting performance of samples

Sample	Chemical recognition targeting	Chemokine targeting
Ang-PMSe	- .1	×
PAMSe	×	- .1
Ang-PAMSe	- .1	- .1

3-8. What is the reason for the 40% death rate in the mice in VI, as the tumor has been depleted?

Answer: We thank the reviewer a lot for this insightful question.

We summarized the median survival, death rate and survival time of mice regarding GBM treatment published in recent years, and it can be seen from Table R5 that improving median survival, reducing death rate and prolonging survival time of mice are more important indicators to assess the effectiveness of GBM treatment. Fig. 5e and Fig. 5f show the GBM imaging images of mice during treatment and the average of GBM fluorescence intensity.

In this work, the imaging data at the end of the treatment showed the average total fluorescence flux (Total Flux) for each group of samples. The VI group (Ang-PAMSe/TLND) noted by the reviewer was about 3.9×10^7 [(p/s)/(μ W/cm²)], which did not disappear completely, but has been significantly lower than the PBS group (283.3×10^7 [(p/s)/(μ W/cm²)]). In addition, the average fluorescence flux was calculated by averaging the fluorescence intensities of three parallel samples, some of which had 0 fluorescence intensity and some of which still had the fluorescence intensity of tumor tissue, and this part of the residual tumor continued to grow which might be the cause of lethality, while those with 0 or near 0 fluorescence intensity could survive for a long time because they had developed a good immune memory in vivo.

Table R5 The median survival, death and survival time of GBM bearing mice after treatment with different therapeutic agents.

Therapeutic agent	Median survival time	Death rate	Survival time	Ref
STICK-NP@ VCR	21.3 d	66.7%	50 d	Adv. Mater. 2020, 32, 1903759.
HALF-cRGD	35 d	100%	40 d	Adv. Funct. Mater. 2020, 30, 1907077.
Ang-3I-NM@si(PLK1+VEGFR2)	36 d	100%	52 d	Adv. Mater. 2019, 31, 1903277.
Ang-NCss(siPLK1)	42 d	100%	52 d	Adv. Mater. 2020, 2000416.
Ang-RBCm-CA/siPLK1	43 d	100%	52 d	Nano Lett. 2020, 20,

hM@HLPC	48 d	100%	68 d	1637. Nat. Commun. 2022, 13,4214.
n(Nimo)	49 d	50%	60 d	Adv. Mater. 2019, 31, 1805697.
ANCSS(Cas9/sgPLK1)	55 d	100%	68 d	Sci. Adv. 2022, 8, eabm8011.
DOX/Ce6-LiPTD NPs + US	57 d	99.7 %	60 d	Adv. Sci. 2022, 9, 2203894.
PTX-CL/NEs	61 d	75%	120 d	Nat. Nanotech. 2017, 12, 692.
Ang-PAMSe/TLND	71 d	62.5%	120 d	This work

Other technical questions/comments

3-9. The labeling with angiopeptide is not quantified.

Answer: Thousands of thanks for the valuable question of the reviewer.

Angiopep-2 is a peptide consisting of 12 amino acids (Angiopep-2:NH₂-Thr-Phe-Phe-Tyr-Gly-Gly-Ser-Arg-Gly-Lys-Arg-Asn-Asn-Phe-Lys-Thr-Glu-Glu-Tyr-COOH) with a UV absorption peak, we detected Angiopep-2 at concentrations 0-200 µg mL⁻¹ using a functional UV-Vis spectrophotometer (UH5300) and made a standard curve.

Angiopep-2 is a peptide consisting of 12 amino acids (Angiopep-2:NH₂-Thr-Phe-Phe-Tyr-Gly-Gly-Ser-Arg-Gly-Lys-Arg-Asn-Asn-Phe-Lys-Thr-Glu-Glu-Tyr-COOH) with a UV absorption peak at 214 nm. We detected Angiopep-2 at concentrations of 0-200 µg mL⁻¹ using a functional UV-Vis spectrophotometer (UH5300) and made a standard curve (Supplementary Fig. 28). From the standard concentration curve of Angiopep-2 ($Y=0.01106X+0.11075$), the content of Angiopep-2 in Ang-PAMSe was calculated to be about 12.0±0.05 %.

Supplementary Fig. 28 Standard concentration curve of Angiopep-2 (UV absorption wavelength at 214 nm).

3-10. What are the iNOS levels of the cancer cells used in fig 3?

Answer: We thank the reviewer a lot for his/her valuable question.

As suggested by the reviewers, we examined the concentration of iNOS in the cellular environment.

Fig. 3a-3e show the NO and ROS content assay, Fig. 3f-3g show the content of lactate inside and outside the cell, Fig. 3h,i show the intracellular pH assay experiment, and Fig. 3j shows the cellular ATP concentration assay. We repeated the experimental steps in Fig. 3, respectively, collected the supernatant and cells in the corresponding well plates, and followed the procedure for using the human inducible nitric oxide synthase (iNOS) ELISA kit (MEIMIAN, MM-1514H1, 96T) to obtain the concentration of iNOS in the cell suspension. We first produced a standard curve of iNOS concentration versus absorbance (Fig. R8a). Based on this, the iNOS concentrations involved in Fig. 3 were $5.1 \mu\text{M}$ (3a-3e), $3.4 \mu\text{M}$ (3f-3i) and $7.1 \mu\text{M}$ (3j), respectively (Fig. R8b). Since iNOS is only expressed in inflammatory or cancer cells, this significantly increased iNOS concentration may provide the motivation for nanomotor chemotaxis.

Fig. R8 **a** Standard curves of iNOS concentration versus absorbance and **(b)** corresponding iNOS concentration in the cellular environment in Fig.3.

3-11. Video S2: it is unclear what the 10 frames are and what we are looking at. A better explanation should be provided.

Answer: We thank the reviewer a lot for his/her valuable suggestions.

According to the reviewer's suggestion we added an explanation of the content of Movie S2. In order to compare the advantages of PAMSe nanomotors synthesized using the covalent grafting method in this paper, we set up nanomotors prepared by the post-loaded arginine method (PMSe/A) separately as a control to evaluate the motor performance of both types of nanomotors in the cancer cell environment in terms of parameters such as motor speed and MSD. Ten nanomotors of each type were selected as typical representatives and their trajectories were tracked over 10 seconds (numbers 1-10 represent the number of each motor).

Fig. R9 Screenshot of the last frame of the video S2. The motion track of nanomotor (PMSe/A) prepared by the method of post loading L-arginine in the cell environment.

3-12. From video S4 the cell uptake efficiency is not clear.

Answer: Thanks a lot for the reviewer's questions.

Movie S4 shows the Merge field of the bright field channel and the cy5 fluorescence channel. To visualize the difference in cellular uptake more, we extracted the cy5 fluorescence channel (red color) of the confocal microscope capturing pictures at t=1 h separately and analyzed its fluorescence intensity. As shown in Fig.R10, cancer cells treated with PAMSe nanomotors had stronger red fluorescence with approximately 4.5 times higher fluorescence intensity than the PMSe/A nanomotor group, suggesting a facilitation of the motion effect of nanomotors on the increased cellular uptake efficiency.

Further, to more accurately assess the uptake efficiency of cancer cells on PMSe/A and PAMSe nanomotors, we supplemented the assay with a quantitative assay of cellular uptake. Cancer cells (5×10^4 cell mL^{-1}) were treated with 1 mL of cy5 fluorescently labeled samples (cy5-PMSe/A and cy5-PAMSe nanomotors) at a concentration of $200 \mu\text{g mL}^{-1}$ for 1 h. Then, the supernatant, cell wash solution, and cell lysate from the well plates after lysing the cells with cell lysis solution were collected separately and detected using a fluorescence spectrophotometer (Ex: 650 nm, Em: 667 nm) for the determination of each sample. Where the amount of material detected in the cell lysate is the uptake value, and the amount of material contained in the mixture of supernatant, cell wash solution and cell lysate is recorded as the total value. Cell uptake efficiency (%) = uptake value / total value \times 100%. The results of Supplementary Fig. 15 showed that the cellular uptake efficiency of the PAMSe nanomotor group was 15.2%, which was about 4.1 times higher than that of the PMSe/A nanomotor group (3.7%).

In summary, we have reached the consistent conclusion that cancer cells have higher uptake efficiency of PAMSe nanomotors compared to PMSe/A nanomotors by real-time imaging of live cells, CLSM picture taking of cell uptake and quantitative detection of cell uptake (Supplementary Fig. S18), respectively.

Fig. R10 Real-time imaging of live cells after nanomotor (PMSe/A or PAMSe) treatment at t=1 h and the corresponding fluorescence intensity (F. I.). (Bright, cells; red, nanomotors were fluorescently labeled with cy5; scale bar: 100 μm).

Supplementary Fig. 15 Representative CLSM images and uptake ratio of cancer cells after uptaking the cy5- PMSe/A or cy5-PAMSe nanomotors (Green, cell membranes stained with Dio; red, nanomotors were fluorescently labeled with cy5; blue, cell nuclei stained with hochest33342; Scale bar: 20 μ m, n=3). Asterisk (*) denotes statistical significance between bars (*p < 0.05) using one-way ANOVA analysis.

3-13. Regarding the BBB in vitro model. The observed transcytosis transfer of the controls is very high (20%), which could indicate that there is notight cell layer. The authors should have done controls with different particles and for example dextran to establishthe integrity of the cell layer.

Answer: A lot of thanks for the valuable suggestion of the reviewer.

“The observed transcytosis transfer of the controls is very high (20%)” mentioned by the reviewer is due to the fact that the sample is Ang-PMSe (Table R6) prepared by functionalization of angiopep-2 peptide (Ang), and the high BBB permeation behavior of this sample is mainly attributed to its ability to specifically bind to BBB endothelial cells highly expressing receptor-related protein 1 (LRP-1), which mediates the cellular transport mechanism to promote BBB permeation (*Sci. Adv.* 2022, 8, eabm8011; *Adv. Mater.* 2020, 2000416; *Adv. Mater.* 2019, 31, 1903277). To confirm this, we set PMSe, a passive nanoparticle with neither Ang transporter peptide modification nor motility properties, as a control group and examined its BBB transport rate. As shown in Supplementary Fig. 30, the BBB cellular transport rate of PMSe nanoparticles was only 7.9%.

Table R6 BBB Transport ratio of different samples

Sample	Ang	Movement ability	Transport ratio (%)
PMSe	X	X	7.9%
Ang-PMSe	- . 1	X	24.4%
PAMSe	X	- . 1	30.3%
Ang-PAMSe	- . 1	- . 1	68.1%

Supplementary Fig. 30 BBB transport ratio. **a** Concentration and fluorescence-worthy standard curve of PMSe. **b** BBB transport ratio of different sample (n=3). Asterisk (*) denotes statistical significance between bars (*p < 0.05) using one-way ANOVA analysis.

Meanwhile, we examined the structural integrity of the nanomotors before and after penetration of the BBB, using laser confocal microscopy (CLSM) to observe the changes in structural compactness of the BBB layer before and after incubation with the material. In this case, the control was the replacement of fresh cell culture medium in the transwell upper chamber, and the material group was the addition of Ang-PAMSe nanomotors ($200 \mu\text{g mL}^{-1}$). Cells were labeled with fluorescence using a nuclear dye (DAPI) and a cell membrane dye (DiO), and the structure of the BBB was observed by CLSM. As shown in Supplementary Fig. 31, a dense BBB cell layer had been formed before the addition of Ang-PAMSe nanomotors. After adding Ang-PAMSe nanomotors and incubating for 24 h, the dense BBB cell structure remained dense.

Supplementary Fig. 31 CLSM plots of the BBB in vitro (before) and after the addition of fresh medium (control) and Ang-PAMSe nanomotors ($200 \mu\text{g mL}^{-1}$) and incubation for 24 h (after). (Green, cell membranes stained with DiO; Blue, cell nuclei stained with DAPI; Scale bar: $100 \mu\text{m}$)

As suggested by the reviewers, in this process, the integrity of the tight junctions of BBB model cells is particularly important. It is well known that dextran is a polysaccharide composed of branching glucose molecules of different lengths. FITC-dextran (FD-4, 4000 Da) is a polysaccharide composed of fluorescein isothiocyanate coupled to dextran coupling, which can be used to identify markers of BBB leakage (*Curr. Protoc. Neurosci.* 2017, 9, 58.1; *Toxicol. Lett.* 2015, 237, 79; *Food Funct.* 2017, 8, 1166). Therefore, we used FD-4 (Glpbio) to assess the integrity of BBB after the material penetrated the BBB.

The experimental procedure is referred to Supplementary Fig. 32a as follows:

(1) On Day 0, 200 μL bEnd.3 cells (cell density of $1.5 \times 10^5 \text{ cell mL}^{-1}$) was inoculated in the transwell upper chamber for continuous incubation to build the BBB layer.

(2) On Day 8, 500 μL of U-87 MG (cell density of $2.5 \times 10^5 \text{ cell mL}^{-1}$) was inoculated in the transwell lower chamber after continuous incubation to create ROS/iNOS concentration gradient.

(3) At Day 10, the original medium in the chambers was discarded, the control group was replaced with fresh cell culture medium, and 200 $\mu\text{g mL}^{-1}$ of Cy5 fluorescently labeled Cy5-Ang-PAMSe nanomotors were added to the experimental group. Then, the above chambers were transferred to the wells inoculated with cancer cells and incubated for 24 h.

(4) At Day 11, 200 μL of hank's balanced salt solution (HBSS) containing 1 mg mL^{-1} FD-4 solution was added to the transwell upper chamber and incubated for 2 h. After that, liquid in the chambers, liquid and cells in the well plates were collected separately and the fluorescence intensities of these materials were measured using a fluorescence spectrophotometer (Ex: 650 nm, Em: 670 nm) to calculate the transport rate.

(5) The liquid in the lower chamber was collected and the fluorescence intensity of FD-4 was measured using a fluorescence spectrophotometer (Ex: 492 nm and Em: 520 nm).

As seen from Supplementary Fig. 32b, the fluorescence intensity of FD-4 detected in the lower chamber did not increase significantly after 24 h of Ang-PAMSe nanomotor treatment compared with the control, further demonstrating that the Ang-PAMSe nanomotor penetrated the BBB and still maintained the structural compactness of the BBB.

Supplementary Fig. 32 a Schematic representation of BBB integrity assessment using FD-4. b Paracellular permeability of FD-4 after 24 h treatment with Ang-PAMSe nanomotor (n=3). Asterisk (*) denotes statistical significance between bars (* $p < 0.05$) using one-way ANOVA analysis.

In addition to the transwell monolayer cell penetration model available in the preliminary manuscript, we constructed a transwell model with multilayer of cells (bEnd.3 and U-87 MG). Compared with bEnd.3 monoculture, binary co-culture of bEnd.3 cells with U-87 MG glioma cells can better mimic in vivo microenvironment, increase tight junctions of BBB, and induce expression of specific receptors and transporters (*Acta Neurobiol. Exp. Wars, 2011, 71, 113; Neurosci. Methods 2013, 212, 211*).

As shown in Supplementary Fig. 33a, we constructed an in vitro multilayer symbiotic BBB model. The specific steps are as follows.

(1) At Day 0, 100 μL , U-87 MG cancer cells (cell density of cell density of $1.0 \times 10^6 \text{ cell mL}^{-1}$) were inoculated into the inverted transwell chamber bottom surface and incubated continuously to form a tumor cell layer.

(2) On Day 1, the chambers from step 1 were orthotopically placed and 200 μL bEnd.3 cells (cell density of $1.5 \times 10^5 \text{ cell mL}^{-1}$) were added to their interior for continuous incubation to build the BBB layer.

(3) On Day 8, 500 μL of U-87 MG (cell density of $2.5 \times 10^5 \text{ cell mL}^{-1}$) was inoculated in the transwell lower chamber after continuous incubation to create ROS/iNOS concentration gradient.

(4) At Day 10, the original medium in the chambers was discarded, the control group was replaced with fresh cell culture medium, and 200 $\mu\text{g mL}^{-1}$ of fluorescently labeled Cy5-PMSe and Cy5-Ang-PAMSe nanomotors were added to the experimental group, respectively, and the above chambers were transferred to the wells inoculated with cancer cells and incubated for 24 h.

(5) At Day 11, liquid in the chambers, liquid and cells in the well plates were collected separately and the fluorescence intensities of these materials were measured using a fluorescence spectrophotometer (Ex: 650 nm, Em: 670 nm) to calculate the transport rate.

As shown in Supplementary Fig. 33b, in the *in vitro* multilayer symbiotic BBB model, the BBB transport rate (61.1%) of Ang-PAMSe nanomotor with both BBB transport performance and motility was significantly higher than that of the passive nanoparticle PMSe (7.0%). The results were similar to that of the monolayer BBB model, which confirmed that the motility effect of the nanomotor and the transport function of Ang can help it penetrate the BBB.

On Day 11, the incubated chambers were transferred to a well plate with fresh medium and 200 μL of a solution containing 1 mg mL^{-1} FD-4/HBSS was added to the chambers and incubated for 2 h (Supplementary Fig. 33c). Next, the liquid in the well plate was collected and the fluorescence intensity of FD-4 was measured using a fluorescence spectrophotometer (Ex: 492 nm, Em: 520 nm). After incubation with PMSe and Ang-PAMSe, the fluorescence intensity of FD-4 detected in the transwell lower chamber was not significantly different from that of the control (relative fluorescence intensity was maintained at 764~930 a.u.), indicating that the BBB remained intact after nanomotor penetration and the confocal laser microscope (CLSM) images of the BBB (Supplementary Fig. 34) also further confirmed this result.

Supplementary Fig. 33 **a** Schematic diagram of *in vitro* multilayer symbiotic BBB model preparation. **b** Cellular transport rates of PMSe and Ang-PAMSe in a multilayer symbiotic BBB model. **c** Paracellular permeability of FD-4 after 24 h treatment with PMSe and Ang-PAMSe (The control group was fresh cell culture medium) (n=3). Asterisk (*) denotes statistical significance between bars (* $p < 0.05$) using one-way ANOVA analysis.

Supplementary Fig. 34 CLSM plots and fluorescence quantification of the upper co-cultured BBB and cancer cell layers in an in vitro multilayer symbiotic BBB model after adding fresh medium (control), PMSe and Ang-PAMSe nanomotors ($200 \mu\text{g mL}^{-1}$) and incubating for 24 h (after) (Green, cell membranes stained with DiO; blue, cell nuclei stained with DAPI; Scale bar: $100 \mu\text{m}$) ($n=3$).

3-14. Supp fig 8 doesn't provide any useful information.

Answer: Thanks so much for the comments of the reviewer.

Fig. S8 shows the TEM-mapping element distribution of the PAMSe nanomotor, and we have improved the picture quality, from which it can be seen that the elements (C, N, O, Se) are more uniformly distributed in the nanoparticles, which also confirms the presence of several important elements of the reacting species in the products.

Supplementary Fig. 8. TEM-assisted element mapping images of PAMSe nanomotors (Scale bar: 100nm).

3-15. Fig S36 cell viability is surprisingly high, in comparison with the later in vivo studies. Can the authors provide an explanation?

Answer: Many thanks for the valuable question of the reviewer.

The main principles of the nanomotor designed in this work to exert anti-tumor effects in vivo are

as follows. The Ang-PAMSe nanomotor can release NO in response to the tumor ROS/iNOS microenvironment, and NO can intervene in several aspects of the above immune cycle with the chemotactic behavior of the nanomotor (Fig. 1). For example, high concentrations of NO generated during nanomotor movement can act as an ICD inducer to enhance the production of tumor immune antigens (*Crit. Rev. Oncog.* 2016, 21, 365), thereby promoting the maturation of antigen-presenting cells and the activation of T cells (sessions 1-3) (*Front. Immunol.* 2013, 4, 438). Further, NO normalizes abnormal blood vessels in tumor tissues, thus improving the efficiency of T cell transport at tumor sites (session 4) (*Nat. Nanotechnol.* 2019, 14, 1160); NO also promotes tumor extracellular matrix degradation by reacting with ROS to produce ONOO⁻ upregulated matrix metalloproteinases (*Nano Lett.* 2019, 19, 997), thus effectively enhancing T cell and drug infiltration in tumor tissues (session 5). Also, NO can regulate macrophages to polarize them from M2 to M1 type; downregulate the expression of PDL-1 on the surface of tumor cells, thus reversing the microenvironment of tumor immunosuppression and improving the recognition of tumor cells by T cells (session 6) (*Nat. Nanotechnol.* 2019, 14, 1160). In summary, the nanomotor responds to ROS/iNOS in the tumor microenvironment and releases NO through the matrix component (arginine derivative), and with the chemotactic behavior of the nanomotor, it becomes an important therapeutic agent for immune-enhancing multistage intervention in brain tumors, thus constituting a cascade effect of the therapeutic process.

Among them, NO exerts anti-tumor effects, but more importantly, it activates the immune circulation in vivo (Fig. 5). As shown in Fig. 5 h,i, compared with the PBS group (48.2%), the performance of Ang-PAMSe in promoting DC maturation was somewhat enhanced, about 71.4%, almost 1.5 times larger than the PBS group. In addition, an increase in the proportion of CD3⁺CD8⁺ cytotoxic T cells (approximately 3.3-fold higher than in the PBS group) and CD3⁺CD4⁺ helper T cells (approximately 2.3-fold higher than in the PBS group) was found in the brain tissue of mice treated with Ang-PAMSe; the number of CD3⁺CD4⁺Foxp3⁺ Treg cells was reduced to 9.6% and 5.1% (Fig. 5j-5m), thus somewhat enhancing the immunotherapeutic effect and achieving improved treatment of GBM. In the in vitro cellular environment, because of the lack of immune cells such as DCs and T cells, cancer cells cannot be killed by the purpose of immune enhancement, but only by the anti-tumor effect of NO itself, which is an important reason for the limited killing ability. The limited killing ability of NO alone is also reflected in many other literatures (Table R7). Therefore, the Ang-PAMSe nanomotors designed in this paper did not show a strong ability to kill cancer cells in in vitro cellular assays (Supplementary Fig. 40), while they could exert some degree of antitumor effect in in vivo (Fig. 5e-5g). The ability of Ang-PAMSe nanomotors to activate the immune system in vivo can be confirmed by Fig. 5h-5o, where DCs maturation, increased percentage of cytotoxic T cells, and elevated cytokine levels including IL-6 and TNF- α . All of the above can indicate activation of the immune system, which is not achieved at the cellular level in vitro.

Fig. 1 | Preparation of nanomotor and schematic of multi-step targeting strategy/cascade effect for enhanced immunotherapy of glioblastoma.

Fig. 5 | Assessment of targeting and therapeutic efficacy of Ang-PAMSE/TLND nanomotors in vivo. **a** Chemiluminescence imaging of GBM in C57BL/6 mice before administration and fluorescence images of the brains after injection of different cy5-labeled samples via tail vein injection (Scale bar: 2 cm). **b**

Quantitative analysis of sample accumulation in main organs. Cy5-sample levels were determined by fluorescence spectroscopy and expressed as injected dose per gram of tissue (%ID g⁻¹). **c** Representative CLSM images of brain tumor tissue in mice 24 h after injection of different samples via tail vein and red fluorescence intensity profiles as a function of the distance from a blood vessel in the representative region marked by the white rectangular frames. (Green, blood vessels are stained with CD31; red, samples are labeled with red fluorescence by cy5; blue, cell nuclei stained with DAPI; N: normal brain tissue; T: tumor; Scale bar: 100 μm); (The names corresponding to the samples in a-c are (I) PBS, (II) Ang-PMSe, (III) PAMSe, (IV) Ang-PAMSe). **d** Treatment protocols for orthotopic brain-GBM-tumor-bearing models. Where the red arrow is the time point of the drug administration and the blue arrow represents the time point of bioluminescence imaging. **e** Representative IVIS spectrum images and **(f)** quantified signal intensity (n = 3 biologically independent animals per group). **g** Survival analysis of the mice that loaded with GL261-Luc glioblastoma during treatment process (n=8, biologically independent animals per group). **h** Quantitative analysis of the DCs maturation level in vivo and **(i)** flow cytometry analysis of the expression of CD80 and CD86 on the surface of DCs extracted from the tumor draining lymph nodes of mice after various treatments. For the gating strategy for DCs maturation analysis refer to Supplementary Fig. 49. (n=3 biologically independent experiments). **j** Quantitative analysis of CD3⁺CD8⁺ cytotoxic T cells, and **(k)** representative flow cytometric plots of CD8⁺ T cells in GBM-bearing brain tissue gating on CD3⁺ cells in each group. For the gating strategy for T cells analysis refer to Supplementary Fig. 51 (n=3 biologically independent experiments per group). **l** Quantitative analysis of CD3⁺CD4⁺ T cells. **m** The quantitative analysis of CD3⁺CD4⁺Foxp3⁺ Tregs in the brain tissue of each group according to flow cytometry in each group. **n** Cytokine levels including IL-6 and **(o)** TNF-α in the serum of mice at the end of treatment (n=3). (The names corresponding to the group in e-o are (I) Sham, (II) PBS, (III) TLND, (IV) Ang-PMSe/TLND, (V)) Ang-PAMSe and (VI) Ang-PAMSe/TLND). Asterisk (*) denotes statistical significance between bars (*p < 0.05) using one-way ANOVA analysis.

Supplementary Fig. 40 **a** The viability of cancer cells after treatment with degraded PAMSe (20, 50, 100, 150 and 200 μg mL⁻¹) for 48 h. **b** The viability of endothelial cells (HUVECs) after incubation with different concentrations of PAMSe nanomotors (20, 50, 100, 150 and 200 μg mL⁻¹) for 48 h. **c** The viability of cancer cells after incubation with different concentrations of PAMSe nanomotors (20, 50, 100, 150 and 200 μg mL⁻¹) for 48 h (n=3).

Table R7 The viability of cancer cells after treatment with different samples.

Samples	Concentration	Cell type	Cell viability	Ref.
HM/CPT-11/NONOate	4 mg mL ⁻¹	MCF-7/ADR	~50.0 %	Angew. Chem. 2015, 54, 9890.
NanoNO-SNP	~0.5 mM	4T1	~42.5%	ACS Nano 2022, 16, 3881.
S1P/JS-K/Lipo	JS-K equivalent dose 20 μg mL ⁻¹	U87	~43.0%	Adv. Mater. 2021, 2101701.
PAMSe	200 μg mL ⁻¹	U87	~49.7%	Our work

3-16. Fig S37, the X axis is unspecified, It is not clear what is compared.

Answer: Thousands of thanks for the helpful question of the reviewer.

According to the reviewer's questions, we have added a detailed explanation of the sample

information on the X-axis in the figure and the illustration respectively.

Supplementary Fig. 35. Cellular viability of GL261 cells after 24 h of sample treatment (n=3). The samples are PMSel/A, PAMSe, TLND and PMSel/A/TLND with concentration of 200 μg mL⁻¹ and PAMSe/TLND with concentration of 20 μg mL⁻¹, 50 μg mL⁻¹, 100 μg mL⁻¹, 150 μg mL⁻¹ and 200 μg mL⁻¹ respectively. Asterisk (*) denotes statistical significance between bars (*p < 0.05) using one-way ANOVA analysis.

3-17. Fig 5 d please specify groups V and VI in the figure caption.

Answer: Thanks a lot for the reviewer's questions.

The group names corresponding to (I) -(VI) in figure 5d-5n were added in the figure caption of the revised manuscript.

Reviewer #4 (Remarks to the Author): with expertise in glioblastoma, cancer nanotechnology

Chen et al present the design of a complex nanoparticle with multiple functions. The overall premise is intriguing but not all aspects are well supported, especially the antitumor immune response. First, the nanoparticle has interesting release profiles especially in different media. Second, the in vitro data are supportive of the chemotactic spread of the nanomotors across an iNOS gradient. Further, the in vivo data seem to corroborate the in vitro findings. However, the data supporting the main claim of the work, that is activation of the innate arm of immunity within the tumor, are not convincing. Therefore, this work requires new essential experimentation and significant revisions.

Answer: We are pleased that the referee finds our work of general intriguing. We are also grateful to have this opportunity to further clarify and substantiate the working mechanism of the nanomotor for intratumoral activation of the innate immune system.

Major issues

1. - The immunological aspects of the work are not well supported. The issue in GBM is not necessarily the absence of neoantigens but the rather immunosuppressed nature of the tumor immune microenvironment. It is not clear why DCs and TAMs increase their cross-presentation while no direct function of the NO is targeting the activation and proper reprogramming of the antigen-presenting cells. Since this is the main key function of the particle, the paper is missing important data supporting this function.

Answer: Thanks a lot for the reviewer's question.

Glioblastoma (GBM) is an extremely malignant tumor. As the reviewer pointed, the key factor limiting immunotherapy in GBM may not only be the absence of neoantigens but also its complex tumor immune mechanism, including the issue of immunosuppression mentioned by the reviewer. As early as 2013, researchers proposed the concept of tumor immune cycle, including the following steps. (1,2) release and presentation of the tumor antigen; (3,4) activation and transport of the effector T cell; (5) infiltration of T cell into tumor tissue; (6,7) recognition and clearance of tumor cells by T cell (*Immunity* 2013, 39, 1). Current studies often use different drugs to intervene in the limited steps of the above cycle, for instance, chemotherapeutic drugs were often applied to induce tumor immunogenic cell death (ICD), so as to expose tumor immune antigens and promote the presentation of tumor antigens (for steps 1 and 2) (*Chem. Soc. Rev.* 2019, 48, 5506); or immune checkpoint inhibitors were used to enhance the activity of T cells (for steps 6 and 7) (*Int. Immunopharmacol.* 2018, 62, 29). Yet, the effect of intervention in limited steps may be not enough to effectively enhance the effect of immunotherapy due to immune escape phenomenon in other steps (*Front. Oncol.* 2019, 9, 1143). Moreover, if different drugs are used at the same time to intervene above steps, the therapeutic agent and administration mode need to be designed in a complex way, and there is a risk of excessive immune response, namely immune related adverse events (irAEs) (*Cell* 2018, 175, 313).

For the problems faced by immunotherapy of brain tumors, the nanomotor constructed in this work is intended to carry out the multi-step intervention synergism in the tumor immune cycle process through its beneficial product NO released in the microenvironment of brain tumors and the loaded drug TLND (Fig. 1). The high concentration NO produced during the chemotaxis process of the nanomotor can be used as an ICD inducer to enhance the production of tumor immune antigen (*ACS Nano* 2022, 16, 3881). Thus, it can promote the maturation of antigen presenting cells (steps 1, 2) (*Nat. Rev. Cancer* 2012, 12,

860); Furthermore, NO can normalize abnormal blood vessels in tumor tissue, thereby improving the transport efficiency of T cells at tumor sites (step 3, 4) (*Nat. Nanotechnol.* 2019, 14, 1160); NO can also produce ONOO⁻ to up-regulate matrix metalloproteinase by reacting with ROS, and promote the degradation of tumor extracellular matrix (*Nano Lett.* 2019, 19, 997), thus effectively improving the infiltration of T cells and drugs in tumor tissue (step 5); Meanwhile, the specific selective inhibition of the functions of aerobic glycolysis and energy metabolism in tumor cells by drug TLND makes it possible to improve the tumor immune suppression microenvironment by destroying the tumor metabolic symbiosis process (*Biochem. J.* 2016, 473, 1503) (step 6 and 7). In a word, the nanomotor releases NO in response to the tumor microenvironment through the matrix component (L-arginine derivative), and becomes an important therapeutic agent for brain tumor to enhance the immune multi-step intervention by virtue of the chemotaxis behavior of the nanomotor, which constitutes a cascading effect in the treatment process.

Meantime, in vitro and in vivo experiments were designed to verify the above process in detail. The detailed information is as follows.

Steps 1 and 2 (NO induces ICD and promotes dendritic cells (DCs) maturation): Supplementary Fig. 38 and Supplementary Fig. 39 show the levels of calreticulin (CRT) exposed on the tumor cell surface and high mobility group box 1 (HMGB1) released extracellularly after treatment with different samples, respectively. As shown in these figures, treatment with PAMSe and PAMSe/TLND nanomotors with NO-releasing properties can lead to significant increase in CRT exposure and extracellular HMGB1 content on the tumor cell surface. And the tumor cells treated with PAMSe/TLND nanomotors showed significant cell surface CRT exposure with a fluorescence intensity about 25 times higher than that of the control group; and a large amount of extracellular HMGB1 release with a release amount about 5 times higher than that of the control group.

In addition, the proportions of mature DCs (CD11b⁺CD80⁺CD86⁺) in mouse cervical lymph nodes (Fig. 5h and Fig. 5i) and brain tumor tissue (Supplementary Fig. 50) were examined by flow cytometry, respectively. The results showed that compared with the PBS group, the percentage of mature DCs in the draining lymph nodes of mice treated with PAMSe/TLND nanomotors increased from 48.2% to 90.8%; the percentage of mature DCs in tumor tissues increased from 12.6% to 34.1%, which confirmed that NO could promote the maturation of DCs.

Steps 3-5 (Nanomotors modulate the tumor microenvironment and enhance the activity of T cells and their infiltration in tumor tissues): To explore the potential mechanism of immune activation triggered by Ang-PAMSe/TLND nanomotors, we evaluated the infiltration of immune cells (cytotoxic T cells (CD3⁺CD8⁺) in tumor tissues. Among them, toxic T cells represent cells with direct tumor-killing effects, helper T cells represent the proliferation and spread to activate other types of immune cells that produce direct immune responses. Serum and brain tumors were collected from mice at the end of treatment for T-cell infiltration analysis and cytokine assay. Fig. 5j-5m showed that more CD3⁺CD8⁺ cytotoxic T cells (about 5.2 times more than in the PBS group) and CD3⁺CD4⁺ helper T cells (about 4.1 times more than in the PBS group) were found in the brain tissue of mice treated with Ang-PAMSe/TLND nanomotors.

Step 6 (Nanomotors reverse the immunosuppressed tumor microenvironment and enhance T cell recognition of cancer cells): The immunosuppressive microenvironment in tumors, such as macrophages polarization from M1 phenotype to M2 phenotype, will affect the immune response of T cell, thus promoting the immune escape of tumor cells and tumor progression. The production of NO can reverse the immunosuppressive microenvironment of tumors, for example, it can regulate the polarization of

macrophage to the M1 phenotype, inhibit the expression of PDL1 in cancer cells, and suppress Treg cell levels, (*Nat. Nanotechnol.* 2019, 14, 1160; *Nature* 2017, 544, 250; *J. Am. Chem. Soc.* 2021, 143, 12025). Therefore, we examined these two indicators separately. According to the treatment procedure in Supplementary Fig. 47, after 5 times of treatment with different samples (PBS, TLND, Ang-PMSe/TLND, Ang-PAMSe and Ang-PAMSe/TLND), there was no difference in the proportion of M1-type macrophages (CD11b⁺CD86⁺) in GBM tissues of mice treated with TLND and Ang-PMSe/TLND groups compared with the PBS group (Supplementary Fig. 52), while Ang-PAMSe and Ang-PAMSe/TLND treatment groups increased to 10.6% and 13.4%, which were approximately 3.6 and 4.5 times higher than those in the PBS group. We also examined the expression levels of PDL1 of cancer cells in treated GBM tissues. As shown in Supplementary Fig. 53, compared to the PBS group (73.8%), the PDL1 expression level in Ang-PAMSe (40.8%) and Ang-PAMSe/TLND (31.4%) groups decreased significantly, about 0.6 and 0.4 times lower than that of the PBS group. Then, in Ang-PAMSe and Ang-PAMSe/TLND-treated groups, the number of CD3⁺CD4⁺Foxp3⁺ Treg cells was significantly decreased to 5.1% and 3.5%, respectively (Fig. 5m), indicating that NO produced by Ang-PAMSe and Ang-PAMSe/TLND not only enriched CD3⁺CD8⁺ cytotoxic T cells in brain tissue, but also downregulated the frequency of CD3⁺CD4⁺Foxp3⁺ Treg, alleviated Treg-associated immune braking, and further enhanced antitumor immunity. The secretion of immune-related cytokines interleukin-6 (IL-6) and tumor necrosis factor- α (TNF- α) in the serum of mice was also detected. As shown in Fig. 5 n,o, the serum levels of IL-6 and TNF- α cytokines were higher in mice treated with Ang-PAMSe/TLND than in other treatment groups.

Thus, PAMSe and Ang-PAMSe/TLND can continuously release a large amounts of NO in the tumor environment, which can be fully used to regulate the polarization of macrophages from M2 to M1 phenotype, inhibit the expression of PDL1 in cancer cells, and suppress Treg cell levels, thus achieving the reversal of the immunosuppressed tumor microenvironment.

Also, regarding the reviewer's mention of “It is not clear why DCs and TAMs increase their cross-presentation while no direct function of the NO is targeting the activation and proper reprogramming of the antigen-presenting cells.” We carefully reviewed the literature and found the mechanism of NO leading to the maturation and activation of DCs in the following process. That is, it depends on the increase of endoplasmic reticulum (ER) stress and mitochondrial dysfunction in tumor cells, which contributes to the release of DAMPs, leading to DCs maturation and activation (*ACS Nano* 2022, 16, 3881). The specific mechanism is that NO-induced ER stress leading to CRT translocation. During ER stress, the release of ER Ca²⁺ promotes the loss of $\Delta\Psi_m$ and the production of ROS on mitochondria, thus triggering the permeability of the outer membrane of mitochondrial. In turn, the increase of ROS feeds back to the ER, further causing ER stress and Ca²⁺ release, leading to stronger ER pressures and the momp-induced apoptotic pathways. The permeability of mitochondrial outer membrane in turn triggers the release of cytochrome c, NF- κ B activation and mitochondrial autophagy, leading to a massive release of immunogenic DAMPs. DAMP mainly includes CRT exposure on the cell surface and extracellular release of HMGB1 (*Nat. Immunol.* 2022, 23, 487; *Nat. Rev. Cancer* 2012, 12, 860). As mentioned previously, we demonstrated the final results generated by this process in detail.

Step 7 (Nanomotors and TLND synergistically regulate the abnormal metabolic patterns of tumor cells): As shown in Fig. 3f, the contents of intracellular and extracellular lactate after treatment with different samples were examined separately. The intracellular lactate content of TLND-treated cells increased to some extent (2.6 mmol g⁻¹ protein in 24 h), while the extracellular lactate content did not change much (maintained at about 0.9 mmol g⁻¹ protein), which confirmed that TLND could indeed inhibit the lactate excretion from tumor cells. The trend of lactate production in the cells after PAMSe

nanomotor treatment was like that of the control group, indicating that nanomotors with autonomous motility only have little effect on this metabolic process. Treatment of cells with PAMSe/TLND nanomotors with autonomous motility and TLND release capacity revealed a significant increase in intracellular lactate content with time (4.2 mmol g⁻¹ protein in 24 h) and a slight increase in extracellular lactate content (1.1 mmol g⁻¹ protein in 24 h). This result confirmed that PAMSe/TLND nanomotors can induce lactate accumulation in cancer cells. To describe this result more clearly, Fig. 3g summarized the ratio of intracellular to extracellular lactate content, and the ratio of intracellular to extracellular lactate content after treatment with PAMSe/TLND nanomotors was 3.9, which was significantly higher than that after treatment with free TLND (2.8).

We further assessed whether the accumulation of intracellular lactic acid induced by PAMSe/TLND nanomotors could lead to acidosis within cancer cells. As shown in Fig. 3h,i, consistent with the trend of intracellular and extracellular lactate accumulation, tumor cells treated with PAMSe/TLND nanomotors had the lowest intracellular pH value (~5.9). In addition, we also evaluated the effect of PAMSe/TLND nanomotors on the generation of adenosine triphosphate (ATP) in tumor cells. As shown in Fig. 3j, the ATP concentration of cells treated with TLND and PAMSe decreased within 24 h with the extension of time. The former may be since TLND itself, as an inhibitor of hexokinase during aerobic glycolysis of cancer cells, can inhibit the production of ATP, while the latter may be caused by the fact that PAMSe nanomotors produced a certain concentration of NO in the tumor cell environment, which played a certain degree of killing effect on tumor cells. Compared with the above groups, the PAMSe/TLND nanomotors had an inhibitory effect on the cellular ATP content with time extension (the ATP content decreased from 11.8 μM to 1.5 μM after 24 h).

Meanwhile, the final therapeutic effect brought by the synergistic effect of treatment was also verified by *in vivo* experiments. As shown in Fig. 5e-5g, the PAMSe/TLND nanomotor showed significant inhibition effect on GBM in the 15 day treatment cycle. The median survival time of mice was 71 days, and 37.5% of mice still survived on the 100th day.

Further, the PAMSe/TLND nanomotor can trigger the activation of the complete immune cycle *in vivo*. The activated immune cells can not only play a role in the tumor tissue, but also effectively kill tumor cells infiltrating other parts of the brain and other parts of the body. This is an important reason why the therapeutic agent involved in this paper can prevent tumor metastasis, which can be confirmed in Fig. 6a-6i. More importantly, the immune system activated in this way also has a memory function, which can prevent tumor recurrence (Fig. 6j).

Once again, we thank the reviewers for their questions in immunology, and we have improved and added to the manuscript on antitumor immune response based on the reviewers' suggestions, sorting out the problems faced in GBM immunotherapy and focusing on the problems faced in the treatment process.

Fig. 1 | Preparation of nanomotor and schematic of multi-step targeting strategy/cascade effect for enhanced immunotherapy of glioblastoma.

Supplementary Fig. 38 Representative CLSM images corresponding quantitative analysis of cell surface exposure of CRT (The fresh cell culture medium as control; Green, CRT was immunofluorescently (FITC-488) labeled; blue, cell nuclei stained with hochest33342; Scale bar: 20 μ m n=3). Asterisk (*) denotes statistical significance between bars (* $p < 0.05$) using one-way ANOVA

analysis.

Supplementary Fig. 39 Quantitative analysis of extracellular release of high mobility group box 1 (HMGB1), and the fresh cell culture medium as control (n=3). Asterisk (*) denotes statistical significance between bars (* $p < 0.05$) using one-way ANOVA analysis.

Fig. 5 | Assessment of targeting and therapeutic efficacy of Ang-PAMSe/TLND nanomotors in vivo. **a** Chemiluminescence imaging of GBM in C57BL/6 mice before administration and fluorescence images of the brains after injection of different cy5-labeled samples via tail vein injection (Scale bar: 2 cm). **b** Quantitative analysis of sample accumulation in main organs. Cy5-sample levels were determined by fluorescence spectroscopy and expressed as injected dose per gram of tissue (%ID g⁻¹). **c** Representative CLSM images of brain tumor tissue in mice 24 h after injection of different samples via tail vein and red

fluorescence intensity profiles as a function of the distance from a blood vessel in the representative region marked by the white rectangular frames. (Green, blood vessels are stained with CD31; red, samples are labeled with red fluorescence by cy5; blue, cell nuclei stained with DAPI; N: normal brain tissue; T: tumor; Scale bar: 100 μ m); (The names corresponding to the samples in a-c are (I) PBS, (II) Ang-PMSe, (III) PAMSe, (IV) Ang-PAMSe). **d** Treatment protocols for orthotopic brain-GBM-tumor-bearing models. Where the red arrow is the time point of the drug administration and the blue arrow represents the time point of bioluminescence imaging. **e** Representative IVIS spectrum images and **(f)** quantified signal intensity ($n = 3$ biologically independent animals per group). **g** Survival analysis of the mice that loaded with GL261-Luc glioblastoma during treatment process ($n=8$, biologically independent animals per group). **h** Quantitative analysis of the DCs maturation level in vivo and **(i)** flow cytometry analysis of the expression of CD80 and CD86 on the surface of DCs extracted from the tumor draining lymph nodes of mice after various treatments. For the gating strategy for DCs maturation analysis refer to Supplementary Fig. 49. ($n=3$ biologically independent experiments). **j** Quantitative analysis of CD3⁺CD8⁺ cytotoxic T cells, and **(k)** representative flow cytometric plots of CD8⁺ T cells in GBM-bearing brain tissue gating on CD3⁺ cells in each group. For the gating strategy for T cells analysis refer to Supplementary Fig. 51 ($n=3$ biologically independent experiments per group). **l** Quantitative analysis of CD3⁺CD4⁺ T cells. **m** The quantitative analysis of CD3⁺CD4⁺Foxp3⁺ Tregs in the brain tissue of each group according to flow cytometry in each group. **n** Cytokine levels including IL-6 and **(o)** TNF- α in the serum of mice at the end of treatment ($n=3$). (The names corresponding to the group in e-o are (I) Sham, (II) PBS, (III) TLND, (IV) Ang-PMSe/TLND, (V) Ang-PAMSe and (VI) Ang-PAMSe/TLND). Asterisk (*) denotes statistical significance between bars (* $p < 0.05$) using one-way ANOVA analysis.

Supplementary Fig. 47 Schematic illustration of the orthotopic GL261 glioblastoma tumor-bearing C57BL/6J mice experimental design.

Supplementary Fig. 50 a Flow cytometry analysis of DCs maturation with each formulation, and **(b)** Frequency of DCs activation from the tumor tissue ($n=3$ biologically independent experiments). Asterisk (*) denotes statistical significance between bars (* $p < 0.05$) using one-way ANOVA analysis.

Supplementary Fig. 52 Representative flow cytometric plots of (a) M1-like macrophages (CD11b⁺CD86⁺) and (b) M2-like macrophages (CD11b⁺CD206⁺) in glioma-bearing brain tissue (n=3 biologically independent experiments per group) and (c, d) the corresponding quantitative analysis. e TAM M1/M2 polarization ratio. Asterisk (*) denotes statistical significance between bars (*p < 0.05) using one-way ANOVA analysis.

Supplementary Fig. 53 Representative flow cytometric plots of CD45⁺PDL1⁺ glioma cells in glioma-bearing brain tissue (n=3 biologically independent experiments per group) and quantitative analysis of CD45⁺PDL1⁺ cancer cells. Asterisk (*) denotes statistical significance between bars (*p < 0.05) using one-way ANOVA analysis.

Fig. 3 | Cascade effect of PAMSe/TLND nanomotors on the regulation of cellular metabolism. **a** Schematic illustration of the cellular metabolism regulated by PAMSe/TLND nanomotors. **b** CLSM images of intracellular ROS (labeled with the ROS fluorescent probe, DCFH-DA) in cancer cells after PAMSe/TLND treatment for 24 h (Scale bar: 100 μ m). **c** Fluorescence intensity (F. I.) of intracellular ROS in cancer cells after different samples treatment for 24 h (n=3). **d** CLSM images of intracellular NO (labeled using the NO fluorescent probe, DAF-FM DA) in cancer cells after PAMSe/TLND nanomotors treatment for 24 h (Scale bar: 100 μ m). **e** Fluorescence quantification of intracellular NO in cancer cells after different samples treatment for 24 h (n=3). **f** The intracellular and extracellular increased content of lactic acid after being treated with different samples (n=3). **g** The ratio of intracellular and extracellular lactic acid increased content after being treated with different samples (n=3). **h** The CLSM images with pH fluorescent probes (with pseudo color) (Scale bar: 20 μ m) and **(i)** corresponding pH values of cancer cells after treatment with different samples for 24 h (n=3). **j** The ATP concentration in cancer cells after incubating with different samples for 24 h (n=3). (The samples in this figure are fresh cell culture medium as control, TLND, PAMSe and PAMSe/TLND) Asterisk (*) denotes statistical significance between bars (* $p < 0.05$) using one-way ANOVA analysis.

Fig. 6 | Localized metabolic immunotherapy to achieve distal effect for the GBM model and prevention of GBM recurrence. **a** The schematic illustration of the experimental design to determine the resistance to metastasis of the therapeutic agent. The blue arrow represents the time point of imaging. **b** The bioluminescence imaging of the sham-operated group and GBM mice (red circles indicate chemiluminescence imaging) and the bioluminescence imaging was performed on the 7 th and 14 th day after the treatment process (n=3). **c** Representative flow cytometric plots showing different groups of T

cells in abscopal GBM (the right tumor). For the gating strategy for T cells analysis refer to Supplementary Fig. 55 (n=3). **d** Quantitative analysis of CD3⁺CD8⁺ cytotoxic T cells (n=3). **e** Quantitative analysis of CD3⁺CD4⁺ T cells (n=3). **f** The quantitative analysis of CD3⁺CD4⁺Foxp3⁺ Tregs in the distant brain tumor tissue (n=3) (The names corresponding to the samples in b-f are (I) Sham, (II) PBS, (III) TLND, and (IV) Ang-PAMSe/TLND). **g** Cytokine levels including IL-6 and **h** TNF- α in the serum of mice at the end of treatment. **i** Determination of IFN- γ content in mouse serum (n=3). **j** Monitoring of GBM recurrence after various treatments for primary orthotopic tumors (n=3). (The names corresponding to the samples are (I) PBS, (II) TLND, (III) Ang-PMSe/TLND (IV) Ang-PAMSe, and (V) Ang-PAMSe/TLND). Asterisk (*) denotes statistical significance between bars (*p < 0.05) using one-way ANOVA analysis.

2. - The authors should clarify which are the tumor-draining lymph nodes for a GBM. This is not trivial or standard. Why didn't the authors collect and analyze DCs and macrophages from the tumor itself?
Answer: We thank the reviewer a lot for this insightful question.

Tumor-draining lymph nodes are important sites of the human body's anti-tumor immune response, containing a large number of immune cells and located downstream of the tumor, which can reflect to some extent the effect of activation of antigen-presenting cells in vivo. Therefore, we preferentially collected GBM tumor-draining lymph nodes located in the neck of mice and assessed the maturation of DCs therein (Fig. 5h and Fig. 5i), similar to many studies in this direction (*Sci. Adv.* 2020, 6, eaaz4204; *Angew. Chem. Int. Ed.* 2020, 59, 2).

In addition, we also collected GBM tissues and evaluated their DCs maturation according to the reviewer's suggestion. As shown in Supplementary Fig. 50, compared with PBS treatment group (12.6%), the percentage of mature DCs (D11c⁺ CD80⁺CD86⁺) in tumor tissues after Ang-PAMSe nanomotor treatment increased, and the percentage of mature DCs in brain tumor tissues of mice treated with Ang-PAMSe/TLND with cascade effect further increased to 34.1%. These results further confirmed that nanomotors with cascade effect can release more NO, thus effectively triggering the ICD of tumor cells, and ultimately inducing DCs to mature.

Supplementary Fig. 50 a Flow cytometry analysis of DCs maturation with each formulation, and (b) frequency of DCs activation from the tumor tissue (n=3 biologically independent experiments). Asterisk (*) denotes statistical significance between bars (*p < 0.05) using one-way ANOVA analysis.

3. - The main hypothesis of the paper is based on the activity of NPs in cancer cells. this assumes the NPs were taken up by cancer cells. The in vivo studies should include flow cytometry studies analyzing the cellular uptake of NPs in the tumor. Many studies show NPs are predominantly taken up by phagocytic cells and not glioma cells.

Answer: Many thanks for the helpful suggestion from the reviewer.

To fully verify whether NPs are taken up by glioma cells in the tumor, tumor-bearing mice with successful GBM construction were injected with PBS via tail vein as a negative control, while the rest of the mice were injected with Cy5-Ang-PAMSe ($200 \mu\text{L}$, 3 mg mL^{-1}) once daily, and GBM tissues were collected from the mice after 3 days. Tumor tissue was cut and transferred to a sieve and ground to obtain a single cell suspension. Macrophages were specifically labeled with FITC-anti-mouse CD45 (Biolegend) and Percp-cy5.5- anti-mouse-CD11b (Biolegend), and cells engulfed with Cy5-Ang-PAMSe were sorted out using flow cytometry. As shown in Fig. R11a, macrophages ($\text{CD11b}^+\text{CD45}^+$) were first sorted out in the cell population, and on this basis the negative region of Cy5-Ang-PAMSe was delineated using the PBS group as a negative control, and the procedure was applied to the sample group as well. Next, we analyzed the uptake of Cy5-Ang-PAMSe by tumor cells. As shown in Fig R11b, tumor cells (CD45^-) were first sorted out in the cell population, and the negative region of Cy5-Ang-PAMSe was delineated using the PBS group as a negative control, and the procedure was applied to the sample group as well. The results showed that Cy5-Ang-PAMSe nanomotors distributed in GBM tissue are mainly uptake by tumor cells compared to macrophages.

Fig. R11 Representative flow cytometric plots of the uptake of Cy5-Ang-PAMSe nanomotors by (a) macrophages ($\text{CD11b}^+\text{CD45}^+$), (b) cancer cells (CD45^-), and (c) the corresponding quantitative values.

4.- Are DC activated due to the immunogenicity of the NPs (NP being the antigen) or due to cancer-associated antigens?

Answer: Thanks a lot for the reviewer's questions.

In this work, DCs are activated due to the release of cancer-associated antigens. Supplementary Fig. 38 and Supplementary Fig. 39 showed the levels of calreticulin (CRT) exposed on the tumor cell surface and high mobility group box 1 (HMGB1) released extracellularly after treatment with different samples, respectively. As shown in these figures, treatment with PAMSe and PAMSe/TLND nanomotors with NO-releasing properties can lead to significant increase in CRT exposure and extracellular HMGB1 content on the tumor cell surface. And the tumor cells treated with PAMSe/TLND nanomotors showed significant cell surface CRT exposure with a fluorescence intensity about 25 times higher than that of the control group; and a large amount of extracellular HMGB1 release with a release amount about 5 times higher than that of the control group. Dependent on increased endoplasmic reticulum (ER) stress and mitochondrial dysfunction in tumor cells, which facilitates the release of DAMPS leading to DCs maturation and activation (*ACS Nano* 2022, 16, 3881). The specific mechanism is NO-induced ER stress leading to CRT translocation. During ER stress, the release of ER Ca^{2+} promotes the loss of NPM and the production of ROS on mitochondria, triggering outer mitochondrial membrane permeability. In turn, the increase in ROS is fed back to the ER, further causing ER stress and Ca^{2+} release, leading to stronger ER stress and mOMP-induced apoptotic pathways. Mitochondrial outer membrane permeability in turn triggers the release of cytochrome c, NF- κ B activation and mitochondrial autophagy, leading to a massive release of immunogenic DAMPs. DAMP consists mainly of exposure to CRT on the cell surface and extracellular release of HMGB1 (*Nat. Immunol.* 2022, 23, 487; *Nat. Rev. Cancer* 2012, 12, 860).

Also, as stated by the reviewer, nanoparticles may interact with different components of body fluids in vivo and adsorb different biomolecules (mainly protein molecules) on the surface of nanomaterials to form biomolecular crowns (e.g. protein crowns), which in turn are recognized by immune cell surface receptors and easily captured and phagocytosed by immune cells or accumulated in the mononuclear phagocyte system, producing immunogenicity and immunotoxicity and also leading to allergy-like reactions (*Nat. Biotechnol.* 2015, 33, 941). The nanomotors constructed in this paper have good resistance to protein adhesion and do not readily form protein crowns (*Sci. Adv.* 2020, 6, eaba0754), and thus may be able to greatly reduce the possible adverse immune responses caused by nanoparticles in vivo without causing immunogenicity. To test this speculation, the inflammatory responses of healthy SD rats were examined 48 h after receiving tail vein injection of nanomotors. As shown in Fig. R12, there was no significant difference in the levels of pro-inflammatory factors such as IFN- γ , TNF- α and IL-6 in the serum of rats injected with Ang-PAMSe/TLND nanomotors and untreated rats, indicating that the nanomotors injected in healthy SD rats did not induce significant immune response and no immunogenicity in vivo.

Fig. R12 Serum levels of inflammatory factors IFN- γ , TNF- α and IL-6 in healthy SD rats after receiving tail vein injection of PBS (control) and Ang-PAMSe/TLND nanomotors for 48 h.

5. - It is not clear why or how Tregs were decreased.

Answer: We thank the reviewer a lot for his/her valuable question.

Regulatory T cells (Treg cells) are a subset of T cells that inhibit T cell activity and cytokine production to suppress immune responses. Transforming growth factor beta (TGF13) is essential for the differentiation of Treg cells from primary CD4⁺ cells and is a regulator of the differentiation of CD4 T cells into Treg cells. Lee and colleagues have reported that exogenous NO donors such as DETA-NONOate and SNAP can modulate TGF-13 to promote its differentiation to type 1 helper T (Th1) cells and antagonize Foxp3⁺-induced Treg differentiation (*J. Immunol.* 2011, 186, 6972; *Cell Res.* 2002, 12, 311). Accordingly, we hypothesize that NO produced by nanomotors has a regulatory effect on TGF13 secreted by glioma cells, which in turn affects the ratio of Tregs. To test this hypothesis, we divided the tumor-bearing mice (GL261-C57BL/6J mice, modeling time 10 days) into 5 groups and started to administer the drugs (PBS, TLND, Ang-PMSe, PAMSe and Ang-PAMSe) on day 11, once every 2 days for 5 doses. On day 20, blood was obtained via the mouse orbit and brain tumor tissue was collected. Blood was centrifuged (3000 rpm, 30 min) and serum was collected; Tumor tissues were homogenized by grinding using a tissue homogenizer and the supernatant was collected by centrifugation (3000 rpm, 10 min) and assayed step by step using the Rat TGF-131 ELISA Kit (EK981-96, Multi sciences) according to the kit assay instructions.

As shown in Fig. R13, there was a significant decrease in TGF-13 levels in serum and in brain tumor tissues of mice treated with PAMSe and Ang-PAMSe with NO-releasing ability compared with the PBS group. Among the serum samples, there was a significant downward trend in the concentration of TGF-13 in the PAMSe group (27.4 ng mL⁻¹) and Ang-PAMSe group (25.2 ng mL⁻¹) compared to the PBS group (56.8 ng mL⁻¹). In tumor tissues, the levels of TGF-13 in the PAMSe group and Ang-PAMSe group were 985.0 ng g⁻¹ tumor tissue and 764.0 ng g⁻¹ tumor tissue, respectively, which were only about 1/3 of those in the PBS group. Based on the above experimental results, we concluded that NO released by PAMSe and Ang-PAMSe in the tumor microenvironment significantly inhibited the secretion of TGF13 from glioma cells, which in turn inhibited the level of CD4⁺ cells towards Foxp3⁺ regulatory T cell differentiation.

Fig. R13 Standard concentration curve of TGF-13, and TGF-13 levels in serum and tumor tissues of tumor-bearing mice after treatment with different samples.

Minor issues

6.- A detailed biodistribution and PK study should be performed providing the % ID in each organ.

Answer: Thousands of thanks for the valuable question of the reviewer.

Based on the reviewer's suggestion we added the quantitative assay of material biodistribution in the following steps:

(1) The tumor-bearing mice were randomly divided into 4 groups, and Cy5-Ang-PMSe (chemical recognition targeting), Cy5-PAMSe (chemokine targeting) and Cy5-Ang-PAMSe (Table R4) were injected via tail vein respectively, where PBS was injected into tail vein as blank control.

(2) After 24 h, the major organs of the mice including heart, liver, spleen, lung, kidney, brain and whole brain were collected, the surfaces were rinsed with PBS, blotted with filter paper and weighed. The tissues were then cut into grinding tubes and added to RIPA lysate at a ratio of 1 mL:100 mg tissue, and the tissues were ground with preset parameters on a tissue grinder to prepare a homogenate, centrifuged at 3000 rpm for 5 min, the supernatant was removed and fixed to 2 mL with RIPA lysate, and the fluorescence intensity of the material was measured using a fluorescence spectrophotometer (Ex: 650 nm, Em: 667 nm). The amount of material in each tissue was calculated based on the fluorescence intensity-concentration standard curve. The percent injected dose rate (%ID) was then calculated according to the following formula: $ID\% = (\text{mass of sample in organ tissue} / \text{total mass of injected sample}) \times 100\%$. As shown in Fig. 5b, compared to other samples, Ang-PAMSe accumulated significantly in the brain of mice. Moreover, quantitation of fluorescence in the brain and other organs showed that the brain has a maximum accumulation of 39% of the injected dose (ID) for Ang-PAMSe (Fig. 5b), which confirmed that Ang-PAMSe nanomotors showed better BBB penetration and GBM tissue accumulation compared to the three control treatments.

In addition, based on the reviewer's suggestion we also added a pharmacokinetic (PK) study to monitor the concentration of the drug in vivo versus time. The specific steps are as follows:

Healthy female SD rats (6-8 weeks) were injected with Ang-PAMSe/TLND (200 μL , 3 mg mL^{-1}) via tail vein and blood was collected from the orbit using capillary tubes at 1 h, 2 h, 4 h, 8 h, 12 h, 24 h and 48 h, respectively. The blood was centrifuged (2500 rpm, 10 min) and the serum was collected. 100 μL of serum was fixed to 1 mL with PBS and the TLND concentration was measured using a UV spectrophotometer. As shown in Supplementary Fig. 42, Ang-PAMSe/TLND nanomotor can maintain blood circulation for at least 48 h. Compared with the free drug TLND (*ACS Appl. Mater. Interfaces* 2021, 13, 26682), the blood circulation time of Ang-PAMSe/TLND was significantly longer. This result suggested that the nanomotor had good circulatory stability.

Table R4 Targeting performance of samples

Sample	Chemical recognition targeting	Chemokine targeting
Ang-PMSe	- ' I	×
PAMSe	×	- ' I
Ang-PAMSe	- ' I	- ' I

Fig. 5 | Assessment of targeting and therapeutic efficacy of Ang-PAMSe/TLND nanomotors in vivo. **a** Chemiluminescence imaging of GBM in C57BL/6 mice before administration and fluorescence images of the brains after injection of different cy5-labeled samples via tail vein injection (Scale bar: 2 cm). **b**

Quantitative analysis of sample accumulation in main organs. Cy5-sample levels were determined by fluorescence spectroscopy and expressed as injected dose per gram of tissue (%ID g⁻¹). **c** Representative CLSM images of brain tumor tissue in mice 24 h after injection of different samples via tail vein and red fluorescence intensity profiles as a function of the distance from a blood vessel in the representative region marked by the white rectangular frames. (Green, blood vessels are stained with CD31; red, samples are labeled with red fluorescence by cy5; blue, cell nuclei stained with DAPI; N: normal brain tissue; T: tumor; Scale bar: 100 μm); (The names corresponding to the samples in a-c are (I) PBS, (II) Ang-PMSe, (III) PAMSe, (IV) Ang-PAMSe). **d** Treatment protocols for orthotopic brain-GBM-tumor-bearing models. Where the red arrow is the time point of the drug administration and the blue arrow represents the time point of bioluminescence imaging. **e** Representative IVIS spectrum images and **(f)** quantified signal intensity (n = 3 biologically independent animals per group). **g** Survival analysis of the mice that loaded with GL261-Luc glioblastoma during treatment process (n=8, biologically independent animals per group). **h** Quantitative analysis of the DCs maturation level in vivo and **(i)** flow cytometry analysis of the expression of CD80 and CD86 on the surface of DCs extracted from the tumor draining lymph nodes of mice after various treatments. For the gating strategy for DCs maturation analysis refer to Supplementary Fig. 49. (n=3 biologically independent experiments). **j** Quantitative analysis of CD3⁺CD8⁺ cytotoxic T cells, and **(k)** representative flow cytometric plots of CD8⁺ T cells in GBM-bearing brain tissue gating on CD3⁺ cells in each group. For the gating strategy for T cells analysis refer to Supplementary Fig. 51 (n=3 biologically independent experiments per group). **l** Quantitative analysis of CD3⁺CD4⁺ T cells. **m** The quantitative analysis of CD3⁺CD4⁺Foxp3⁺ Tregs in the brain tissue of each group according to flow cytometry in each group. **n** Cytokine levels including IL-6 and **(o)** TNF-α in the serum of mice at the end of treatment (n=3). (The names corresponding to the group in e-o are (I) Sham, (II) PBS, (III) TLND, (IV) Ang-PMSe/TLND, (V)) Ang-PAMSe and (VI) Ang-PAMSe/TLND). Asterisk (*) denotes statistical significance between bars (*p < 0.05) using one-way ANOVA analysis.

Supplementary Fig. 42 In vivo pharmacokinetics of Ang-PAMSe/TLND nanomotor in healthy SD rats. Experimental data are shown as mean ± s.d. of samples in a representative experiment (n = 3).

7.- iNOS inhibition has been implicated in the reduction of glioma stem cell population. The authors should investigate whether the antitumor effect is due to decrease of the stem cell content of the GBM and not directly linked to an antitumor immune response.

Answer: Thanks a lot for the reviewer's question.

We complemented the assay by measuring the content of glioma stem cells (CD45⁺CD133⁺) in brain tumor tissues of tumor-bearing mice. The tumor-bearing mice (GL261-C57BL/6J mice, modeling time

of 10 days) were divided into 5 groups and dosed (PBS, Ang-PMSe, PAMSe and Ang-PAMSe) starting on day 11, every 2 days for a total of 5 doses, and brain tumor tissues were obtained from mice on day 20. The tumor tissues were cut with scissors, digested with collagenase type IV containing 0.1%, and then prepared into cell suspensions by sieve grinding, and the samples were resuspended by 40% and 70% percoll and centrifuged to collect non-lymphocytes and lymphocytes for use. FITC-anti-mouse CD45 (Biolegend) and PE-anti-mouse CD133-PE (Biolegend) were added to the collected tumor cell suspensions for specific labeling of tumor stem cells.

As shown in Fig. R14, there was no statistical difference in the change in the proportion of glioma stem cells in brain tumor tissues of mice treated with Ang-PMSe, PAMSe and Ang-PAMSe compared with the PBS control group.

Fig. R14 Representative flow cytometric plots of glioma stem cells in glioma-bearing brain tissue (n = 3 biologically independent experiments per group) and quantitative analysis of CD45⁺CD133⁺ tumor stem cells.

8.- Unacceptable language is used: “Mice were executed...”

Answer: Many thanks for the helpful comments from the reviewer.

We replaced the original counterpart with the following expression. “Lymph nodes and brain tumors of mice were collected for DCs maturation analysis on day 15 according to the procedure shown in Fig. 5a.” Also, we have checked and corrected the relevant expressions in the full text, and we thank the reviewers again for their help.

REVIEWERS' COMMENTS

Reviewer #1 (Remarks to the Author):

My comments have been properly addressed. I suggest a further language revision before acceptance.

Reviewer #2 (Remarks to the Author):

The authors nicely complied with the comments made and the concerns raised. I was however not able to open and access the video supplements. All in all, I feel the manuscript is ready for publication in Nat Comm and the authors are to be commended for this original and innovative paper.

Reviewer #3 (Remarks to the Author):

The authors have extensively dealt with the major comments raised by the reviewers. Concerning my questions, I feel the answers and additional experiments have addressed my main concerns. The manuscript now provides significant proof for the hypothesis of the authors.

Reviewer #4 (Remarks to the Author):

The revisions were satisfactory. I recommend the manuscript to be published in its current form.

REVIEWERS' COMMENTS

Reviewer #1 (Remarks to the Author):

My comments have been properly addressed. I suggest a further language revision before acceptance.

Answer: We thank the reviewer so much for his/her comment. We are also grateful for this opportunity to revise the problems in the manuscript. The language of the manuscript have been carefully revised. I hope our efforts can improve the readability of the manuscript and reach the publication level of Nature Communications.

Reviewer #2 (Remarks to the Author):

The authors nicely complied with the comments made and the concerns raised. I was however not able to open and access the video supplements. All in all, I feel the manuscript is ready for publication in Nat Comm and the authors are to be commended for this original and innovative paper.

Answer: We are pleased that the reviewers have given positive comments on the innovation of our work. We apologize for not being able to open and access the supplementary movie. We have uploaded the video again to ensure that it can be accessed normally.

Reviewer #3 (Remarks to the Author):

The authors have extensively dealt with the major comments raised by the reviewers. Concerning my questions, I feel the answers and additional experiments have addressed my main concerns. The manuscript now provides significant proof for the hypothesis of the authors.

Answer: Many thanks for the comments and suggestions of the reviewer, which are very helpful for improving the quality of the manuscript.

Reviewer #4 (Remarks to the Author):

The revisions were satisfactory. I recommend the manuscript to be published in its current form.

Answer: We thank the reviewer for his/her comment and recommendation to publish.